# Multimodal analysis demonstrating the shaping of functional gradients in the marmoset brain

Chuanjun Tong ⓘ [1,2,3], Cirong Liu ⓘ [2], Kaiwei Zhang [2], Binshi Bo [2], Ying Xia [2], Hao Yang [2], Yanqiu Feng ⓘ [1,3,4] ✉ & Zhifeng Liang ⓘ [2,5] ✉

The discovery of functional gradients introduce a new perspective in understanding the cortical spectrum of intrinsic dynamics, as it captures major axes of functional connectivity in low-dimensional space. However, how functional gradients arise and dynamically vary remains poorly understood. In this study, we investigated the biological basis of functional gradients using awake resting-state fMRI, retrograde tracing and gene expression datasets in marmosets. We found functional gradients in marmosets showed a sensorimotor-to-visual principal gradient followed by a unimodal-to-multimodal gradient, resembling functional gradients in human children. Although strongly constrained by structural wirings, functional gradients were dynamically modulated by arousal levels. Utilizing a reduced model, we uncovered opposing effects on gradient dynamics by structural connectivity (inverted U-shape) and neuromodulatory input (U-shape) with arousal fluctuations, and dissected the contribution of individual neuromodulatory receptors. This study provides insights into biological basis of functional gradients by revealing the interaction between structural connectivity and ascending neuromodulatory system.

The brain is a complex network of anatomically connected and functionally interacting neuronal populations[1]. Numerous studies have examined how the whole-brain network is parceled and organized[2]. Recent advances in human fMRI connectome have brought an analytic tool, i.e., large-scale cortical gradients, to capture the intrinsic dimensions of cortical organization[3]. The principal functional gradient in humans is the unimodal-to-transmodal spatial gradient, tracking a functional hierarchy from direct perception and action to integration and abstraction of information[4]. MRI studies of human cortex have revealed similar spatial distributions of cortical thickness[5] and myelin content[6]. The secondary intrinsic dimension reveals a visual-to-sensorimotor gradient which differentiates between the different sensory modalities[3,7]. The functional gradients provide a novel perspective of how the large-scale functional networks are organized[3,7–9].

In addition to functional gradients and hierarchy, recent studies in rodents and non-human primates described cortical hierarchy based on tract-tracing connectivity data[3,8,10]. Both interneuron density and inter-areal axonal connectivity vary along a functional hierarchy of cortical areas in mouse[8]. And also, retrograde tracing based structural connectivity (SC) of marmoset[10] and macaque[11] reveals that a gradient of brain networks hierarchically extends outward from primary cortices to progressively high-order transmodal association cortices. Although previous evidences have suggested the structural relevance of functional cortical organization, it remains unclear whether the

[1]School of Biomedical Engineering, Southern Medical University, Guangzhou, China. [2]Institute of Neuroscience, CAS Key Laboratory of Primate Neurobiology, Center for Excellence in Brain Science and Intelligence Technology, Chinese Academy of Sciences, Shanghai, China. [3]Guangdong Provincial Key Laboratory of Medical Image Processing & Guangdong Province Engineering Laboratory for Medical Imaging and Diagnostic Technology, Southern Medical University, Guangzhou, China. [4]Guangdong-Hong Kong-Macao Greater Bay Area Center for Brain Science and Brain-Inspired Intelligence & Key Laboratory of Mental Health of the Ministry of Education, Southern Medical University, Guangzhou, China. [5]Shanghai Center for Brain Science and Brain-Inspired Intelligence Technology, Shanghai, China. ✉e-mail: foree@smu.edu.cn; zliang@ion.ac.cn

structural connectivity supports the functional connectivity in gradient aspects.

Besides the stationary features of functional gradients, it is very likely that the large-scale cortical functional gradients are dynamic across multiple time scales, ranging from the time scales of evolution[12], lifespan[13,14] and even instantaneous fluctuations. Previous studies described the age-dependent gradient variations across human lifespan[13,14], highlighting the long-term gradient changes of cortical organizations across development. However, whether and how functional gradients vary instantaneously and in short time scale is much less clear. Studies have shown the rsfMRI activity involves dynamic reconfiguration into transient network states occurring on the time scale of seconds[15], often attributing to the arousal fluctuation[16,17]. Importantly, widespread variations in fMRI cortical activity are associated with the changes in the basal forebrain and midline thalamus[18,19], which are nodes of ascending neuromodulatory system[20]. Although previous studies demonstrate the arousal contributions on the function connectivity dynamics, it is yet to be determined whether arousal fluctuations influence the functional gradient dynamics.

While many studies have already examined the structural or arousal contribution to resting-state functional connectivity as mentioned above, very few of them directly examined whether and how those factors are related to (dynamic) FC gradients. A previous human study showed the structure-function tethering was heterogeneous and negatively correlated with the principal functional gradient[21]. Another mouse study found significant correlations between mouse functional gradients and gene expression patterns[12]. Although previous studies have suggested the link between functional gradients and structural connectivity or gene expression profiles, it remains unclear whether the functional gradients are dynamic with regard to arousal fluctuations, and if so, how structural connectivity and arousal related gene expression jointly contribute to such dynamics. And marmosets, as an emerging neuroscience animal model, is uniquely suited to address this question. Collaborative efforts in marmoset research have yielded large-scale datasets, including large awake rsfMRI data[22], comprehensive retrograde tracing based structural connectivity database[23] and in situ hybridization (ISH) gene expression database[24]. Such rich information provides unique advantage to examine the biological basis of functional gradients. Compare to the diffusion MRI derived structural connectivity[25] and limited tracing data in macaque[26,27], the 116 source and 55 target regions based on retrograde tracing[23] in marmoset represents a most complete, curated connectivity dataset in primates. Moreover, marmoset ISH database provides gene expression profiles at cellular resolution, while preserving key morphological and anatomical characteristics[24]. Combined with large awake rsfMRI data in marmoset, the functional gradient research in marmoset is uniquely posed to reveal its biological basis.

In this work, we set out to investigate whether and how structural connectivity and ascending neuromodulatory system shapes functional gradients, and ultimately, the dynamic functional gradients, based on multimodal analysis. We systemically characterized the marmoset functional gradients, which resembles the children cortical organization of human[13]. Combined with the marmoset retrograde tracing atlas[23], we revealed structural gradients strongly shaped the functional gradients. Furthermore, based on the marmoset gene expression atlas[24], we found the opposing effects on gradient dynamics between structural connectivity and ascending neuromodulatory system, as the neuromodulatory system provided higher modulation on functional gradients at the very low or high arousal levels. Finally, we showed that the axes of functional gradients were closely related to spatial patterns of gene expression for specific families of neuromodulatory receptors, which provides a biological substrate for the modulation of large scale functional dynamics. In summary, we demonstrate the strong structural basis of functional gradients and highlight the association between distinct neuromodulatory receptor families and large scale brain dynamics with instantaneously arousal fluctuations.

## Results

### Marmoset functional connectivity gradient and its structural basis

To investigate the intrinsic low dimensional topography of marmoset cortex, we decomposed the functional connectivity matrix from our dual-center awake resting state fMRI data into a set of gradients[3,8,9,12] via diffusion embedding mapping. The principal functional gradient, which captured the highest explained variance (ION: 29.00%, NIH: 30.61%) in the cortical functional connectivity (Supplementary Fig. 1), separated the sensorimotor cortex and visual cortex (Fig. 1a, b, Supplementary Fig. 2 and Supplementary Table 1). The second functional gradient (explained variance: ION: 17.13%, NIH: 14.10%) was anchored at one end by the sensorimotor and auditory cortex, while the other end were multimodal regions, part of which has been described as marmoset's default-mode network (DMN)[28] (Fig. 1a, b). The third functional gradient separated the anatomically defined sensorimotor cortex into two clusters, i.e., dorsal and ventral parts. The fourth functional gradient reflected the dimension between auditory and frontal pole network versus other regions. The above functional gradients were highly reproducible in both ION and NIH marmoset datasets (Supplementary Fig. 3), showing similar spatial topographies and high between-dataset correspondence (Fig. 1d). In addition, we systematically evaluated the individual variability and head motion effects on marmoset functional gradients, and found high stability across individual subjects (Supplementary Fig. 4) and minimal motion influence (Supplementary Fig. 5) on gradient results.

Furthermore, the principal gradient in marmosets closely resembled the second gradient identified in human adults (Fig. 1e and Supplementary Fig. 6a), showing an intrinsic visual-to-sensorimotor dimension[3,13]. Conversely, the second gradient in marmosets resembled the principal gradient in human adults, showing a multimodal-to-unimodal dimension. Using the fingerprinting method[29,30] (Supplementary Fig. 7), we quantitatively examined the similarity of the first two gradients between marmoset and human (children and adults), and found significantly higher similarity of the gradient order between marmoset and human children, compared to human adults (Supplementary Fig. 7). This result suggested a potential link between developmental and evolutionary processes[31]. Importantly, this cross-species gradient similarity was highly stable at the individual marmoset level (Supplementary Fig. 8), and remained stable using different preprocessing pipelines on fMRI data (Supplementary Fig. 9).

To investigate whether the structural connectivity governs the functional architecture in gradient space, we took advantage of a recently published marmoset retrograde tracing database, which provides a directional structural connectivity matrix[23]. The marmoset structural gradients exhibited highly similar spatial characteristics comparing to functional ones (Fig. 2a, Supplementary Figs. 1 and 2b), with the first and second structural gradients reflecting "visual-to-sensorimotor" and "multimodal-to-unimodal" dimensions, respectively. However, the third structural gradient showed less similar topographical profile, peaking on attention-related networks, including frontal parietal and middle temporal areas. The fourth structural gradient reflected the dimension between salience-like network and other brain regions (Supplementary Fig. 2). Quantitatively, first and second structural topographies were strongly correlated with their functional gradients, while the third and fourth gradients showed less similarity for both ION and NIH datasets (Fig. 2b, c). The above statistical significance of similarity of structural-functional gradients was corrected using the null distributions of spatial autocorrelation preserving surrogate maps (Supplementary Fig. 10).

In addition, we evaluated the similarity between functional gradients and other structural features, such as cortical thickness and

myelin patterns in marmosets and humans (Supplementary Fig. 11). Human cortical thickness and myelin gradients showed significant similarities with functional gradients, consistent with previous studies[5,7]. Intriguingly, such correlations between cortical thickness (and myelin maps) and functional gradients was not statistically significant in marmosets (Supplementary Fig. 11).

### Arousal relevant functional gradient dynamics

The convergence and divergence of structure-function gradient similarity suggested that the anatomical connectivity may not be the only contributor for functional gradients (Fig. 2b,c). Furthermore, the structure-function gradient similarity showed high scan-by-scan variability (Fig. 2d). One known factor contributing to resting-state dynamics is arousal fluctuation[19]. Therefore, we aimed to examine whether arousal fluctuation might partially account for dynamics of functional gradients. Using the 2nd-order polynomial fitting, we found an invert U-shape relationship (Fig. 2d and Supplementary Fig. 12) between the marmoset eye open ratio (used as a proxy for arousal levels[32,33]) and structure-function gradient similarity, prompting further investigations of arousal contribution.

Accordingly, we adopted a previous approach for inferring arousal fluctuations from fMRI data[16], achieving a refined frame-by-frame arousal estimation. Briefly, we generated the "arousal spatial template" from the correlation between voxel-wise BOLD signals and HRF-convolved pupil size variations (Supplementary Fig. 13a–c). After projecting the "template" onto successive fMRI volumes (Supplementary Fig. 13d), we obtained a continuous time series of estimated arousal level, termed "fMRI based arousal index" (Supplementary Fig. 13e). Significant correlation was observed between behavioral and fMRI-based arousal index at both individual scan level and group level (Supplementary Fig. 13f). In addition to being able to estimate arousal level during the eye closed condition[16], the above approach allowed to examine arousal contributions in NIH dataset that lacked marmoset pupillometry data. This method has been previously validated[16,34], and thus only fMRI based arousal index was used in later results.

Based on the above fMRI based arousal index, we binned dynamic function connectivity (dFC) matrices into 10 bins from low (drowsy) to high (alert) arousal levels (Fig. 3a). The dFC matrices were calculated by voxel-wise dynamic conditional connectivity[35]. These matrices within each arousal bin were averaged and used to derive the dynamic connectivity gradients. Interestingly, the explained variance (i.e., strength) exhibited an inverted U-shape relationship with arousal level for all four gradients (Fig. 3b and Supplementary Fig. 14), whereas the ordering of those gradients remained the same across arousal levels. In

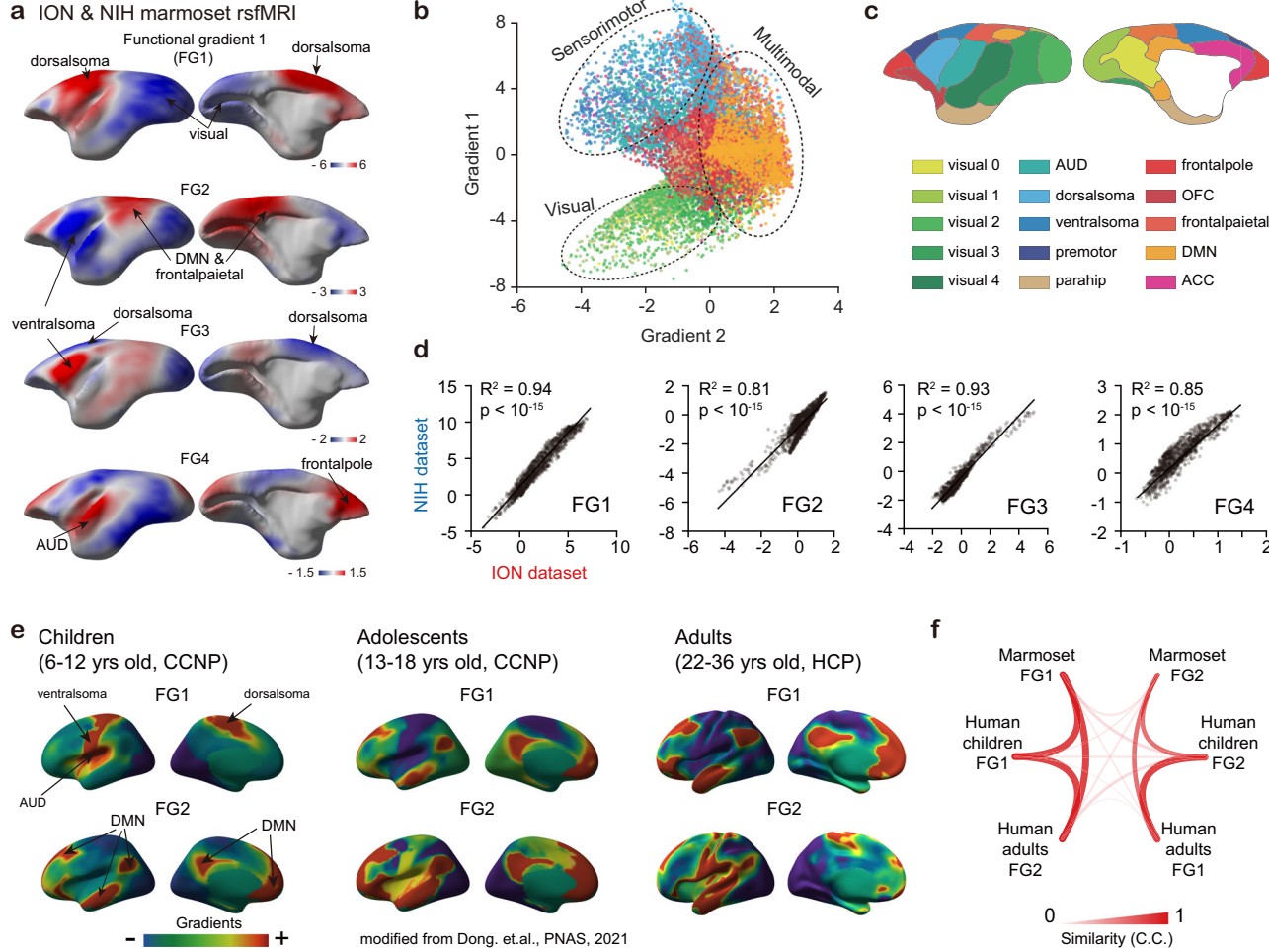

**Fig. 1 | Large scale functional connectivity gradients in marmoset and human.**
**a** First four gradients of marmoset function connectivity. FG, functional gradient.
**b** Scatter plot of marmoset first two function gradients. Each dot represented a cortical voxel, colored by the Fig. **c.** Gradient 1, "visual-to-sensorimotor" intrinsic dimension; Gradient 2, "multimodal-to-unimodal" intrinsic dimension. **c** Marmoset brain networks were parceled based on the MBMv4-Network[22]. The abbreviation of marmoset networks was summarized in Supplementary Table 1. **d** Cross-dataset comparison of first four gradients (two-tailed *t*-test). Each dot represented a cortical voxel. **e** First two human function connectivity gradients of children, adolescents and adults. The gradients of children and adolescents were modified from Dong et al.[13]. **f** Cross-species comparison between adult marmoset and developmental human in gradient aspect using the fingerprinting analysis method (details in Supplementary Fig. 7). C.C., Pearson's correlation coefficients. Source data are provided as a Source Data file.

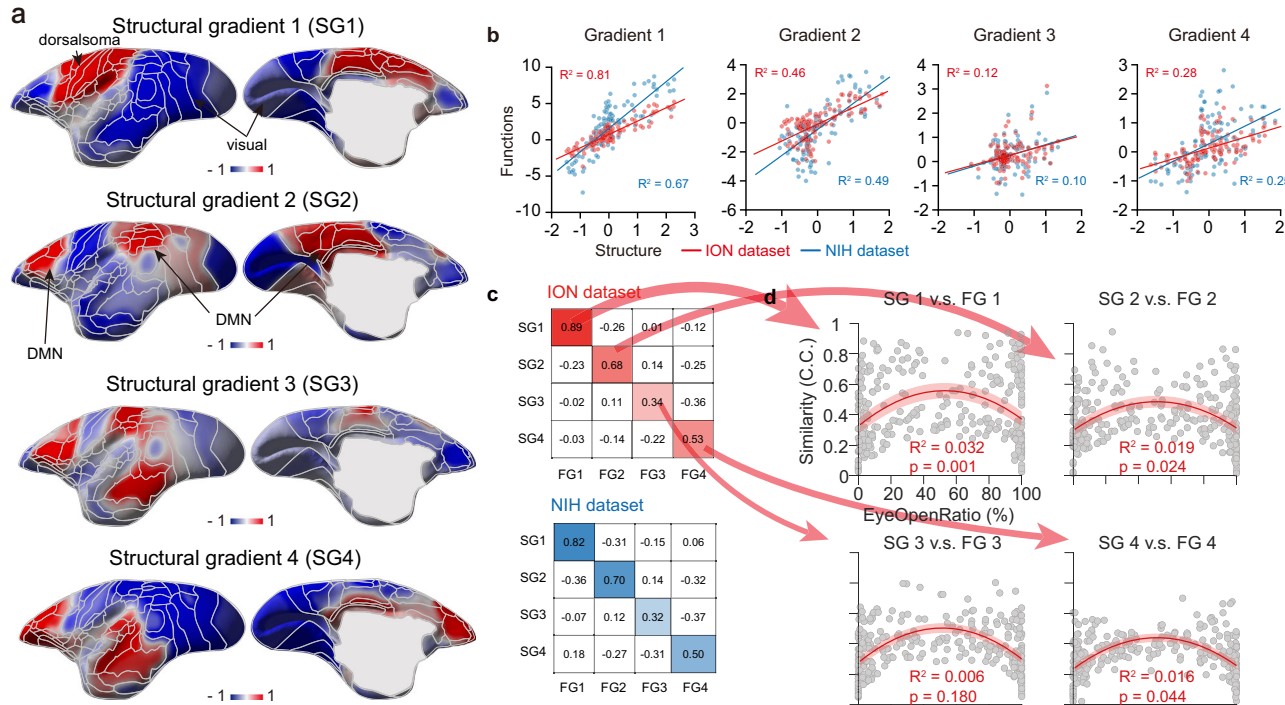

**Fig. 2 | Structure basis of functional connectivity gradients in marmoset. a** First four gradients of marmoset structure connectivity based on marmoset brain connectivity atlas[23]. Areal borders were based on the Riken Brain/MINDS cortical parcellation[70]. SG, structural gradient. **b** Topographical similarity between structure and function gradients (two-tailed *t*-test). Each dot represented a brain region. **c** High similarity (C.C., Pearson's correlation coefficients)

between low-dimensional structure and function gradients in both ION and NIH datasets. **d** Relatively low structure-function gradient similarity with extremely high or low eye open ratio per scan in ION dataset. Each dot represented one scan. Red solid line (or shade), the fitting curve (or the 95% prediction interval) of the structure-function gradient similarity. Source data are provided as a Source Data file.

addition, we conducted similar analysis on the sampled dFCs, i.e., picking the first frame every 10 s or 20 s, and observed consistent inverted U-shape relationship with arousal level for all four gradients (Supplementary Fig. 14). We next explored the trajectories of dynamic gradients by projecting the low dimensional topography into gradient space[14]. Although trajectories of fifteen brain regions showed stable topological properties in gradient space, they exhibited "flood and ebb" dynamics with arousal fluctuations (Fig. 3b, c). To quantitatively evaluate this phenomenon, the arousal index was shuffled by scan and then same analysis was applied to generate a null model control (Supplementary Fig. 15). We found most brain regions showed statistically significant shift of such arousal relevant flow compared to the null model (Supplementary Fig. 15), including ventral and dorsal somatosensory, primary to higher order visual, auditory, frontal pole, auditory, default model, mACC and premotor networks. We found that the BOLD functional connectivity (FC) was more heterogeneous at mid-arousal, compared to low and high arousal (Supplementary Fig. 16a, b) as evaluated using entropy of FC. The entropy of FC was significant correlated with the mean absolute strength of functional gradients, which was the average of the absolute gradient values across cortical voxels (Supplementary Fig. 16c). Furthermore, we found significant correlations between the entropy of FC and the relative distance in the gradient space (Supplementary Fig. 16d, e). These results suggested that the heterogeneity of FC is likely related to the "flood and ebb" effect of gradient dynamics with arousal fluctuations.

Likewise, the structure-function gradient similarity showed an apparent inverted U-shape relationship with arousal level (Fig. 3d). In addition, the above results of gradient dynamics were not significantly correlated with head motions across arousal levels (Supplementary Fig. 17).

Similar results (Fig. 3e, f) were also observed in human using human HCP dataset[36] and a previously published EEG-based vigilance

correlation coefficients map as the "arousal spatial map"[34] (Supplementary Fig. 18). The above result highlights the arousal modulation of gradient dynamics is likely to be conserved across species.

## Structural connectivity and neuromodulatory similarity collectively contributed to functional gradients

To quantitatively dissect how brain structure and arousal fluctuations influence functional gradients, we utilized the general linear model (GLM) to generate the predicted function connectivity[21,37] and derived the predicted functional gradients via diffusion embedding mapping (Fig. 4a). This GLM framework assumed that cortical function connectivity depends on structural and neuromodulatory inputs[38]. The relationship of structure factor is intuitive[1,21], i.e., brain regions that have strong structural connections are more likely to be functionally connected (Supplementary Fig. 19). In addition, neuromodulatory contribution was included in this model, as ascending neuromodulatory system is known to regulate arousal[20,39]. Thus, we hypothesized that brain regions which have stronger functional connections are more likely to exhibit similar neuromodulatory receptor expression profiles[38]. Such expression profiles were obtained from marmoset gene atlas[24] and the neuromodulatory similarity (NS) matrix was calculated using the correlation of gene expression level across each pair of brain regions[38] (Supplementary Fig. 20). We found the neuromodulatory similarity gradients were also correlated with the functional gradients, but to a less extent compared to the structural gradients (Supplementary Fig. 21).

The significant correlation (cvR$^2$ = 0.18 ± 0.002; mean ± sem) between predicted and empirical FC (Fig. 4b, c, "leave-one-out" cross validation) promoted us to investigate the characteristics of predicted gradients. The predicted functional gradient resembled the empirical one, achieving a significant correlation of $R^2 > 0.5$ (Fig. 4d). To examine the individual contribution of the two factors, single variable model

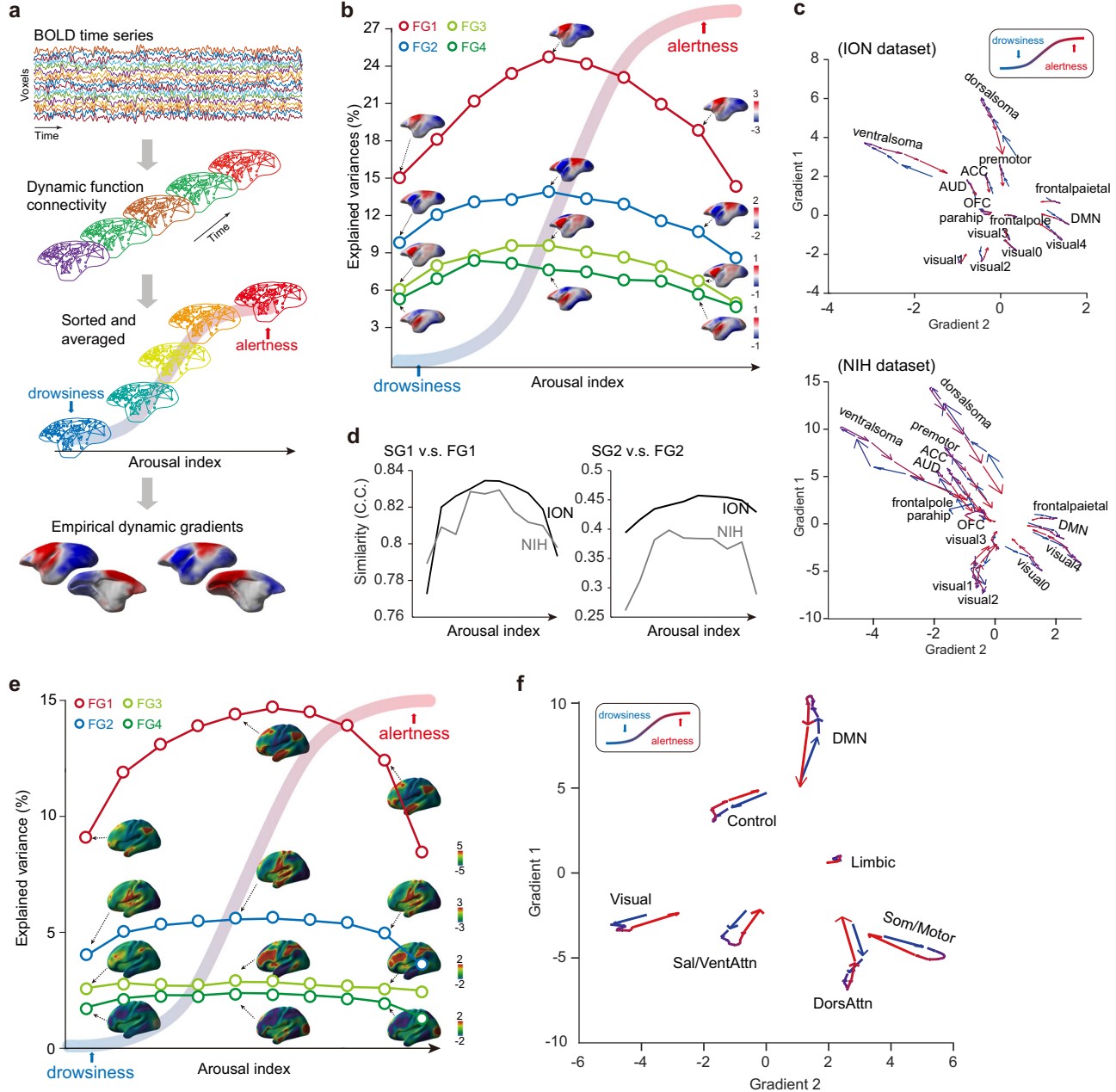

**Fig. 3 | Arousal fluctuation modulated functional gradient dynamics in marmoset and human. a** Pipeline for estimating arousal relevant dynamic function gradients. For each scan, we obtained dynamic connectivity matrices using the dynamic conditional correlation strategy[35]. After binned by the arousal index, connectivity matrices were averaged and projected into dynamic function gradients, respectively. **b** Inverted U-shape relationship between arousal level and explained variance across gradients. Details were shown in Supplementary Fig. 14. **c** Arousal relevant flow of each network in gradient space. Arrow reflected the direction of the shift along with arousal dynamics for both ION (upper panel) and NIH (lower panel) datasets. Statistical evaluation of these regional dynamics was shown Supplementary Fig. 15. **d** Inverted U-shape relationship between the arousal index and structure-function gradient similarities. C.C., Pearson's correlation coefficients. **e, f** As in (**b, c**) but for human functional gradients. The gradient values were parceled based on the Yeo et al.'s seven networks[2]. Source data are provided as a Source Data file.

was performed for structure connectivity and neuromodulatory similarity[37], and the predicted gradients from two factors alone still exhibited high similarities (Fig. 4d). This result suggested that explanatory power of these two factor may largely overlap[37,40] (Fig. 4e).

To address this issue, we utilized the reduced model to capture the unique contribution of each variable[40]. Notably, the usage of the "reduced model" was adopted from a previous study[40], in which unique contributions of various spontaneous behaviors to calcium imaging data in mice were dissected. In our reduced model, a particular variable was randomly shuffled 1000 times, and the resulting

difference compared to the full model, provides a lower bound for the unique contribution of each predictor (Fig. 4e, non-overlapping part), while the single variable model provides a upper bound of each predictor (Fig. 4e, circle). Supplementary Fig. 22 further illustrated the computational pipeline of functional gradients (dynamics) using the full general linear model (GLM) and reduced model. Quantitatively, we found structural connectivity (SC) provided more predictive power than neuromodulatory similarity (NS) in both single variable (Fig. 4f, $cvR^2_{SC} = 0.13 \pm 0.001$ v.s. $cvR^2_{NS} = 0.10 \pm 0.002$) and reduced models (Fig. 4f, $\Delta R^2_{SC} = 0.07 \pm 0.0005$ v.s. $\Delta R^2_{NS} = 0.04 \pm 0.001$). In contrast to

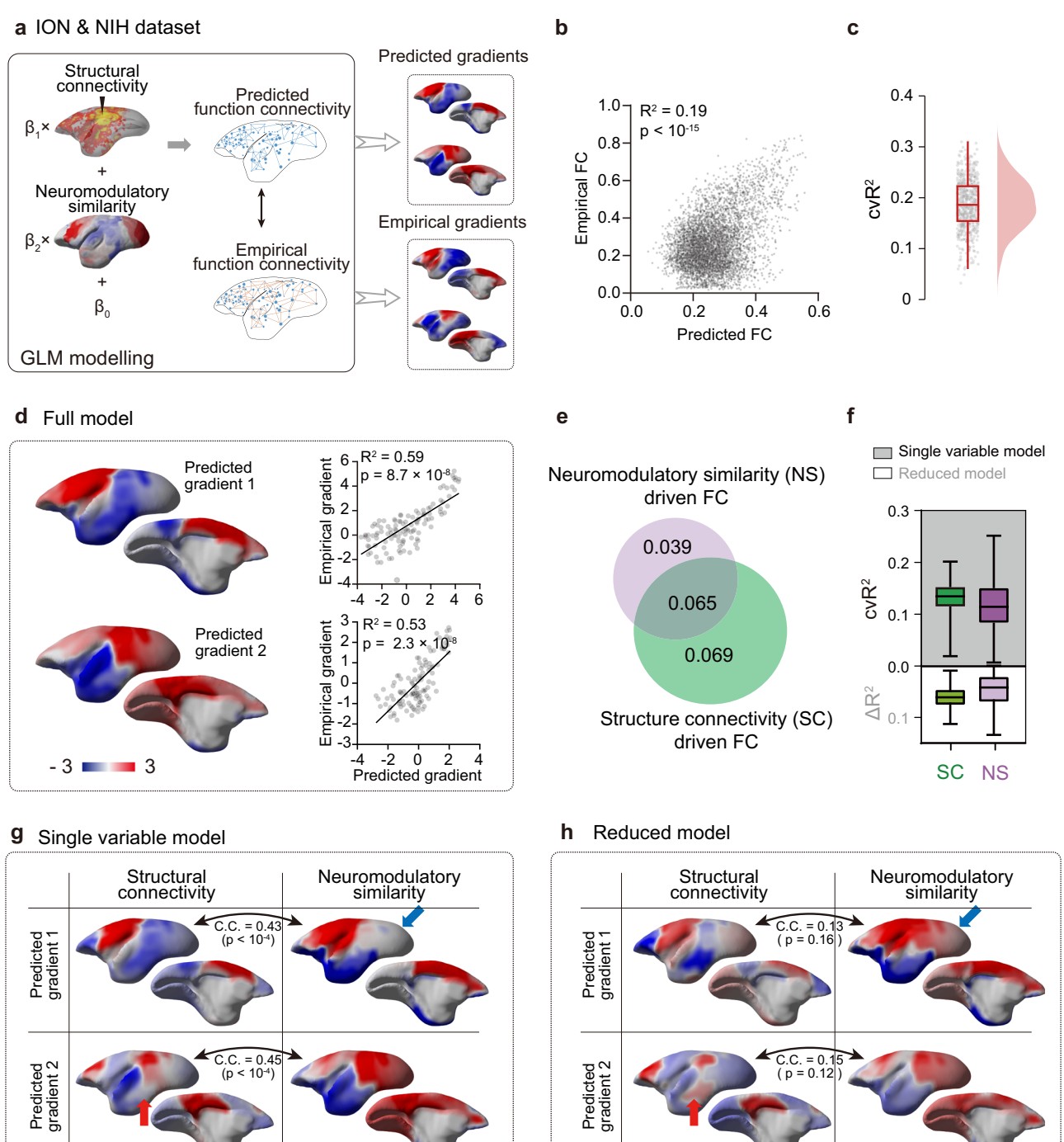

the predicted gradient maps from single variable model (Fig. 4g), the unique contribution of gradient maps in the reduced model exhibited highly spatially localized features (Fig. 4h). For the structure connectivity variable, the second gradient exhibited an apparent "multi-modal vs. unimodal" spectrum as well (Fig. 4h). Moreover, the second gradient values in temporal cortex for reduced model were higher than that of single variable model (Fig. 4h, red arrow), adding more similarity to the principal functional gradient (or DMN network) in human adults[3]. Moreover, for the neuromodulatory similarity variable, the first gradient maps for reduced model displayed a more homogeneous profile in dorsal regions compared with the single variable model (Fig. 4h, blue arrow). It suggests that the neuromodulatory influence is widely distributed across cortex, in line with the ubiquitous expression

of neuromodulatory receptor genes[20,39,41]. As expected, the reduced model clearly reduced the overlap and allowed us to disentangle the unique contribution of structure connectivity and neuromodulatory similarity, respectively.

## Ascending neuromodulatory system modulated functional gradient dynamics under structural constraint

We next extended the modelling framework to the dynamic regime. Predicted dFC matrices were binned by arousal indices and were used to derive the arousal relevant dynamic gradients (Fig. 5a). Then, these predicted dFC matrices were fitted to the empirical ones using frame-by-frame GLM modelling. The predicted gradients exhibited similar topological properties to those of empirical ones (Supplementary

**Fig. 4 | Reduced GLM model dissected unique contributions of structural connectivity and neuromodulatory inputs to the functional gradients.**
**a** Computational strategy of function gradients modeling. We utilized a general linear model (GLM), in which the dependent variable was the empirical function connectivity (FC) and the independent variables were the structure connectivity[23] (details in Supplementary Fig. 19) and neuromodulatory receptor similarity[24] (details in Supplementary Fig. 20). Then, the resulting predicted FC was projected into gradients via the diffusion embedding methodology and compared with the empirical gradients. **b** Significant correlation between predicted and empirical FC (two-tailed $t$-test). **c** Cross-validated explained variance (cvR$^2$) across all EPI runs ($n = 709$ runs). Each dot represented an EPI run. The box showed the first and third quartiles; inner line was the median over EPI runs; whiskers represented minimum and maximum values (outliers removed). **d** High performance of function gradients prediction. The predicted function gradient (left panel) were significantly correlated (right panel) with the empirical one (two-tailed $t$-test). Each dot represented a brain region based on the Riken's marmoset

parcellation[70]. **e** Reduced model for investigating the unique distribution of each variable (non-overlapping part). The structure connectivity or neuromodulatory receptor similarity (circle) may have overlapped information with the other one, thus the reduced model (non-overlapping part) provides the unique contribution of each predictor. **f** Top: cross-validated explained variance (cvR$^2$) maps for different single-variable models. Bottom: unique contribution ($\Delta$R$^2$) maps for the same variables. The box showed the first and third quartiles; inner line was the median over EPI runs ($n = 709$ runs); whiskers represented minimum and maximum values (outliers removed). **g** Low dimensional topography contributed by structure connectivity and neuromodulatory similarity, respectively. Significant spatial correlation indicated largely overlaps between predictors (two-tailed $t$-test). C.C. Pearson's correlation coefficients. **h** As in (**g**) but for reduced model. Distinct gradient profile of the unique contribution from structure connectivity and neuromodulatory similarity exhibited non-significant spatial correlation (two-tailed $t$-test). Source data are provided as a Source Data file.

Fig. 23), suggesting good modelling performance. The explained variances in full model exhibited an inverted U-shape relationship with arousal level in first and second predicted gradients, showing a striking similarity to the empirical results (Fig. 5b). Moreover, the gradient flows were very similar to the empirical ones (Fig. 5c). In contrast to full model derived gradient maps, the unique contribution for gradients in reduced model exhibited distinctively spatial topographies of structural connectivity (Fig. 5d) and neuromodulatory similarity (Fig. 5e), respectively. Importantly, the explained variances in reduced model between the two variables revealed opposing relationship with arousal level. Unlike the inverted U-shape for structural connectivity, the neuromodulatory similarity showed a positive U-shape relationship with peak explained variance at the two extremes of the arousal fluctuation (Fig. 5d, e, right panel). To further evaluate whether the U-shape modulation was specific to the neuromodulatory similarity (NS), we conducted three control analysis (Supplementary Fig. 24) by (1) adding a noise matrix (NM) as another variable, (2) replacing the NS with random ISH expression similarity (RS) and (3) replacing the NS with glutamate receptor ISH expression similarity (GluS) (see Supplementary Table 2–4). The results showed the U-shape modulation was more pronounced to neuromodulatory receptors, compared to the above three control variables (NM, RS and GluS).

Likewise, the gradient flows of the unique contribution of structural connectivity and neuromodulatory similarity presented opposite directions with arousal fluctuations (Fig. 5f, g). The gradient flow of structure connectivity (Fig. 5f) showed a similar trajectory to that of full model, but it was the opposite for the flow of neuromodulatory similarity (Fig. 5g). In conclusion, the structure topography served as the backbone of the functional gradients, while the neuromodulatory similarity modulated the dynamics with higher contribution at the two extremes of arousal fluctuations (Fig. 5h).

## Neuromodulatory receptors differentially modulated functional gradient dynamics

Arousal fluctuations typically involve multiple co-varying neuromodulatory systems or receptors. Thus, we sought to dissect the individual modulations of neuromodulatory receptors on functional gradient dynamics. To obtain the unique contribution of each neuromodulatory receptor, we utilized the reduced model similar to Fig. 5, in which we spatially shuffled a particular receptor expression map 1000 times and obtained the surrogate neuromodulatory similarity matrix, respectively.

We ranked the contribution strength of receptors according to the mean explained variance (Fig. 6a) and significance level (Supplementary Fig. 25). Because the expression patterns of neuromodulatory receptors are spatially auto-correlated, we adopted a procedure from previous studies[42–44] to overcome this issue and test the significance level of a given receptor (Supplementary Fig. 25). Then, we applied the

false discovery rate (FDR) correction on all receptors and found several receptors showing statistically significant contributions on the gradient dynamics with arousal fluctuations (Fig. 6b and Supplementary Fig. 26). With these approaches, we observed four groups of neuromodulatory receptors regarding their relationship with the first two functional gradients (Fig. 6b). Four receptors (Group 1) positively loaded onto the gradient 1 (sensorimotor-to-visual) in marmosets, including dopaminergic (DRD1 and DRD4), noradrenergic (ADRA2A) and cholinergic receptors (CHRM3). Five receptors (Group 2) positively loaded on the gradient 2 (multimodal-to-unimodal), including dopaminergic (DRD3), noradrenergic (ADRA1A), serotonergic (HTR1B and HTR2A) and cholinergic receptors (CHRNA6). Two receptors (Group 3) positively loaded onto both gradients, both being cholinergic (CHRM1 and CHRM5) receptors. The rest of receptors (Group 4) showed no significant contribution to functional gradient dynamics.

Moreover, to investigate the low dimensional features among those groups of neuromodulatory receptors, we applied principal component analysis (PCA) to neuromodulatory receptor maps (Fig. 6c). PC1 was significantly correlated with both "sensorimotor-to-visual" and "multimodal-to-unimodal" gradients in marmoset, suggesting a homogeneous distribution of neuromodulatroy receptors across the cortex[20,39] (Fig. 6d, green lines, Fig. 6e). PC2 represented "Group 1 & 3" dominated features (Fig. 6d, orange lines, Fig. 6e), showing significantly correlation with the primary "sensorimotor-to-visual" gradient only, and "Group 2 & 3" dominated PC3 (Fig. 6d, blue lines, Fig. 6e) with the secondary "multimodal-to-unimodal" gradient only. The spatial similarity between neuromodulatory PC maps and functional gradients (Fig. 6d, e) confirmed that large scale cortical dynamics of different modalities were modulated by the recruitment of distinct neuromodulatory receptor classes[45].

In summary, our results demonstrated that while functional gradients are strongly shaped by structural connectivity gradients, arousal translates the static anatomical wiring into dynamic functional configurations via the ascending neuromodulatory system.

## Discussion

Mechanistic understanding of large-scale functional dynamics requires multimodal integration of functional, structural and molecular signatures. The current study systematically combined our large dual-center marmoset resting-state fMRI dataset[22] with marmoset structural connectivity atlas[23] and marmoset gene atlas[24], to uncover the large-scale functional organization and its biological underpinnings in the marmoset brain. Our result revealed marmoset functional gradients exhibited human children-like functional organization and were powerfully shaped by structural gradients. Furthermore, the functional gradients showed a "flood and ebb"-like dynamics with arousal fluctuations, in which structure connectivity and neuromodulatory system exhibited opposing effects. At the two extremes (very drowsy and

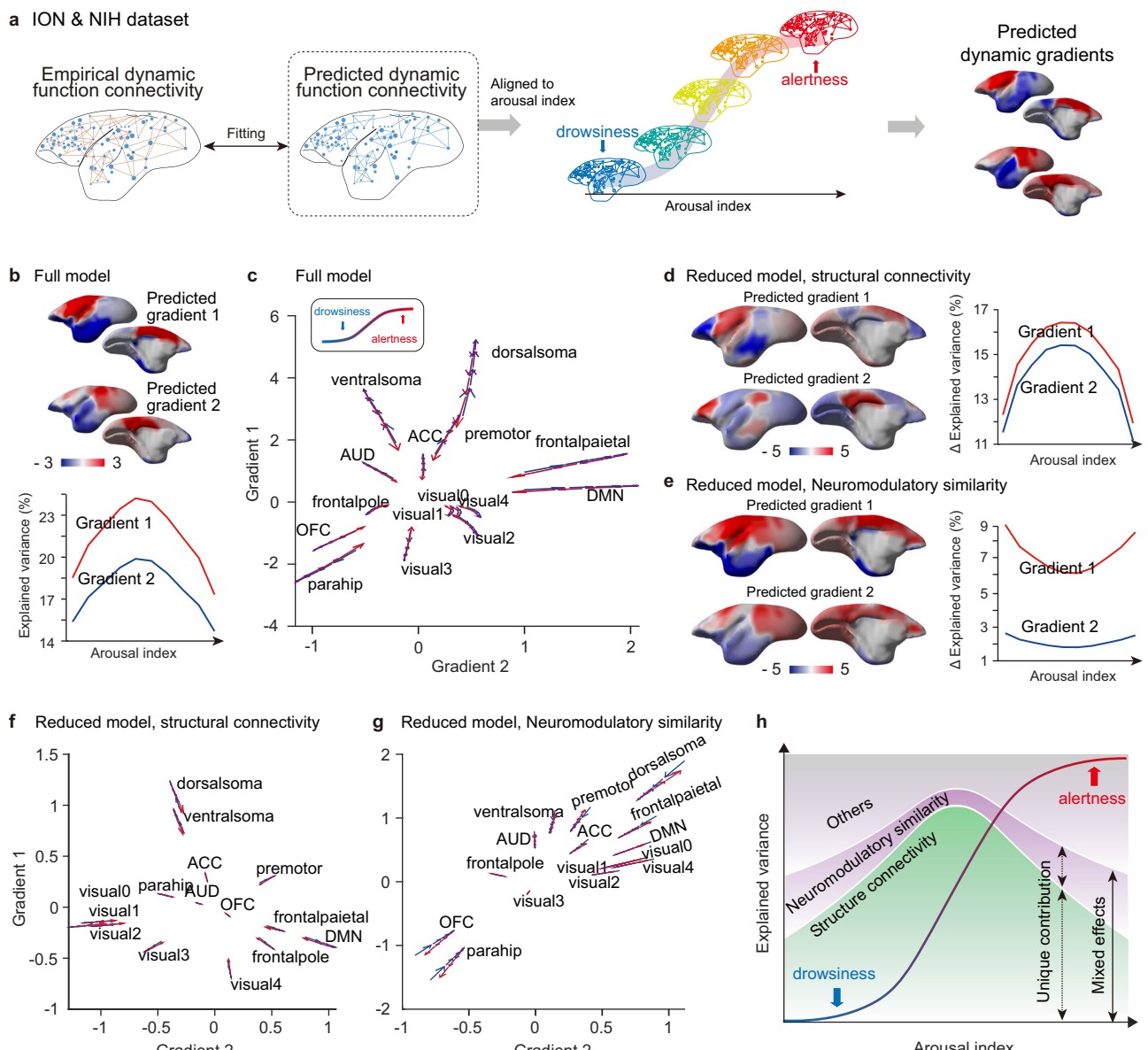

**Fig. 5 | Opposing effects on gradient dynamics between structural connectivity and ascending neuromodulatory system. a** Pipeline for the dynamic function gradients prediction. Briefly, we fitted the empirical dynamic function connectivity (dFC) frame-by-frame utilizing the same GLM strategy as Fig. 4a. After binned by the arousal index, the predicted dFC matrices were averaged and projected into dynamic function gradients, respectively. **b** Predicted dynamic function gradients and corresponding explained variances (Full model). Details were shown in Supplementary Fig. 23. **c** Arousal relevant flow of each network in gradient space (Full model). Arrow reflected the direction of the shift along with arousal dynamics. **d**, **e** As in (**b**) but for the unique distribution of structure connectivity (**d**) and neuromodulatory similarity (**e**), respectively. **f**, **g** As in (**c**) but for the unique distribution of each predictor. Notably, the flow of arrows was in an opposite way between structure connectivity **f** and neuromodulatory similarity **g**. **h** Conceptual summary of the underlying basis of complex global functional processing in large scale cortical organization. Source data are provided as a Source Data file.

alert) of arousal fluctuations, neuromodulatory contribution was elevated, while the contribution from structure connectivity was partially suppressed. Moreover, our results revealed the receptor specific neuromodulatory modulations on large scale functional topographies.

The marmoset functional gradients, highly reproducible across individual animals and between ION and NIH datasets, revealed a spatial arrangement for functional specializations of different modalities, sharing an analogous gradient space with human children[13]. Unlike the cortical organization of macaque[11] and human[3,11], marmoset exhibited a sensory specialized profiles ("sensorimotor-to-visual" gradient dominated), rather than global processing hierarchies[7] ("unimodal-to-multimodal" gradient dominated). This finding resonates with a recent study of mouse functional gradients[12] which showed a

principal spatial progression from archicortex (hippocampus) to palecortex (piriform area), in line with the dual origin theory of cortical evolution[46]. A previous study suggested that the massive expansion of multimodal areas in humans has untethered these area from the influence of molecular gradients that constrain the organization of sensory regions[47], which may in part explain the functional gradient differences across species.

Despite an increasing number of observations and modelling strategies on hierarchical structure-function relationship with the underlying cortical microstructure[8] and gene expression[8,37], it remains unclear whether and how the functional organization arise from structural constraint in large scale gradient aspect. We observed spatial localized cortical structural gradients utilizing a comprehensive marmoset

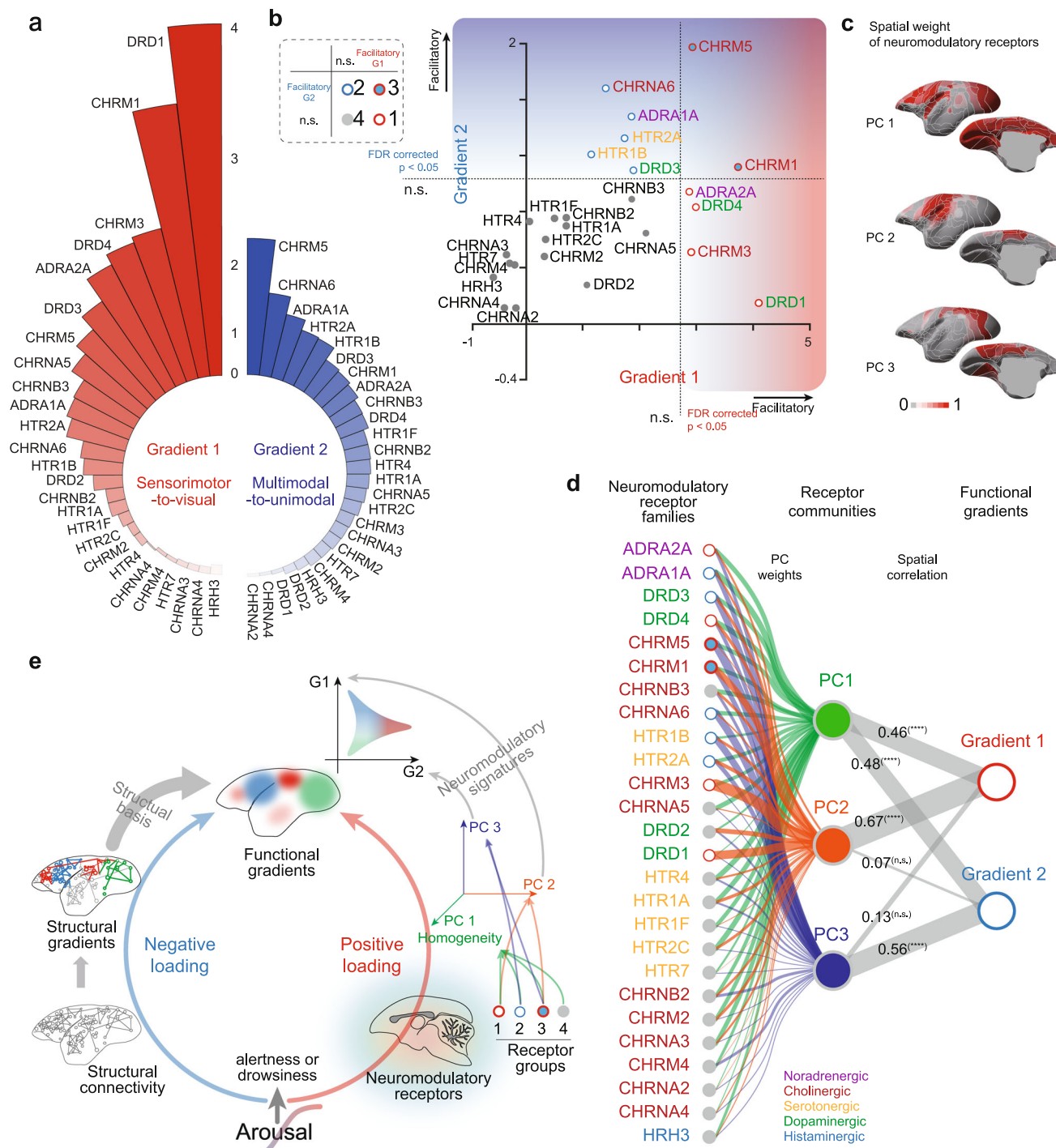

**Fig. 6 | Receptor specific neuromodulatory signatures of functional gradient dynamics. a** The unique contribution on explained variance of first two gradient dynamics for neuromodulatory receptor maps. The bar was ranked and scaled according to the mean explained variance (%) of each receptor map. **b** Scatter plot of neuromodulatory receptors according to the unique contribution on explained variance of first two gradient dynamics. The receptors were divided into four groups based on the statistical significance (FDR corrected $p < 0.05$, right-tailed $t$-test). **c** Spatial maps for the first three principal components of neuromodulatory receptors. **d** Associations between functional gradients and neuromodulatory receptor communities. The color rings of receptors were based on the inset in **b**.

The lines were scaled according to the absolute principal component coefficients (green, orange and blue) or spatial correlation coefficients (gray, Pearson's correlation). ****, $p < 10^{-4}$; n.s., no significance. **e** Large scale functional gradients were strongly shaped by structural wiring and dynamically modulated by ascending neuromodulatory system. At the extremes of arousal fluctuations (very drowsy or alert), both group 1 and group 3 receptors (PC2 dominated) positively loaded onto the "sensorimotor-to-visual" gradient (G1), while group 2 and group 3 receptors (PC3 dominated) positively loaded onto the "multimodal-to-unimodal" gradient (G2). Subsequently, the strong structure-function gradient correspondence was partially suppressed. Source data are provided as a Source Data file.

structural connectivity matrix[23]. Thus, we presented new evidence that functional gradients were highly constrained by structural wiring in gradient aspect. However, whether such strong coupling is specific to marmoset or it also applies in other species such as human[25] remains to be answered. We found weaker correspondence of structure-function gradients (Supplementary Fig. 6c) in human compared to the result of marmosets. For example, a noticeable difference in the angular gyrus (ag) of principal structural (Supplementary Fig. 6b) and functional gradients can be clearly observed. Such disparity may be in part attributed to inaccuracy in diffusion MRI based tractography, as it is well known that diffusion tractography suffers from low reliability[48]. In contrast, retrograde tracer injection in marmoset yielded precise structural connectivity that can be considered gold standard[23,49].

While the functional gradient is highly constrained by structural wiring, the current study provided empirical evidence that such gradient organization is dynamic. Previous studies found functional connectivity fluctuates over brain states[15], but whether such arousal related FC dynamics modulate the macroscopic topography is rarely reported. This recent human task fMRI study[50] suggested that among well rested, sleep deprived and sleep recovered states, there was very minor changes of functional gradients, suggesting low contribution of arousal modulation on the functional gradients. This might be related to the difference of fMRI paradigms (tasked based v.s. resting state) and the resulting difference between task regressed gradients and resting-state gradients. Nevertheless, in the current study the inverted U-shape relationship between arousal level and gradient values in the present study presents first clear evidence of arousal modulation on functional gradients. As the gradient values could be considered as a spectrum of connectivity similarity, the gradient value reflects difference of functional connectivity patterns. From low (drowsy) to intermediate (possibly quiet awake) arousal levels, the increase of gradient values indicated larger separations among brain networks[14]. In dynamic functional connectivity analysis, the global signal removal caused anatomically heterogeneous increases in functional connectivity and its dynamics[51]. Thus, the increased inter-network separations might be contributed by the global decrease of functional connectivity[17]. The reduction in whole-brain connectivity has been identified from N1 (and N2) sleep to awake state in human EEG-fMRI study[52], which may contribute to the above mentioned increasing trend of gradient values in the left half of the U-shaped dynamics. Yet, from intermediate to high (vigilant) arousal levels (i.e., the right half of the U-shape), the decrease of functional gradients indicated higher degree of integration among brain networks. Previous studies indicated the lysergic acid diethylamide (LSD) enhanced global and between-module communication while diminished the integrity of individual modules[41], and such effect is mediated by the brain's key integration centers that are rich in $5-HT_{2A}$ receptors in human[41]. Furthermore, chemogenetic locus coeruleus (LC) activation increased whole-brain functional connectivity in mouse[53], accompanied by significant pupil size increase[32]. The above results suggested the activation of ascending neuromodulatory system leads to the increases of global functional connectivity, which provides a putative substrate for our finding in the right half of the U-shape. Overall, at the extremes (drowsy and alert) of arousal level, higher neuromodulatory contribution results in the increase of global functional connectivity, and the resulting less inter-class discriminations of whole-brain connectivity might contribute to lower functional gradient values.

A recent study revealed that the infra-slow global waves of resting-state fMRI signals propagates along the primary gradient in humans, and these propagations are highly sensitive to the brain arousal state[54]. Such propagation of the infra-slow waves may be related to both anatomical connectivity among cortical hierarchy and ascending neuromodulatory system, providing more plausibility that arousal may translate the static anatomical wiring into dynamic functional configurations via the ascending neuromodulatory system.

As we presented empirical evidence of how structure wiring and neuromodulatory system shape functional gradients, it is important to further disentangle their contributions quantitatively[41]. Past studies developed a series of modelling strategies on the link of structural and functional connectivity, achieving improved model performance[21,37,42]. However, in these models multicollinearity among predictors was often ignored, potentially leading to unreliable and unstable estimates. In the current study, we focused on characterizing the unique distribution of structural connectivity and neuromodulatory information under a GLM framework. Our approach was in line with a previous wide field calcium imaging study in mouse[40], in which the difference in explained variance between the full GLM and the reduced model yielded the unique contribution of the corresponding predictor. We observed significant spatial similarity between two predictors using single variable model, further confirming the multicollinearity of model inputs. The association between those two inputs are not unexpected, as structural covariance was associated with transcriptomic similarity[38]. Using the reduced model approach, higher predicted gradient value in the temporal cortex was found for the second gradient of structural connectivity, adding more similarity to the DMN of macaque[55] and human[56]. Meanwhile, the principal gradient of neuromodulatory similarity (unique contribution) exhibited higher cortical homogeneity, consistent with our prior knowledge of neuromodulatory system[20,57]. Therefore, our approach of the reduced model allowed more precise dissection of unique contributions of structural and neuromodulatory inputs to the functional macroscopic topography.

Interestingly, our reduced model revealed a U-shape relationship between the unique contribution of neuromodulatory similarity and arousal fluctuations, while full model and the unique contribution of structural connectivity showed an inverted U-shape relationship with arousal. This result is in agreement with studies reporting higher neuromodulator releases in sleep[57] and active awake state[32], compared to quiet awake state. And also, intracellular recordings in awake behaving rodents revealed a U-shape dependence of average membrane potential and cortical activation on arousal[58]. Therefore, this result suggested the neuromodulatory system may contribute more to the functional gradients either at very low or very high arousal level, conferring temporal dynamics to functional macroscopic topography under the structural constraint[1,41].

The ascending neuromodulatory system is highly inter-connected and individual receptor system often co-varies with arousal[59], thus it is difficult to dissect the specific contribution of each arousal nucleus or neuromodulatory receptor on large-scale functional dynamics. Utilizing the reduced model approach with shuffled individual receptor map, our study was able to disentangle the co-variation among receptor systems. For the dopaminergic receptors, *DRD1* and *DRD4* showed significant contributions to sensory specialization (gradient 1), with only *DRD3* contributing to hierarchical processing (gradient 2). Studies using genetic knock-out mice reported similar effects of dopaminergic receptor subtypes, including *DRD1* mediated motor dysfunctions[60], *DRD3* related higher-order spatial working memory[61] and *DRD4* related specific exploration[62]. Moreover, a number of studies revealed tight associations between whole brain neural activities and neuromodulatory receptors, including LSD mediated global integration[41] (*HTR2A*), chemogenetic LC-activation induced increase of brain network communications[53] (*ADRA1A* and *ADRA2A*) and reduction of REM and NREM sleep[63] and *CHRM3* knockout mice. The significant effects on sensory specialization or hierarchical processing were consistent with our findings of unique and individual contribution of each receptors (Fig. 6b) on functional dynamics. Notably, computational work linked neuromodulatory system to the alteration of the brain state and cognitive performance[45]. Our results provided evidence for these concepts, and further dissected the mechanisms of each neuromodulatory receptor on cortical activities.

In the current study, we used the fMRI based arousal index based on previous work in macaque[16] and human[34]. Our marmoset arousal template showed widespread significant negative correlations across cortex, which closely resembled the previous result in macaque[16]. The fMRI based arousal estimation provides an avenue to infer arousal fluctuations from fMRI data alone when external measures are not available in scanner, such as EEG (ION and NIH dataset) or pupilometery (NIH dataset). The previous study suggested that fMRI based arousal index was sensitive to brain states and showed high correlation with the electrophysiological arousal index[16]. Nevertheless, unlike the standard EEG based arousal measurement, this approach detects primarily relative, rather than absolute, arousal index across each scan. In addition, as fMRI signals are often detrended to remove scanner artifacts, such index may not detect slower baseline shifts of arousal level. Another limitation is that the present resource of marmoset structural connectivity[23] does not provide full and unbiased whole-brain coverage, which reduced our ability of accurate modelling.

Finally, for the cross-species comparison of the functional gradient characteristics in marmosets and humans, several factors are different across the animal and human datasets, which may complicate such comparison. For example, the high prevalence of nested family relationship in the human HCP dataset does not exist in our marmoset dataset. However, as we have shown in Supplementary Figs. 4 and 8, the overall spatial patterns and the ordering of the functional gradients in marmoset are relatively stable at the individual level, so it's unlikely that family relationship difference would significantly affect our cross-species comparison. Also, the current study is limited by the fact that functional connectivity, structural connectivity and gene expression data from different marmosets, and as such, age, sex and the individual differences may limit our inference. In particular, the functional connectivity and structural connectivity data were both from adult marmosets, while the gene expression data were largely from infant or juvenile marmosets (Supplementary File 1). Such age difference may potentially lead to biases across the three data types. Nevertheless, we examined expression patterns of *DRD1* and *CHRM3* (Supplementary Fig. 27) and found relative stable patterns across age and sexes. Future detailed examination is required to systemically investigate the age and sex dependence of the neuromodulatory receptor gene expression, especially when adult gene expression data become more readily available. In addition to age and sex, the individual difference may also lead to potential instability in our results. However, due to the nature of the tracer injection and ISH experiments, it is not feasible to collect all data from one single animal and will require technical improvement in the future, such as spatial transcriptomics.

In conclusion, through multimodal analysis of functional, structural and molecular datasets in marmoset monkeys, we revealed the structural basis and arousal modulation of the large-scale functional gradients in the awake marmoset brain. Those results provide concrete and specific insights of the global functional organization, and provide a solid foundation for utilizing marmosets for studying large-scale functional dynamics and arousal. The current study also opens a number of new research directions for future work. First, the cross-species comparison of functional gradients requires further comprehensive investigations. For example, whether the strong coupling of structural and cortical gradients holds for other species, particularly human, remains unclear. Second, with increasingly sophisticated tools available in marmosets, it would be beneficial to extend the current framework to neurological and psychiatric marmoset models, to examine the pathological impacts on the functional gradients and their biological underpinnings.

## Methods

### Animals and MRI scanning

A dual-center (ION and NIH) marmoset (*Callithrix jacchus*) resting state fMRI (rsfMRI) dataset was utilized including 39 adult marmosets with 709 17-min functional scans (12 males and 1 female were from ION, age $3 \pm 1$ years old; and 19 males and 7 females from NIH, age $4 \pm 2$ years old). The experimental procedures were approved by the Animal Care and Use Committees from the Institute of Neuroscience (ION) at the Chinese Academy of Sciences and National Institute of Neurological Disorders and Stroke at the National Institutes of Health (NIH).

All rsfMRI data followed a standardized imaging protocol to ensure consistent data quality. All marmosets underwent a 3-to-4 week acclimatization protocol as previously described[64]. Briefly, in the first week, only body restraining was applied with an increasing period from 15 to 60 min. In the second week, recorded MRI noise was added and habituation periods gradually increased to 120 min. In the third week, head fixation using the customized helmet was added. After the 3-week training period, marmosets were fully acclimated to lay in the sphinx position, with their heads comfortably restrained by 3D-printed helmets[22].

Briefly, un-anesthetized marmosets were scanned in horizontal MRI scanners (ION, 9.4 T/30 cm; NIH, 7 T/30 cm, Bruker, Billerica, USA, software ParaVision for MRI acquisition). For each session, multiple runs of rsfMRI data were collected using 2D gradient echo EPI sequence with the following parameters: TR = 2000 ms, TE = 18 ms (ION) or 22.2 ms (NIH), flip angle = 70.4°, FOV = 28 × 36 mm, matrix size = 56 × 72, 38 axial slices, slice thickness = 0.5 mm, 512 volumes per scan. Two sets of spin-echo EPI with opposite phase-encoding directions (LR and RL) were also collected for EPI-distortion correction with following parameters: TR = 3000 ms, TE = 37.69 ms (ION) or 36 ms (NIH), flip angle = 90°, FOV = 28 × 36 mm, matrix size = 56 × 72, 38 axial slices, slice thickness = 0.5 mm, 8 volumes for each set. After each rsfMRI session, a T2-weighted structural image was acquired for co-registration with following parameters: TR = 8000 ms (ION) or 6000 ms (NIH), TE = 10 ms (ION) or 9 ms (NIH), flip angle = 90°, FOV = 28 × 36 mm, matrix size = 112 × 144, 38 axial slices, slice thickness = 0.5 mm.

Additionally, in the ION dataset, an infrared MR compatible video camera (sampling rate of 60 fps, 12 M or 12M-I camera, MRC Systems GmbH) inside the bore was used to record the pupil size of the animal, which was later used to estimate the behavior arousal level (details in Behavior arousal index below).

### Marmoset fMRI preprocessing

After data format conversion, EPI distortion correction was applied using FSL's *topup*. The marmoset brain was extracted manually using ITK-SNAP (http://www.itksnap.org/). All subsequent procedures were performed using custom scripts in MATLAB 2020a (MathWorks, Natick, MA) and SPM12 (http://www.fil.ion.ucl.ac.uk/spm/). First, each fMRI scan was registered to the scan-specific structural image using rigid body transformation and the scan-specific structure image was then nonlinearly transformed to a study-specific marmoset template[65] (https://marmosetbrainmapping.org/atlas.html).

After the registration, the resting state fMRI data were further regressed by 22 "nuisance signals" to reduce motion artifacts, including 6 head motion parameters, their 1st order derivatives[66] and 10 non-brain tissue based principal components (PCs)[67]. We also conducted parallel analyses on data with ICA-FIX de-noising[68] and obtained very similar results. The main results reported in this study were from the regression based de-noised data only. A light spatial smoothing (0.5 mm FWHM isotropic) and a band-pass filter (0.001−0.1 Hz) were also performed. The BOLD signals were normalized by subtracting its temporal mean and dividing by its temporal standard deviation on a voxel-by-voxel basis.

### Human fMRI preprocessing

We used the HCP 500-subject data release, which includes 526 subjects with eye-open (https://www.humanconnectome.org/). We restricted our analysis to 469 subjects (age = $29.2 \pm 3.5$, range: 22−36, 275

females) who have all four sessions of resting-state fMRI in full length, resulting 1755 runs in total. The acquisition parameters were described in details in previous studies[36].

The HCP data were preprocessed using the HCP MR minimal preprocessing pipeline, which combines a set of tools from FSL, FreeSurfer, and the HCP Connectome Workbench. After the minimal preprocessing pipeline, the resting-state fMRI data were further denoised using the ICA-FIX method[36,68]. In addition to the preprocessing steps implemented by the HCP, we applied smoothing both spatially (Gaussian filter with the FWHM = 2.4 mm) and temporally (band-pass filtered at 0.005–0.1 Hz). Following preprocessing, the mean time series was extracted from 1000 predefined cortical regions-of-interest using Schaefer. et al. human parcellation[2].

### Gradient analysis
Diffusion embedding mapping was a nonlinear dimension reduction method, seeking to project a set of "symmetric" connectivity or similarity matrix into low-dimensional space upon the Markov chain on the network[3,12,13].

Voxel-wise function connectivity (FC) matrices were first generated for each scan by calculating the Pearson's correlation coefficient between any two pair of voxels. Scan-wise FC matrices were next averaged across scans to form a study-specific FC matrix. Consistent with previous studies[3,12,13], only the top 10% connections were retained and others in the matrix were set to 0. The resulting asymmetric matrix was converted into normalized cosine angle matrix and nonlinearly reduced the dimensionality via the diffusion embedding mapping. The gradients were ordered by the explained variance. To determine the arousal relevant characteristics of connectivity gradients, scan-wise embedding solutions were aligned to the study-specific gradients via Procrustes rotations[9]. The Procrustes alignment enabled comparison across scan-wise results and provided the original data is equivalent enough to produce comparable Euclidean spaces[14].

### Behavior arousal index
To reduce the computing load, the eye monitoring videos were down-sampled from 60 fps to 6 fps. We adopted a U-Net architecture for pupil segmentation[33] in 2D grayscale images and achieved reproducibly accurate segmentation outputs. Then, the median value within the time bin corresponding to each fMRI volume (TR = 2 s) was calculated, yielding a time series sampled at the same rate as the fMRI data. Thus, fast blinks (less than 0.5 s) was most likely excluded in this process and very unlikely to affect further analysis. For each scan, this time series was subsequently normalized by dividing by the maximum value, corresponding to the eye being fully open, so that the units were rendered comparable across sessions despite slight variations in the positioning of the camera relative to the eyes. The resulting normalized variation of pupil size was termed as the "behavior arousal index".

### fMRI template based arousal index
fMRI based arousal index was calculated using a previously established approach[16]. The "arousal spatial template" was generated by the correlating between resting state BOLD signal and hemodynamic response function (HRF) convolved pupil size on a voxel-by-voxel basis. This arousal spatial template was used to calculate the spatial correlation between each successive fMRI frame and this template, and the resulting time series of correlation was termed fMRI based arousal index[16,34].

### Dynamic function gradient
We calculated the dynamic function connectivity using dynamic conditional connectivity on a voxel-by-voxel (Fig. 3e) or region-by-region (Fig. 5a) basis[35]. Dynamic connectivity was computed using the dynamic conditional correlation approach (https://github.com/canlab/Lindquist_Dynamic_Correlation), a multivariate volatility

method[35]. Briefly, the dynamic conditional correlation model was used to deal with the temporal autocorrelation and non-stationarity in fMRI time-series. This model assumes that the brain time courses follow a multivariate Gaussian distribution, and that the conditional mean, variance and co-variances change in an autoregressive form. Unlike sliding-window approaches that estimate connectivity over a fixed window length, this is a model-based method that estimates the contribution of surrounding time points to the covariance matrix. Pairwise dynamic connectivity values were obtained for every time point of each resting-state run. This resulted in a matrix of connectivity values that was M (time points) × N (connections) for each run. We binned the dynamic function connectivity matrices according to arousal index with equal samples, yielding same degree of freedoms. Then, dynamic function connectivity matrices were averaged across bins and mapped to low dimensional space via the diffusion embedding methodology.

### Marmoset structural connectivity
The marmoset structural connectivity matrix was obtained from Marmoset Brain Connectivity atlas (https://www.marmosetbrain.org/) and the procedure of generating the connectivity matrix was described in details previously[23]. Briefly, the raw data include 143 injections of retrograde tracers in 52 young adult (1.4–4.6 years, median age: 2.5 years, 31 males, 21 females), and standard histological procedure was applied. Digitized histological sections were 3D reconstructed and registered to a template. Injection sites and retrograde labeled cells were assigned to cortical areas based on the atlas parcellation. And finally the structural connectivity matrix was generated by compiling data from all injection experiments. Notably, the primary marmoset neuronal tracing connectome matrix, comprising of 116 source and 55 target areas, was directional and not a square matrix. The unidirectional tracing connectome matrix was transformed by logarithm operation, and then Pearson correlation between each pair of regions was calculated to generate the symmetric similarity matrix, which was used as input to calculate structural connectivity gradient (Supplementary Fig. 19).

### Marmoset neuromodulatory similarity
The neuromodulatory receptor information was obtained from the marmoset gene atlas database (https://gene-atlas.brainminds.riken.jp/). Registration of marmoset ISH images to MRI space was summarized in Supplementary Fig. 20. Briefly, we downloaded the Nissl stained coronal images and neuromodulatory receptor related gene expression maps. Next, we mapped the receptor expression maps to the Nissl stained images, using "rigid-body transformation" for the coarse whole brain registration and "large deformation diffeomorphic metric mapping[49,69] (LDDMM)" for more subtle slice-by-slice registration. To facilitate the comparison with fMRI results, the Nissl stained images were registered to the study-specific MRI template by affine nonlinear transformation ("oldnormalize" of SPM12), and the affine transformation matrix was then applied to the receptor expression map to bring it to the MRI space. Also, the median filter was applied to each set of ISH data to improve the data quality. Next, receptor expression data were parcellated into 116 cortical regions of interest, based on the Riken Brain/MINDS cortical parcellation[70]. Finally, the resulting neuromodulatory receptor similarity was calculated using the correlation of gene expression level across each pair of regions.

### General linear model
A general linear model was used to predict the functional connetivity[21,37]. The predictors were modified structural connectivity and the neuromodulatory receptor similarity matrix. The model was then constructed as

$$FC = b_0 + b_1 \times SC + b_2 \times NS, \quad (1)$$

where the output variable FC was the set of whole brain functional connectivity (116 × 116 regions), and the input variables were modified structural connectivity (SC) and neuromodulatory receptor similarity (NS). The regression coefficient $b_0$, $b_1$ and $b_2$ were then solved by ordinary least squares techniques with Euclid norm constraint. The resulting best fit of empirical FC was termed as the predicted FC, i.e., the linear combination of the regression coefficients and corresponding input variables (SC and NS).

We constructed three GLM models (full model, single variable model and reduced model) to dissect the contribution on functional gradients from structural connectivity and neuromodulatory similarity[40] with the "leave-one-out" cross-validation. For the full model, all variables were included in the GLM model (Fig. 4e, the union of two circles). The single variable model only included one variable in the GLM model (Fig. 4e, circle). However, we concluded that the two predictors shared large overlap, evidenced by high correlation between predicted gradients from two single variable models (Fig. 4d). To address this issue, we constructed the reduced model to capture the unique contribution of each variables by applying random shuffling to a particular variable (1000 times). The resulting loss of explained variance captured the unique contribution of the corresponding variable (Fig. 4e non-overlapping part). Notably, variable shuffling was better than directly removing one variable, as it keeps the same degree of freedom as the original model[40]. The single variable model provided an upper bound for the given variable, while the reduced model provided a lower bound for the unique contribution of the corresponding variable.

In addition, the "reduced model" in our results was not meant to refer to the "reduced model" as conventionally defined in linear regression, i.e., "restricted model" ($y_i = b_0 + \varepsilon_i$) vs. the full general linear model, i.e., "unrestricted model" ($y_i = b_0 + b_1 x_{i1} + \varepsilon_i$). The reduced model we used was defined as the difference between the full GLM model and the "randomly perturbed" GLM model ($y_i = b_0 + b_1 \tilde{x}_{i1} + \varepsilon_i$) in which a particular variable $\tilde{x}_{i1}$ was randomly shuffled 1000 times. The full GLM and reduced model and their applications on functional gradients (and dynamics) were further illustrated in Supplementary Fig. 22.

### Significance test using the spatial autocorrelation preserving shuffling

Because profiles of functional gradients and neuromodulatory receptors are spatially auto-correlated, we adopted a procedure from previous studies[42–44] to overcome this issue and generate statistical significance.

To evaluate the significance level of structural-functional gradient correlation (Supplementary Fig. 10), we generated surrogate maps that randomly varied in their particular topographies ($n = 1000$ times shuffling) but preserved the general spatial autocorrelation (SA) structure. Using null distributions generated from SA preserving surrogate maps, we generated the significance level of empirical structural-functional gradient similarity.

To evaluate the significance level of a given receptor's contribution in Fig. 6b, we firstly generated surrogate maps of corresponding receptors (Supplementary Fig. 25) using the same procedure above. We calculated the Pearson's C.C. (right tail) between the surrogate similarity (to the empirical one, $n = 1000$) and the unique contribution on functional gradients of a particular receptor[42]. If the correlation is not significant, it indicates random receptor maps could contribute similar arousal modulation, i.e., the empirical receptor does not specifically contribute to arousal dynamics. Alternatively, if the correlation is significant, it indicates larger spatial map shuffling causes larger loss of the unique contribution for the corresponding receptor, i.e., the empirical receptor does contribute to arousal dynamics.

### Reporting summary

Further information on research design is available in the Nature Research Reporting Summary linked to this article.

## Data availability

The dual center resting-state functional MRI dataset is publicly available via Marmoset Brain Mapping Resource website (https://marmosetbrainmapping.org). The raw resting-state MRI data are provided in the standard BIDS format for cross-platform sharing. The marmoset neuro-tracing data (https://www.marmosetbrain.org/) and gene expression data (https://gene-atlas.brainminds.riken.jp/) was published previously as open resource. All human fMRI data are provided by the Human Connectome Project (https://www.humanconnectome.org/). The source data underlying Figs. 1–6 and Supplementary Figs. 1, 3–17, 19, and 21 are provided as a Source Data file. Source data are provided with this paper.

## Code availability

Codes used in this study are available at https://github.com/TrangeTung/marmoset_gradient. The Zenodo DOI for this code is https://doi.org/10.5281/zenodo.7215504.

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

## Acknowledgements

We thank the Marmoset Animal Facility of CEBSIT for animal care. This work was supported by the National Science and Technology Innovation 2030 Major Program (2021ZD0200100 to Z.L.), Strategic Priority Research Program of Chinese Academy of Sciences (XDBS01030100 to Z.L.), Pioneer Hundreds of Talents Program from the Chinese Academy of Sciences (to Z.L.), Shanghai Municipal Science and Technology Major Project (2018SHZDZX05 to Z.L.), the National Natural Science Foundation of China (82171899 to Z.L., U21A6005 to Y.F.), Lingang Laboratory (LG202104-02-06 to Z.L.), Key-Area Research and Development Program of Guangdong Province (2018B030340001 and 2018B030333001 to Y.F.).

## Author contributions

Y.F. and Z.L. designed and supervised the study; C.T., K.Z., B.B., Y.X., and H.Y. collected the MRI data; C.T., C.L., Y.F., and Z.L. preprocessed and organized the MRI data; C.T., C.L. and Z.L. preprocessed the neuronal tracing data; C.T., Y.F., and Z.L. conducted the computational modeling. C.T., Y.F., and Z.L. wrote the original draft and revised the draft.

## Competing interests

The authors declare no competing interests.
