## [Peer Review File · Nature Communications]

Multimodal analysis demonstrating the shaping of functional gradients in the marmoset brainREVIEWER COMMENTS

Reviewer #1 (Remarks to the Author):

Thank you for inviting me to review this manuscript by Tong and colleagues, in which the authors use gradient-based methods to analyse resting state fMRI data from awake marmoset monkeys and link the resultant low-dimensional descriptions of the data to white-matter and genetic expression maps. While this is an exciting example of the using of multimodal neuroimaging data to test hypotheses regarding structure-function relationships in the brain, many of the hypotheses appeared somewhat exploratory. In addition, I was concerned that the interpretation of the primary results may have been related to methodological choices, which is a factor that I encourage the authors to rule out with subsequent analyses.

Major comments

- How were the marmosets kept still enough to allow for the collection of BOLD data without substantial motion artifacts?
- P6L94: There are perhaps earlier/better references for the following statement: “ often attributing to the arousal fluctuation22-24.”, such as <https://www.ncbi.nlm.nih.gov/pmc/articles/PMC4843437/> and <https://pubmed.ncbi.nlm.nih.gov/27693256/>
- P8L122: The following line requires statistical support: “We systemically characterized the marmoset functional gradients, which resembles the children cortical organization of human”. See below for further explanation of this issue.
- P9L163: I'm not sure that it's sufficient to state that the gradients in marmoset “closely resembled” patterns in the human. Is there some way to quantify this? Is it only the average maps for which this is true, or do individual marmoset gradients also share the same relationship? I realise that this comparison (i.e., the mapping between human and marmoset) is non-trivial, however given that this is one of the major claims of the manuscript, I was expecting to see a more formal statistical comparison between the maps.
- How distinct was the % explained variance for gradients 1 and 2 in the marmoset data?
- How sensitive was the ordering of the gradients to different potential sources of noise, such as head motion? I could imagine that both human children and awake marmosets move quite a lot in the scanner, so either head motion, or the statistical approaches instigated in order to mitigate against its potential effects could potentially explain why certain gradients were more obviously present/absent in these two cohorts.
- Figure 2D – the curve fits in Figure 2d appear slightly problematic to me. Most of the data appears to exist at the two ‘extremes’. Is there a better way to fit the data, or better yet, to include an estimate of certainty into the fits that is proportional to the amount of data in each bin?
- Figure 3C – these analyses are really nice, however it's hard to know precisely what to infer from the data, particularly given the lack of an appropriate null model that affords an estimate of just how large a shift is to be expected from chance.
- P13L243 - the term “flood and ebb” should be better defined for the reader. There are numerous statistics (such as the distribution of dwell times) that can be used to quantify these concepts, and hence provide robustness to the conclusion.

Minor comments

- Abstract: “In this study, we investigated [the] biological basis of large-scale functional gradients by combining a large awake resting-state fMRI dataset of marmoset monkeys with retrograde tracing and gene expression databases.
- Could the authors please expand on the ‘dynamic conditional connectivity’ analysis used in the study?
- How did the authors account for blinks in their pupillary analysis?

Reviewer #2 (Remarks to the Author):

The study investigated the principal gradients of brain connectivity using multimodal data from marmosets with focusing on anatomical and neuromodulatory contributions. Briefly, the principal gradients were first computed for both functional connectivity measured by resting-state fMRI correlations and structural connectivity assessed by retrograde tracers, and they showed similar spatial topography. The authors also showed that the principal gradients of resting-state fMRI connectivity (i.e., FC gradients) are modulated by arousal levels of the animals, and thus developed a modeling frame to differentiate the contributions from the structural connectivity (SC) and the neuromodulatory similarity (NS), which was assessed by inter-regional correlations of neuromodulatory receptor expression. While the SC-predicted FC gradients showed a similar dependency on the arousal level as the FC gradients, the NS-predicted FC gradient had opposite modulations. Lastly, the paper showed that the spatial distributions of neuromodulatory receptors are related to the topology of the FC gradients. Overall, the study provides an interesting perspective on the potential link between the neuromodulatory system and the FC gradients, with the use of a variety of functional, structural, and genetic data. The part dissecting the contribution of the anatomical and neuromodulatory contributions to the FC gradients is particularly novel. However, I do find some potential technical issues, including in this critical part. My specific comments/questions are shown as below

1. My biggest concern is about the modeling frame, particularly the reduced model, used for the results presented in Figures 4 and 5, which are the major innovation of this study. According to the Methods section, this reduced model is very similar to the single variable model. The only difference is a better control of the degree of freedom with shuffling the other variable. By comparing with the full model, one can estimate the unique contribution of the variable on the response variable (here should be FC) based on the change of explained variance, just like in the ref #60 cited by the paper. But I don't believe it was used properly in this paper. Both Figs. 4d and 4f compared the gradients of predicted FC from these two variables (i.e., the SC and NS), and Fig. 4e is just an illustration without any quantifications about how much SC and NS contribute to the predicted FC and thus the resulting FC gradients.

For this part of the analysis, the authors should show how much of the variance of the response variable, i.e., the FC, is explained by two independent variables (i.e., the SC and NS). Or, if the principal gradient of the FC is of the interests, the authors need to show the similarity between the empirical gradient and predicted gradient derived from the single variable model or the reduced model, as what they did for the full model in Figure 4C.

I also encourage the authors to check the results shown in Figs. 4d and 4f. As discussed above, the reduced model should produce very similar results as the single variable model by adding a randomly shuffled variable. This appears to be true based on the visual inspection on the gradients. They look indeed very similar to each other. However, it is surprising that the spatial similarity of the gradients from the predicted data was changed dramatically.

2. The principal gradient of the neuromodulatory similarity should be presented and compared with the FC gradient, similar to what they've done for the structural connectivity (Fig. 2). The gradient maps of the neuromodulatory similarity shown in Figures 4d and 4f do not appear to be similar to the FC gradients to me.

3. The gradient dynamics shown in Figure 5 is somewhat expected, especially if the neuromodulatory similarity doesn't produce similar gradients as the FC. The predicted FC (fitted FC) is the projection of empirical FC onto a subspace spanned by the independent variables, i.e., the SC or NS, and thus contained the information of real data. As shown in Figure 2, the SC gradients are highly similar to the FC gradients. It is thus not surprising to see a similar invert-U shape of modulation of SC-predicted gradient on the arousal index. At the same time, one would expect an opposing U-shape modulation

of the gradients for any other variables that are not similar to the FC gradients, which could be the case for the NS-predicted FC gradients. In other words, the U-shape modulation should not be specific to the NS-predicted FC gradients but to anything not having a tight link with the FC gradients, as the “others” shown in Figure 5h.

4. There is a very relevant paper the authors may want to include since the major topic of this paper is about how the FC gradients are related to arousal and the neuromodulatory system. It has been shown recently (Gu, et. al. Cerebral Cortex 2021) that the FC gradient is related to the infra-slow global waves of resting-state fMRI signals that propagate along the FC gradient directions, and these propagations are highly sensitive to the brain arousal state. These findings may help understand how the FC gradients form and why they are sensitive to arousal fluctuation. Thus, it would be appropriate to include some discussion about this.

5. I also have some questions regarding the last part of the results related to the receptor maps. First, the spatial correspondence between the receptor maps and the FC gradients are very weak with a R-square of 1~2% (Figs. 6 and S12). The authors showed some of these spatial correlations are statistically significant. But there were two potential issues. 1) These brain maps are spatially continuous, which means that the parcels/voxels are not independent and thus the real degrees of freedom should be much smaller than the number of parcels/voxels. Therefore, simply shuffling the parcels/voxels for the receptor maps is not a correct way of building the null distribution. 2) The reported results were not corrected for multiple comparison.

Second, if the paper meant to emphasize the role of the neuromodulatory system in generating the FC gradients, the proper controls need to be included. The authors need to show that the neuromodulatory receptors show a stronger spatial correspondence with the FC gradients than other non-neuromodulatory receptors. Also, the spatial similarity itself is not a strong indicator of a tight relationship. For example, it has been shown that the primary gradient of gene expression in the brain is similar to the FC gradient in human (the FC gradient #2 in this paper) (Burt, JB et. al., Nature Neuroscience, 2018). One would thus expect that a significant proportion of genes have significant spatial correlation with the FC gradient. However, it doesn't mean that all of these genes would play a role in generating this FC gradient.

Reviewer #3 (Remarks to the Author):

The manuscript addresses dynamic changes in functional organisation of the primate cortex, using a low dimensional account of resting state functional connectivity (“gradients”). Particularly interesting is the focus paid to the role of arousal on functional organisation. Additionally, the study attempts to calculate the contribution of structural connectivity and similarity in receptor expression to the spatial pattern of the principle functional axes.

One major issue is that the methods are incomplete, such as the analyses in Figure 6, and the methods lack key details that would be necessary to replicate the study. For example, how are receptor expression maps “mapped” to the Nissl stained images, and how were Nissl stained images “registered” to the study-specific MRI template?

The conclusions often overstate the findings or rely upon qualitative, subjective comparisons. For example, comparison of gradients with functional networks and comparison of gradients across species are based on visual inspection, but should be empirically evaluated. Another example is that the authors suggest that Fig 2d is an inverted U-shape, but it appears relatively flat with high variance. Again, this should be tested statistically, for example by evaluating the fit of a quadratic model to the data. In terms of overstatement, the variance explained by individual receptors is less than 1% (shown in Supplementary Figure), yet these effect sizes are not mentioned in the main text and their negligible contribution is further dissected. It would be important for the authors to integrate statistics

(e.g. p values, R2, t-statistic, dof) into the main Results to ameliorate this issue.

The terms functional gradient, core axis, functional hierarchy and intrinsic coordinate system are mixed throughout the manuscript without clear definitions. I would suggest that the authors clarify the terminology and are more cautious in their interpretations. For example, the functional gradients are one possible intrinsic coordinate system [see also neurodevelopmental (Nieuwenhuys et al.,) or phylogenetic (Goulas et al.,) perspectives]. Defining the principle functional gradient as “the core axis of human intrinsic coordinate system” appears to dismiss that there is still contention regarding how the cortex is organised.

It is interesting to inspect the arousal-related variations in the functional gradients (e.g. Figure 3f). It would be worthwhile comparing results to Cross et al., (2021) that looked at sleep deprivation. As the results are represented in a low-dimensional space, it is difficult to interpret the cause of the “ebb and flow” effect. One possible explanation is that BOLD timeseries are more heterogeneous at mid-arousal, compared to drowsy and high arousal. But it could also be due to the alignment procedure, which may be biased towards the mid-arousal state, because the study-specific template is an average across different states of arousal. The authors could greatly improve the interpretability of the “ebb and flow” (which is the most interesting insight of the paper in my opinion) by further deconstructing the root cause with secondary analyses of the timeseries and the alignment procedure.

How the predicted functional connectivity is generated is not clear from the Methods, however, I would advise to split the data into train and test subsets, so as to evaluate the predictive performance out of sample. This would also benefit the comparison of models.

Reviewer #4 (Remarks to the Author):

Tong et al. Anatomical and neuromodulatory basis of large-scale functional topography

The authors combined resting-state fmri with retrograde tracing and gene expression data. They observed (i) different gradient order in monkeys, resembling findings in infants, (ii) inverted u type associations to structural connectivity, and (iii) u-shape associations to arousal level.

1) In the introduction, the authors highlight structure function correspondence at the level of structural and functional features but i believe that more specificity may be warranted when referring to cortical thickness and myelin gradients. In particular, there seems to also be data emphasizing that structural gradients may not exactly follow the spatial pattern as the the functional gradients, despite some overall correspondence.

2) Currently the study appears justified primarily by the availability of different complementary resources, and not so much by a conceptual question or hypotheses. Could the authors elaborate on why combining structural connectivity and arousal / gene expression data is interesting, and whether similar approaches have been performed in other species?

3) Gradient ordering is interpreted as a key result (“suggesting a potential link between developmental and evolutionary processes”), but switches in gradient ordering between eg first and second gradient could plausibly happen, even in young adult samples (including different HCP subsamples). This has to do with the eigenvalues of the first and second functional gradients often being quite close to one another. In that respect, the 'link' to children/adolescent/adult findings from completely different datasets with different acquisition parameters etc also appears somewhat selective and qualitative without further analyses, including some more exhaustive assessments of within sample switching in gradient order (eg via bootstraps etc).

4) The study utilizes functional connectivity, structural connectivity and gene expression data from

different marmoset samples. Unless there is validation data from the sample primates available, can the authors comment on potential limitations of such an approach, given that inference is based on between-dataset associations. Furthermore, could they provide further details on how the marmosetbrain.org tracer based connectivity matrix was derived? Likewise for the marmoset gene expression atlas. Are sex distributions similar across datasets, and are ages comparable?

5) The HCP dataset contains nested family relationships. Wouldn't it make more sense to study unrelated individuals instead to make data similar in scope to the unrelated marmoset datasets? Also, there are some divergences with respect to the processing of marmoset and human fMRI data. Couldn't this difference also induce differences across species? Please discuss and/or run supporting analyses based on harmonized processing in both species.

6) Further details on the gene expression dataset may be worthwhile. At the moment, this analysis appears underspecified. a) How were receptor expression maps aligned with nissl images? Are there estimates of accuracy of this cross-modal alignment or prior papers. Ditto for the registration between nissl images and fMRI templates - one could imagine that these sources have quite a different scale. Was there any filtering of gene expression information performed, to e.g. explore cross-subject consistency?

7) A common variable contributing to structure-function relationships in the nervous system of humans and animals is spatial proximity and autocorrelation. Can the authors detail in how far sources of autocorrelation were addressed in the current study. For example, it was not clear to me how spatial autocorrelation was accounted for when assessing correspondences and associated significances of correlations between structural and functional gradients (e.g. Figure 2b, p-values appear unadjusted). For further details, see eg Burt et al (<https://doi.org/10.1016/j.neuroimage.2020.117038>) or Markello et al. (<https://pubmed.ncbi.nlm.nih.gov/33857618/>).

REVIEWER COMMENTS

Response to all reviewers:

We really appreciate all reviewers' highly constructive and insightful comments. Per those comments, we have made substantial revision of the original manuscript, which we believe greatly strengthened the study. Below you can find our point-by-point responses to reviewers' comments.

Finally, we'd like to apologize for submitting the revision right before the deadline. In addition to the large amount of work needed for the revision, our work progress was severely impacted by the harsh COVID lockdown here in Shanghai. For more than two months from late March to the end of May, we were confined within our homes with no access to the normal work environment. Despite this difficulty, we hope we addressed reviewers' comments satisfactorily.

Reviewer #1 (Remarks to the Author):

Thank you for inviting me to review this manuscript by Tong and colleagues, in which the authors use gradient-based methods to analyse resting state fMRI data from awake marmoset monkeys and link the resultant low-dimensional descriptions of the data to white-matter and genetic expression maps. While this is an exciting example of the using of multimodal neuroimaging data to test hypotheses regarding structure-function relationships in the brain, many of the hypotheses appeared somewhat exploratory. In addition, I was concerned that the interpretation of the primary results may have been related to methodological choices, which is a factor that I encourage the authors to rule out with subsequent analyses.

Major comments

1. How were the marmosets (1) kept still enough to allow for the collection of BOLD data (2) without substantial motion artifacts?

Response:

(1) We sincerely apologize for not stating the awake marmoset fMRI procedure very clear. The procedure was described in details in another earlier manuscript of ours (<https://www.biorxiv.org/content/10.1101/2021.11.12.468389v2>), which is also currently under revision at Nat. Comm. In P43 Line 6, we stated that “Details of experimental parameters and marmoset information were reported in our previous study”. However, as the reviewer pointed out, it is important to include its description for the continuity of the manuscript. Now we revised the method section (P43 Line 6) to better describe the procedure as follows:

“All marmosets underwent a 3-to-4 week acclimatization protocol as previously described¹. Briefly, in the first week, only body restraining was applied with an increasing period from 15 to 60 min. In the second week, recorded MRI noise was added and habituation periods gradually increased to 120 min. In the third week, head fixation using the customized helmet was added. After the 3-week training period, marmosets were fully acclimated to lay in the sphinx position, with their heads comfortably restrained by 3D-printed helmets.”

(2) Indeed, the awake marmoset data showed occasional head motion during MRI scanning. We carefully evaluated the frame-wise displacement (FD) of our dual-center marmoset dataset (Fig. R1a), and found head motion level was low across ION and NIH dataset (>99.7% volumes less than half voxel size). Also, we found the “12 rp + 10 PCs” regression based motion correction significantly reduced the motion level in our data (Fig. R1b), which was thoroughly evaluated in our previous awake animal fMRI study². Moreover, we compared the marmoset functional gradients between large (FD > 250 μ m) and low (FD < 50 μ m) head motion groups (Fig. R1c). The significant inter-group similarity indicated the overall architecture of functional gradients was not influenced by the subject’s head motion (Fig. R1d). In addition, we evaluated the explained variance of first 4 gradients along with arousal fluctuations in the above two cohorts, and found consistent inverted-U

shaped relationships with the arousal level (Fig. R1e). Therefore, we can safely conclude the head motion had minimal impact on our gradient results after our preprocessing. The above results (Figure R1) have been added as Figure S5 in our revised Supplementary materials.

Figure R1. Head motion had minimal impact on gradient results after preprocessing. (Now Figure S5 in our revised Supplementary materials)

(a) Distribution of frame-wise displacement (FD) across marmoset raw EPI volumes. (b) Effect of marmoset head motion removal after the "12 rp + 10 PCs" regression. Each dot

represented an EPI run. (c-d) Reproducibility (c) and high inter-group similarity (d) of marmoset functional gradients between large (FD>250 μ m) and small (FD<50 μ m) head motion groups. Each dot represented a voxel of marmoset brain. (e) Consistent inverted-U shaped relationships with the arousal level in large and small head motion groups.

2. P6L94: There are perhaps earlier/better references for the following statement: “often attributing to the arousal fluctuation 22-24.”, such as <https://www.ncbi.nlm.nih.gov/pmc/articles/PMC4843437/> and <https://pubmed.ncbi.nlm.nih.gov/27693256/>

Response:

We really appreciate this suggestion and added these two references in our manuscript (Page 5 Line 18).

3. P8L122: The following line requires statistical support: “We systemically characterized the marmoset functional gradients, which resembles the children cortical organization of human”. See below for further explanation of this issue.

Response:

Following Reviewer’s suggestion, we quantitatively compared the first two gradients between marmoset, human children and adults by the fingerprinting method³⁻⁵. As used in previous studies, the fingerprinting method afforded the ability of cross-species comparison and could bridge the gap between marmoset and human neuroanatomy. We included 11 target regions based on a previous cross-species study⁶. The corresponding brain regions included 6 heavily myelinated and 5 lightly myelinated regions, i.e., early somatomotor (1), auditory (2), early visual (3), middle temporal (MT) complex (4), parietal (intraparietal sulcus) visual (5), and retrosplenial cortex (6), prefrontal (A), lateral parietal (B), lateral temporal (C), medial parietal (D), and insular (E) cortex (Fig. R2a).

First, we extracted gradients values of marmosets, human children and human adults (Fig. R2b) in the 11 target regions and obtained the cross-species gradient fingerprints (Fig. R2c left panel). Then, we calculated the Pearson’s correlation coefficients (C.C.) between each pair of gradient fingerprints (Fig. R2c, middle panel). Finally, we clustered the gradient fingerprints using the method of community detection (Fig. R2c, right panel), and found the marmoset functional gradients significantly resembled the children cortical organization of human (Fig. R2d-e).

We have revised our manuscript in Page10 Line 10.

4. P9L163: I'm not sure that it's sufficient to state that the gradients in marmoset "closely resembled" patterns in the human. (1) Is there some way to quantify this? (2) Is it only the average maps for which this is true, or do individual marmoset gradients also share the same relationship? (3) I realize that this comparison (i.e., the mapping between human and marmoset) is non-trivial, however given that this is one of the major claims of the manuscript, and I was expecting to see a more formal statistical comparison between the maps.

Response:

(1 and 3) We fully agree that more formal statistical comparison is needed here. Details were shown in our response to the above "Major comments 3". We have revised and updated the results in our manuscript (Page 10 Line 10).

(2) Thanks a lot for your comment. As suggested, we compared the first two gradient maps across individual EPI sessions (Fig. R3). We found the marmoset functional gradients showed high stability across EPI sessions in both ION and NIH dataset (Fig. R3a). Meanwhile, we calculated the Pearson's correlation coefficients between marmoset and human children functional gradients (Fig. R3b) at the individual EPI level. The scatter plots showed marmoset gradients at the individual EPI level also shared the same relationship with the human functional gradients on cross-species gradient realignments. We have revised our manuscript in Page10 Line 16.

and NIH dataset. FG, functional gradient.

(b) Highly reproducible similarity between averaged human (children and adults) gradients and individual marmoset gradients. Each dot represented an individual marmoset EPI session.

5. How distinct was the % explained variance for gradients 1 and 2 in the marmoset data?

Response:

We hoped we understand your comment correctly.

(a) Might your confusion attribute to the fact that we forgot to indicate the specific values of cross-validated explained variance (cvR^2) in the manuscript? If so, in the present analyses, the primary marmoset functional gradient (FG 1) accounted for ~30 % (ION: 29.00%, NIH: 30.61%) of the observed functional variance, relative to ~15% (ION: 17.13%, NIH: 14.10%) for the second functional gradient (FG2). We have revised our manuscript in Page 9 Line 8 and Page 9 Line 12.

(b) Alternatively, were you concerned about the variability (or stability) of the % explained variance for gradients 1 and 2? If so, we quantitatively compared the cvR^2 of FG1 and FG2 across marmoset individuals (Fig. R4) and found the high stable order of functional gradient in our dual center dataset. We have revised our manuscript in Page10 Line 1.

Figure R4. Stable order of functional gradient across individual marmosets. (Now Figure S4 in our revised Supplementary materials)

The explained variance of functional gradient 1 was consistently higher than that of functional gradient 2 across individual marmosets. Each dot represented an individual marmoset.

6. (1) How sensitive was the ordering of the gradients to different potential sources of noise, such as head motion? I could imagine that both human children and awake marmosets move quite a lot in the scanner, (2) so either head motion, or the statistical approaches instigated in order to mitigate against its potential effects could potentially explain why certain gradients were more obviously present/absent in these two cohorts.

Response:

We really appreciate the Reviewer's constructive comments.

(1) The details of the gradients ordering evaluation were shown in our response to the above **Comment 5 (2)**. The first two functional gradients showed stable order across all

individual marmosets, i.e., the explained variance (R^2) of functional gradient 1 was consistently higher than that of functional gradient 2 (Fig. R4). Therefore, regardless any individual variability (including different levels of head motion), the ordering of the first two FC gradients remained stable.

(2) Indeed, the awake marmoset data inevitably contained some head motion, and the motion levels may be different across marmoset, human children and adults datasets. For the marmoset data, details of motion artefact evaluation have been shown in **Comment 1 (2)**. Briefly, we found the regression-based motion correction was sufficient to minimize the sparse and few motion artifact in our data (Fig. R1). While there was notable decrease of maximum and mean frame-wise displacement after our preprocessing (Fig. R1b), we also compared the functional gradients and the dynamics across large and small head motion groups (Fig. R1b-e). We found high consistency of functional gradient topography and stable arousal related gradient dynamics, indicating minimal impact of head motion on our results after our preprocessing.

7. Figure 2D – the curve fits in Figure 2d appear slightly problematic to me. Most of the data appears to exist at the two ‘extremes’. Is there a better way to fit the data, or better yet, to include an estimate of certainty into the fits that is proportional to the amount of data in each bin?

Response:

Thank you for pointing out this issue in our curve fitting process. We found the marmoset “eye open ratio” (behavior arousal index) roughly exhibited a semi-Beta distribution (Fig. R5a, left panel), and most of data points existed at the two “extremes”. Therefore, we projected the marmoset “eye open ratio” from “semi-Beta distribution” to “Gamma distribution” (Fig. R5a, right panel) by a Fisher r-z function: $\arctanh(2*r-1)$. After this Fisher’s transformation, the distribution is now similar to the distributions of structural-function gradient similarity (Fig. R5b), which is ideal for the fitting purpose.

Next, we evaluated the optimal order of polynomial fitting, for fitting the relationship between “eye open ratio” and “structural-function gradient similarity”. For both raw and Fisher transformed “eye open ratios”, the fitting errors of 2nd-order fitting was significantly lower than that of 1st-order fitting, but similar to that of 3rd fitting (Fig. R5c-d). Therefore, we believe 2nd-order polynomial fitting was the best choice here.

Finally, we used 2nd order polynomial fitting to describe the relationship between the “eye open ratio” and “structural-function gradient similarity” (Fig. R5e-f). In both raw and Fisher transformed “eye open ratios”, similar inverted u-shape relationship was found in both cases, with the transformed ratio being slightly more significant.

Therefore, we conclude that using Fisher r-z transformation for “eye open ratio” indeed improved the fitting, and the inverted U-shape relationship remained unchanged after such transformation. The above results have been now added as Figure S12 in our revised Supplementary materials. Nevertheless, as the inverted U-shape remains constant, and the raw “eye open ratio” is more intuitive, we still keep the original results in the main figure. And also, we found the shading of curve fitting in our previous Figure 2d might be misleading which is now replaced by the Figure R5f in the main figure. We have revised

Figure R5. Consistent inverted U-shape relationship between Fisher transformed eye open ratio and structure-function gradient similarity. (Now Figure S12 in our revised Supplementary materials)

(a) Distribution of marmoset behavior arousal index, i.e., eye open ratio. The raw eye open ratio showed a “semi- Beta” distribution, while the Fisher transformed eye open ratio showed a gamma-like distribution. Notably, scans with fully eye open or closed

were excluded for the Fisher r-z transformation. Each dot represented a scan.

(b) Gamma-like distributions of first four structure-function gradient similarity. Each dot represented a scan.

(c-d) The fitting error among 1st-, 2nd- and 3rd- order polynomial fitting between Fisher transformed (c) or raw (d) eye open ratio and structure-function gradient similarity. **, p<0.01; ***, p<0.005; n.s., no significance.

(e-f) 2nd- order polynomial fitting between Fisher transformed (e) or raw (f) eye open ratio and structure-function gradient similarity. Each dot represented a scan. Red line indicated the fitting curve. Red shade represented the prediction interval.

8. Figure 3C – these analyses are really nice, however it's hard to know precisely what to infer from the data, particularly given the lack of an appropriate null model that affords an estimate of just how large a shift is to be expected from chance.

Response:

We really appreciate the suggestion about the null model. Following Reviewer's suggestion, we shuffled the arousal index by scan and applied same analysis to generate a null model control, i.e., replacing the arousal index of scan A by that of scan B. For each marmoset brain region, we extracted the relative distances (to the origin of gradient space) along with arousal fluctuations (Fig. R6a) and made a 2nd-order polynomial fitting to the relative distances (Fig. R6b). We quantitatively defined the polynomial coefficients **a** as arousal effects coefficients (AEC) of functional gradient dynamics (Fig. R6c). Positive AEC represented a U-shape relationship with arousal level, while negative AEC represented an inverted U-shape relationship. Higher AEC indicated sharper U-shape relationship, corresponding to a higher arousal contribution (Fig. 6c). Most brain regions showed statistically significant shift of arousal relevant gradient flow compared to the null model in both ION and NIH datasets (Fig. R6d-e), including ventral and dorsal somatosensory, primary to higher order visual, auditory, frontal pole, auditory, default model, mACC and premotor networks. We have revised our manuscript in Page14 Line 1.

9. P13L243 - the term “flood and ebb” should be better defined for the reader. There are numerous statistics (such as the distribution of dwell times) that can be used to quantify these concepts, and hence provide robustness to the conclusion.

Response:

We’d like to thank Reviewer for this constructive comment. If we understand it correctly, this comment is related to the previous comment (R1Q8). First, we’d like to clarify that such

“flood and ebb” phenomenon is meant to describe the arousal related dynamics in the gradient space, not in actual time. Thus, the definition of “dwell time” might not be able to be directly applied here. To better quantify such dynamics and evaluate its statistical significance, in our response to the previous comment, we extracted the relative distances (to the origin of gradient space) along with arousal fluctuations and made a 2nd-order polynomial fitting to the relative distances. The polynomial coefficient **a** was defined as arousal effects coefficient (AEC) of functional gradient dynamics, to quantitatively describe the amplitude of such “flood and ebb” effect. Compared to the null model, we found most cortical regions showed statistically significant shift of arousal relevant gradient flow. Now we revised our manuscript for better statistical evaluation of the phenomenon of “flood and ebb” (Page 14 Line 1).

Minor comments

1. Abstract: “In this study, we investigated [the] biological basis of large-scale functional gradients by combining a large awake resting-state fMRI dataset of marmoset monkeys with retrograde tracing and gene expression databases.

Response:

Thank you very much for pointing out the mistake. We have corrected it.

2. Could the authors please expand on the ‘dynamic conditional connectivity’ analysis used in the study?

Response:

We apologize that we did not provide details of “dynamic conditional connectivity” analysis, and now we revised the method section (Page 49 Line 4) to better describe the procedure:

“Dynamic connectivity was computed using the dynamic conditional correlation approach (https://github.com/canlab/Lindquist_Dynamic_Correlation), a multivariate volatility method⁷ (details in Lindquist, et. al., Neuroimage, 2014). Briefly, the dynamic conditional correlation model was used to deal with the temporal autocorrelation and non-stationarity in fMRI time-series. This model assumes that the brain time courses follow a multivariate Gaussian distribution, and that the conditional mean, variance and co-variances change in an autoregressive form. Unlike sliding-window approaches that estimate connectivity over a fixed window length, this is a model-based method that estimates the contribution of surrounding time points to the covariance matrix. Pairwise dynamic connectivity values were obtained for every time point of each resting-state run. This resulted in a matrix of connectivity values that was M (time points) × N (connections) for each run.”

3. How did the authors account for blinks in their pupillary analysis?

Response:

We sincerely apologize that this point was not made clear in the manuscript. The time series of marmoset pupil size were downsampled from 60 fps to 0.5 fps (corresponding to EPI TR of 2s), which were estimated by the median value within the time bin corresponding to each fMRI volume. Thus, the fast blinks (less than 0.5 second) were most likely excluded in this process and very unlikely to affect our further analysis. We have revised our method section in our manuscript (Page 48 Line 3).

Reviewer #2 (Remarks to the Author):

The study investigated the principal gradients of brain connectivity using multimodal data from marmosets with focusing on anatomical and neuromodulatory contributions. Briefly, the principal gradients were first computed for both functional connectivity measured by resting-state fMRI correlations and structural connectivity assessed by retrograde tracers, and they showed similar spatial topography. The authors also showed that the principal gradients of resting-state fMRI connectivity (i.e., FC gradients) are modulated by arousal levels of the animals, and thus developed a modeling frame to differentiate the contributions from the structural connectivity (SC) and the neuromodulatory similarity (NS), which was assessed by inter-regional correlations of neuromodulatory receptor expression. While the SC-predicted FC gradients showed a similar dependency on the arousal level as the FC gradients, the NS-predicted FC gradient had opposite modulations. Lastly, the paper showed that the spatial distributions of neuromodulatory receptors are related to the topology of the FC gradients. Overall, the study provides an interesting perspective on the potential link between the neuromodulatory system and the FC gradients, with the use of a variety of functional, structural, and genetic data. The part dissecting the contribution of the anatomical and neuromodulatory contributions to the FC gradients is particularly novel. However, I do find some potential technical issues, including in this critical part. My specific comments/questions are shown as below

1. My biggest concern is about the modeling frame, particularly the reduced model, used for the results presented in Figures 4 and 5, which are the major innovation of this study. According to the Methods section, this reduced model is very similar to the single variable model. The only difference is a better control of the degree of freedom with shuffling the other variable. By comparing with the full model, one can estimate the unique contribution of the variable on the response variable (here should be FC) based on the change of explained variance, just like in the ref #60 cited by the paper. But I don't believe it was used properly in this paper. Both Figs. 4d and 4f compared the gradients of predicted FC from these two variables (i.e., the SC and NS), and Fig. 4e is just an illustration without any quantifications about how much SC and NS contribute to the predicted FC and thus the resulting FC gradients. (1) For this part of the analysis, the authors should show how much of the variance of the response variable, i.e., the FC, is explained by two independent variables (i.e., the SC and NS). Or, if the principal gradient of the FC is of the interests, the authors need to show the similarity between the empirical gradient and predicted gradient derived from the single variable model or the reduced model, as what they did for the full model in Figure 4C. (2) I also encourage the authors to check the results shown in Figs. 4d and 4f. As discussed above, the reduced model should produce very similar results as the single variable model by adding a randomly shuffled variable. This appears to be true based on the visual inspection on the gradients. They look indeed very similar to each other. However, it is surprising that the spatial similarity of the gradients from the predicted data was changed dramatically.

Response:

(1) We sincerely apologize that such important information was missing in Fig. 4e. Now the revised Fig. 4e showed the variances partitioning on a full model prediction of empirical gradients, in which the unique contribution of SC and NS was 41.07% and 26.35%, respectively, while the overlap of SC and NS effects was 32.58% (Fig. R7a).

And also, now we added a supplementary figure (Fig. R7f, now added as Figure S22) to better explain the full and reduced model.

(2) Thanks a lot for the reviewer's insightful comment. After close inspection, we indeed found that our original results in Fig. 4d and 4f showed very similar profile, potentially leading to the misunderstanding of the modeling results. We found this phenomenon was mainly due to the narrow range of the color bar. After broadening the range of color bars in revised Fig. 4d and 4f, the predicted functional gradients from single variable model and reduced model appears more dissimilar upon visual inspection (Fig. R7b-e).

Nevertheless, it is somewhat expected that the spatial similarity of the gradients from the predicted data changed dramatically from the single variable model to the reduced model. As shown in Fig. R7a, there is large overlap between the SC (Fig. R7a, green circle) and NS (purple circle), resulting the spatial similarity on the single variable model prediction (Fig. R7b). Thus, we utilized the reduced model to capture the unique contribution of each variable. In the reduced model, the variable of interest was randomly shuffled 1000 times, and the resulting loss of explained variance (compared to the full model) was the unique contribution of this particular variable in the reduced model (Fig. R7f). Therefore, for a particular variable, the reduced model and the single variable model is expected to be different.

The dramatic decrease of spatial correlation on the reduced model attributed to the separation of unique contributions of SC (Fig. R7a, pure green part) and NS (pure purple part). In other words, the non-overlapping parts of SC and NS effects on reduced model resulted the non-significant spatial similarity on predicted gradients (Fig. R7c). We have revised our manuscript and added more details about the reduced model (Line 17 Line 5).

2. The principal gradient of the neuromodulatory similarity should be presented and compared with the FC gradient, similar to what they've done for the structural connectivity (Fig. 2). The gradient maps of the neuromodulatory similarity shown in Figures 4d and 4f do not appear to be similar to the FC gradients to me.

Response:

As suggested, we conducted similar analyses on the neuromodulatory similarity, and found the neuromodulatory similarity gradients were also correlated with the functional gradients (Fig. R8), but to a less extent compared to the structural gradients. We have added the neuromodulatory gradients in our manuscript (Page 16 Line 6).

Figure R8. Gradients of neuromodulatory similarity in marmoset. (Now Figure S21 in our revised Supplementary materials)

(a) First four gradients of marmoset neuromodulatory similarity based on marmoset gene atlas. Areal borders were based on the Riken Brain/MINDS cortical parcellation. NG, neuromodulatory gradient.

(b) Topographical similarity between neuromodulatory and functional gradients. High similarity between neuromodulatory and empirical functional gradients (upper panel). Significant difference between the empirical similarity of neuromodulatory-functional gradients and null distributions (lower panel), in which null distributions was derived from

the Pearson's correlation coefficients (gray shades) between neuromodulatory gradients and the spatial autocorrelation preserving surrogate maps of functional gradients. Each dot represented a brain region. Red and blue lines represented the Pearson's correlation between empirical neuromodulatory and functional gradients.

3. The gradient dynamics shown in Figure 5 is somewhat expected, especially if the neuromodulatory similarity doesn't produce similar gradients as the FC. The predicted FC (fitted FC) is the projection of empirical FC onto a subspace spanned by the independent variables, i.e., the SC or NS, and thus contained the information of real data. As shown in Figure 2, the SC gradients are highly similar to the FC gradients. It is thus not surprising to see a similar invert-U shape of modulation of SC-predicted gradient on the arousal index. At the same time, one would expect an opposing U-shape modulation of the gradients for any other variables that are not similar to the FC gradients, which could be the case for the NS-predicted FC gradients. In other words, the U-shape modulation should not be specific to the NS-predicted FC gradients but to anything not having a tight link with the FC gradients, as the "others" shown in Figure 5h.

Response:

We hope we understand your comment correctly. First, as shown in our response to the previous comment (Fig. R8), the neuromodulatory similarity gradients are also correlated to the FC gradients, but such correlation is lower than that of structural connectivity. This result is also expected, as the two variables (NS and SC) showed large overlap (Figure R8a) in our model. Not surprisingly, NS and SC showed correlation (Fig. R9a).

Next, we do not feel it can be entirely predicted that SC-FC gradient similarity alone could guarantee an "invert-U shape of modulation of SC-predicted gradient on the arousal index". The SC related modulation on the FC gradient dynamics could be any shape, or even no modulation at all, given there's "other" variables in the model. Indeed in our results, we observed a U shape of unique modulation of NS-predicted gradient on the arousal index, opposite to that of SC-predicted gradient (invert-U shape).

To empirically evaluate random or non-NS related variables' contribution, we added a noise term (Fig. R9b) or an expression similarity matrix based on randomly selected genes ((Fig. R9c) (gene list can be found in Supplementary File 1) in our model. The results showed that the random noise component did not exhibit significant contribution across arousal fluctuations (Fig. R9d-e). And the random expression similarity matrix, exhibiting very low degree of correlation with empirical FC (Fig. R9f), showed very minor contribution compared to the NS (Fig. R9g). Therefore, the above new analysis suggests that the NS and SC contribution to the arousal related gradient dynamics is likely to be specific.

However, we fully agree that our current model is in no way exhaustive, in terms of including all potential predictors in the model. We hope we can reveal additional contributing factors in the future, with more marmoset brain resources available. Now, we have revised our manuscript and added such control analysis in Page 19 Line 13.

Figure R9. Noise component and random ISH expression similarity did not drive inverted U-shape relationship on unique gradient dynamics with arousal fluctuation. (Now Figure S24 in our revised Supplementary materials)

(a) Significant correlation between structural connectivity (SC) and neuromodulatory similarity (NS), suggesting the overlap between SC and NS.

(b-c) Control analysis of the U-shape modulation for the neuromodulatory similarity (NS). The control analysis was conducted by adding a noise matrix (NM) as another variable (b) or replacing the NS with random ISH expression similarity (c, RS).

(d) Example of a random noise component which did not have a tight link with the empirical functional connectivity.

(e) No significant arousal relevant unique contribution from the noise component. Across arousal levels, the unique contribution (ΔR^2) was not significantly different from zero (one-sample t-test, two tails).

(f) Example of the random ISH genetic similarity which showed a significant correlation with the empirical functional connectivity.

(g) Inverted U-shape relationship of arousal relevant unique contribution from the random ISH genetic gradient 1, with no significant arousal relevance on the gradient 2.

4. There is a very relevant paper the authors may want to include since the major topic of this paper is about how the FC gradients are related to arousal and the neuromodulatory system. It has been shown recently (Gu, et. al. Cerebral Cortex 2021) that the FC gradient is related to the infra-slow global waves of resting-state fMRI signals that propagate along the FC gradient directions, and these propagations are highly sensitive to the brain arousal state. These findings may help understand how the FC gradients form and why they are sensitive to arousal fluctuation. Thus, it would be appropriate to include some discussion about this.

Response:

We'd like to really thank Reviewer for pointing us to this very relevant study. Now we added the following paragraph in the discussion (Page 27 Line 14):

“A recent study⁸ revealed that the infra-slow global waves of resting-state fMRI signals propagate along the primary gradient in humans, and these propagations are highly sensitive to the brain arousal state. Such propagation of the infra-slow waves may be related to both anatomical connectivity among cortical hierarchy and ascending neuromodulatory system, providing more plausibility that arousal may translate the static anatomical wiring into dynamic functional configurations via the ascending neuromodulatory system.”

5. I also have some questions regarding the last part of the results related to the receptor maps. First, the spatial correspondence between the receptor maps and the FC gradients are very weak with a R-square of 1~2% (Figs. 6 and S12). The authors showed some of these spatial correlations are statistically significant. But there were two potential issues. (1) These brain maps are spatially continuous, which means that the parcels/voxels are not independent and thus the real degrees of freedom should be much smaller than the number of parcels/voxels. Therefore, simply shuffling the parcels/voxels for the receptor maps is not a correct way of building the null distribution. The reported results were not corrected for multiple comparison. (2) Second, if the paper meant to emphasize the role of the neuromodulatory system in generating the FC gradients, the proper controls need to be included. The authors need to show that the neuromodulatory receptors show a stronger spatial correspondence with the FC gradients than other non-neuromodulatory receptors. (3) Also, the spatial similarity itself is not a strong indicator of a tight relationship. For example, it has been shown that the primary gradient of gene expression in the brain is similar to the FC gradient in human (the FC gradient #2 in this paper) (Burt, JB et. al., Nature Neuroscience, 2018). One would thus expect that a significant proportion of genes have significant spatial correlation with the FC gradient. However, it doesn't mean that all of these genes would play a role in generating this FC gradient.

Response:

(1) We really appreciate Reviewer's constructive comment. We fully acknowledge that our original result did not treat the spatial continuity and null models very well. Because the neuromodulatory receptors are spatially auto-correlated, we adopted a procedure from previous studies⁹⁻¹¹ (similar to Fig.3C in Demirtas et al., ref #9) to overcome this issue (Fig. R10a).

First, we generated surrogate maps that randomly vary in their particular topographies (n

= 1000 times shuffling) but preserve the general spatial autocorrelation (SA) structure of corresponding receptors (Fig. R10a). As expected, the SA preserving surrogate maps showed a much wider distribution of correlation with the empirical one, compared to the SA independent surrogate maps (Fig. R10b). Using those SA preserving surrogate maps, we found similar results compared with our original result from SA independent surrogate maps (Fig. R10c, CHRM3 as an example). Moreover, to evaluate whether the arousal modulation from each neuromodulatory receptor was overestimated, we calculated the Pearson's C.C. (right tail) between the surrogate similarity (to the empirical one, $n = 1000$) and the unique contribution on functional gradients of a particular receptor (Fig. R10d, like Fig.3E in Demirtas et al., ref #9). The rationale is: if the correlation is not significant, it means random receptor maps could contribute similar arousal modulation, i.e., the empirical receptor does not specifically contribute to arousal dynamics. Alternatively, if the correlation is significant, it means larger spatial map shuffling causes larger loss of the unique contribution for the corresponding receptor, i.e., the empirical receptor does contribute to arousal dynamics. Our result showed CHRM3 receptor contributed significantly to gradient 1 arousal dynamics, but not significantly to gradient 2 (Fig. R10d). Finally, we applied the false discovery rate (FDR) correction on all receptors and found several receptors showing statistically significant contributions on the gradient dynamics with arousal fluctuations (Fig. R10e).

The above result is an example using the CHRM3 expression profile, and the procedure is summarized in Fig. R10e and applied to all neuromodulatory receptors in Fig. S25. With the above procedure controlling the SA issue, our significance level (dashed lines in Fig. 6b) became more stringent, but still several neuromodulatory receptors exhibited statistical significance in our revised Fig. 6b. We have revised our method section (Page 54 Line 9) and added above results (Page 21 Line 1) in manuscript.

(2) We really appreciate this constructive comment. Per Reviewer's suggestion, we now added two new expression similarity matrices, one based on glutamate receptors and another one based on randomly selected genes (Fig. R11). Those glutamate receptor and random genes are listed in Supplementary File 1. Both of them showed significantly reduced contributions to arousal related dynamics (Fig. R11 c, g, e). Therefore, we can conclude that neuromodulatory receptor expression does specifically contribute to the arousal related dynamics.

(3) We fully agree that the overall gene expression may be related to the FC at the gradient level. However, such static relationship is not the main focus of the current study. Instead we tried to show 1) FC gradient strength is dynamic with arousal fluctuations, and 2) such FC gradient dynamics is related to the neuromodulatory receptors. The above analysis in Fig. R11 demonstrated that such contribution is likely to be more pronounced to neuromodulatory receptors, compared to the non-modulatory receptor of glutamate or randomly selected genes.

Now, we have revised our manuscript accordingly and added such control analysis in Page 19 Line 13.

(e) Revised criterion for identifying the receptors with significant contribution to the arousal relevant gradient dynamics.

Figure R11. Control analysis of the unique U-shape contribution from the neuromodulatory similarity. (Now Figure S24 in our revised Supplementary materials)

(a-b) The control analysis was conducted by replacing the NS with glutamate receptor similarity (a, GluS) or random ISH expression similarity (b, RS).

(c) Significant difference (pair-wise t-test) of cross-validated explained variance (cvR²) from the general linear model (GLM) across the original, control A and control B groups.

(d) Glutamate receptor similarity showed a significant correlation with the empirical functional connectivity.

(e) U-shape relationship of arousal relevant unique contribution (ΔR²) from the random ISH genetic gradient 1 & 2.

(f) Example of the random ISH genetic similarity which showed a significant correlation with empirical functional connectivity.

(g) Inverted U-shape relationship of arousal relevant unique contribution from the random ISH genetic gradient 1, with no arousal relevance on the gradient 2.

Reviewer #3 (Remarks to the Author):

The manuscript addresses dynamic changes in functional organization of the primate cortex, using a low dimensional account of resting state functional connectivity (“gradients”). Particularly interesting is the focus paid to the role of arousal on functional organization. Additionally, the study attempts to calculate the contribution of structural connectivity and similarity in receptor expression to the spatial pattern of the principle functional axes.

1. One major issue is that the methods are incomplete, such as the analyses in Figure 6, and the methods lack key details that would be necessary to replicate the study. For example, how are receptor expression maps “mapped” to the Nissl stained images, and how were Nissl stained images “registered” to the study-specific MRI template?

Response:

We sincerely apologized that we did not describe the methods in more details. As mentioned, we revised our method (Page 51 Line 3) in our manuscript (Fig.R12), as follow:

“The neuromodulatory receptor information was obtained from the marmoset gene atlas database (<https://gene-atlas.brainminds.riken.jp/>). Registration of marmoset ISH images to MRI space was summarized in Fig. S20. Briefly, we downloaded the Nissl stained coronal images and neuromodulatory receptor related gene expression maps. Next, we mapped the receptor expression maps to the Nissl stained images, using “rigid-body transformation” for the coarse whole brain registration and “large deformation diffeomorphic metric mapping (LDDMM)^{12,13}” for more subtle slice-by-slice registration. To facilitate the comparison with fMRI results, the Nissl stained images were registered to the study-specific MRI template by affine nonlinear transformation (“oldnormalize” of SPM12), and the affine transformation matrix was then applied to the receptor expression map to bring it to the MRI space. Next, receptor expression data were parcellated into 116 cortical regions of interest, based on the Riken Brain/MINDS cortical parcellation¹⁴. Finally, the resulting neuromodulatory receptor similarity was calculated using the correlation of gene expression level across each pair of regions.”

Figure R12. Registration of marmoset ISH images to MRI space. (Now Figure S20 in our revised Supplementary materials)

(a) Computational strategy of marmoset ISH registration and generation of neuromodulatory receptor similarity. The neuromodulatory receptor gene expression data were obtained from the marmoset gene atlas database (<https://gene-atlas.brainminds.riken.jp/>). Briefly, we downloaded neuromodulatory receptor related gene expression maps and the corresponding Nissl stained coronal images. Next, we registered the expression maps to the Nissl stained images, yielding a 3D spatial alignment between anatomical and gene expression images (details in Fig. b). The Nissl stained images were registered to the study-specific MRI template (details in Fig. c), and this transformation was applied to the gene expression maps to bring them to the MRI space. Gene expression data were then parcellated into 116 cortical regions of interest, based on the Riken Brain/MINDS cortical parcellation. Finally, the resulting neuromodulatory receptor similarity was calculated using the correlation of normalized

gene expression levels across each pair of regions.

(b) Registration from receptor gene expression maps to Nissl images. Firstly, these ISH images were down-sampled (5 times) in gray scale to reduce the computation load. Then, these Nissl and receptor expression images were iteratively aligned to a weighted average of its neighbors, respectively. The receptor expression images were de-noised using the standard medial filter and further threshold by the median values of the ISH images, termed as the “energy map” (bottom image in b). Finally, the 3D receptor expression maps were registered to the Nissl images using rigid-body transformation, and the transformation matrix was applied to the receptor energy maps. Further adjustment of registration was made slice-by-slice using the method of large deformation diffeomorphic metric mapping (LDDMM).

(c) Cross-modal registration from Nissl images to MRI template. The 3D Nissl images were nonlinearly transformed to the MRI template using the “oldnormalize” of the SPM12. The affine transformation was then applied to the receptor expression energy map.

(d) Example of registered and normalized CHR3 receptor expression energy map. The expression energy was overlaid on the MRI template in coronal slice view (upper panel) and 3D surface view (lower panel).

2. The conclusions often overstate the findings or rely upon qualitative, subjective comparisons. (1) For example, comparison of gradients with functional networks and comparison of gradients across species are based on visual inspection, but should be empirically evaluated. (2) Another example is that the authors suggest that Fig 2d is an inverted U-shape, but it appears relatively flat with high variance. Again, this should be tested statistically, for example by evaluating the fit of a quadratic model to the data. (3) In terms of overstatement, the variance explained by individual receptors is less than 1% (shown in Supplementary Figure), yet these effect sizes are not mentioned in the main text and their negligible contribution is further dissected. It would be important for the authors to integrate statistics (e.g. p values, R2, t-statistic, dof) into the main Results to ameliorate this issue.

Response:

We really appreciate the Reviewer’s constructive comments.

(a) Per reviewer’s suggestion, we quantitatively compared the first two gradients across marmoset, human children and human adults by the fingerprinting method³⁻⁵. As used in previous studies, the fingerprinting method afforded the ability of cross-species comparison and could bridge the gap between marmoset and human neuroanatomy. We included 11 target regions based on a previous cross-species study⁶. The corresponding brain regions included 6 heavily myelinated and 5 lightly myelinated regions, i.e., early somatomotor (1), auditory (2), early visual (3), middle temporal (MT) complex (4), parietal (intraparietal sulcus) visual (5), and retrosplenial cortex (6), prefrontal (A), lateral parietal (B), lateral temporal (C), medial parietal (D), and insular (E) cortex (Fig. R13a). First, we extracted gradients values of marmoset, human children and human adults in the 11 target regions and obtained the cross-species gradient fingerprints (Fig. R13b upper panel). Then, we calculated the Pearson’s correlation coefficients (C.C.) between each pair of gradient

fingerprints (Fig. R13b, lower left part). Finally, we clustered the gradient fingerprints using the method of community detection (Fig. R13b, lower right part), and found the marmoset functional gradients significantly resembled the ones of human children (Fig. R13c-d). We have revised our manuscript and added the above statistical cross-species comparison in Page 10 Line 10.

(2) Thanks a lot for pointing out this issue in our curve fitting process. We found the shading of curve fitting in our previous Figure 2d might be misleading which is now replaced by the Figure R13e in the main figure. Briefly, we used 2nd- order polynomial fitting to describe the relationship between the “eye open ratio” and “structural-function gradient similarity” (Fig. R13e). We found weak but significant 2nd- order polynomial relationship between “eye open ratio” and structural-functional gradient similarity. The inverted U-shape relationship remained unchanged and was evaluated statistically (revised Fig. 2d) per Reviewer’s suggestion. Also, we have revised our manuscript in Page 12 Line 11.

(3) We sincerely apologized for missing such important details. First of all, different neuromodulatory receptors are functionally correlated and their expression patterns are often correlated, resulting potentially large overlapping contributions from these receptors. Using the reduced model, it is somewhat expected that the unique contribution for a given specific receptor would be small.

To evaluate such receptor specific contributions to arousal dynamics in a more statistically proper way, we have now significantly revised our procedure. As these brain maps are spatially continuous, rendering the parcels/voxels not independent, thus the actual degrees of freedom should be much smaller than the number of parcels/voxels. It means simple shuffling of parcels/voxels for the receptor maps in our original manuscript is not an ideal way of building null distributions and resulted the lack of multiple comparison. Thus, we adopted a procedure from previous studies⁹⁻¹¹ (like ref #9 Fig.3C) to overcome this issue. First, we generated surrogate maps that randomly vary in their particular topographies ($n = 1000$ times shuffling) but preserve the general spatial autocorrelation (SA) structure of corresponding receptors. As expected, the SA preserving surrogate maps showed a much wider distribution of correlation with the empirical one, compared to the SA independent surrogate maps (Fig. R13f). Using those SA preserving surrogate maps, we found similar results compared with our original result from SA independent surrogate maps (Fig. R13g, CHRM3 as an example). Moreover, to evaluate whether the arousal modulation from each neuromodulatory receptor was overestimated, we calculated the Pearson’s C.C. (right tail) between the surrogate similarity (to the empirical one, $n = 1000$) and the unique contribution on functional gradients of a particular receptor (Fig. R13h, like Fig.3E in Demirtas et al., ref #9). The rationale is: if the correlation is not significant, it means random receptor maps could contribute similar arousal modulation, i.e., the empirical receptor does not specifically contribute to arousal dynamics. Alternatively, if the correlation is significant, it means larger spatial map shuffling causes larger loss of the unique contribution for the corresponding receptor, i.e., the empirical receptor does contribute to arousal dynamics. Our result showed CHRM3 receptor contributed significantly to gradient 1 arousal dynamics, but not significantly to gradient 2 (Fig. R13h). Finally, we applied the false discovery rate (FDR) correction on all receptors and found several receptors showing statistically significant contributions on the gradient dynamics with arousal fluctuations (Fig.

R13i).

The above result is an example using the CHRM3 expression profile, and the procedure is summarized in Fig. R13i and applied to all neuromodulatory receptors in Fig. S25. With the above procedure controlling the SA issue, our significance level (dashed lines in Fig. 6b) became more stringent, but still several neuromodulatory receptors exhibited statistical significance in our revised Fig. 6b. We have revised our method section (Page 54 Line 9) in manuscript and added above results (Page 21 Line 1) after multiple comparison

Figure R13. More statistical evaluation of our original results.

(a) Eleven cross-species landmark regions for gradient fingerprint analysis. The figure was adapted from Van Essen, et. al., PNAS, 2019. (doi:10.1073/pnas.1902299116)

(b) Marmoset functional gradient closely resembled that of human children. Based on gradient values from 11 cortical landmark regions, we extracted the cross-species

gradient fingerprints (upper) and calculated the Pearson's correlation coefficients (C.C.) between each pair of gradient fingerprints. Throughout the method of community detection, we clustered 2 communities of cross-species gradients, suggesting the marmoset functional gradients closely resembled the children cortical organization of human. *, $p < 0.05$ (Pearson's correlation coefficient)

(c-d) Marmoset FG 1 significant correlated with the human children FG1 and human adults FG2, and Marmoset FG 2 significant correlated with the human children FG2 and human adults FG in both ION (c) and NIH (d) datasets. ****, $p < 0.0001$.

(Now Figure S7 in our revised Supplementary materials)

(e) 2nd- order polynomial fitting between eye open ratio and structure-function gradient similarity. Each dot represented a scan. Red line indicated the fitting curve. Red shade represented the prediction interval.

(Now Figure 2d in our revised manuscript)

(f) Histogram of spatial correlation between surrogate maps and empirical one. SA preserving surrogate maps showed wider distribution of correlation with the empirical one, compared to the SA independent surrogate maps, e.g., CHRM3 receptor.

(g) Arousal relevant dynamics of the unique contribution for CHRM3 receptor. Similar inverted U shape relationship using SA preserving surrogate maps as shuffling, compared with SA independent surrogate maps. Errorbar, Mean \pm SEM, $n=1000$ times shuffling.

(h) Pearson's correlation coefficients (right tail) between surrogate correlation and the variance of the unique contribution of CHRM3 receptor. If the correlation is not significant, it means random receptor maps could contribute similar arousal modulation, i.e., the empirical receptor does not specifically contribute to arousal dynamics. Alternatively, if the correlation is significant, it means larger spatial map shuffling causes larger loss of the unique contribution for the corresponding receptor, i.e., the empirical receptor does contribute to arousal dynamics. Each dot represented a random SA preserving shuffling of corresponding receptor ($n = 1000$ times).

(i) Revised criterion for identifying the receptors with significant contribution to the arousal relevant gradient dynamics.

3. The terms functional gradient, core axis, functional hierarchy and intrinsic coordinate system are mixed throughout the manuscript without clear definitions. I would suggest that the authors clarify the terminology and are more cautious in their interpretations. For example, the functional gradients are one possible intrinsic coordinate system [see also neurodevelopmental (Nieuwenhuys et al.,) or phylogenetic (Goulas et al.,) perspectives]. Defining the principle functional gradient as “the core axis of human intrinsic coordinate system” appears to dismiss that there is still contention regarding how the cortex is organized.

Response:

We really appreciate the Reviewer's for point the difference among these concepts. Now, we have removed all the descriptions about “the core axis of human intrinsic coordinate system” and “intrinsic coordinate system” to avoid the misinterpretation of our results.

4. It is interesting to inspect the arousal-related variations in the functional gradients (e.g. Figure 3f). (1) It would be worthwhile comparing results to Cross et al., (2021) that looked at sleep deprivation. (2) As the results are represented in a low-dimensional space, it is difficult to interpret the cause of the “ebb and flow” effect. One possible explanation is that BOLD timeseries are more heterogeneous at mid-arousal, compared to drowsy and high arousal. But it could also be due to the alignment procedure, which may be biased towards the mid-arousal state, because the study-specific template is an average across different states of arousal. The authors could greatly improve the interpretability of the “ebb and flow” (which is the most interesting insight of the paper in my opinion) by further deconstructing the root cause with secondary analyses of the timeseries and the alignment procedure.

Response:

(1) We’d like to thank Reviewer for pointing to this relevant study. While we already included this paper in our original manuscript (ref #73 in our original manuscript), we now added further discussion as follows: (P25 Line 17):

“However, one recent human task fMRI study¹⁵ suggested that among well rested, sleep deprived and sleep recovered states, there was very minor changes of functional gradients, suggesting low contribution of arousal modulation on the functional gradients. This might be related to the difference of fMRI paradigms (tasked based v.s. resting state) and the resulting difference between task regressed gradients and resting-state gradients.”

(2) We really appreciated the Reviewer’s suggestion on the further secondary analyses on the “ebb and flow” phenomenon of gradient dynamics. As suggested, we found that the BOLD functional connectivity (FC) were more heterogeneous at mid-arousal, compared to drowsy and high arousal (Fig. R14a-b) as evaluated using entropy of FC. The entropy of FC was significant correlated with the mean absolute strength of functional gradients, which was the average of the absolute gradient values across cortical voxels (Fig. R14c). Furthermore, we found significant correlations between the entropy of FC and the relative distance in the gradient space (Fig. R14d-e). These results suggested that the heterogeneity of FC is likely related to the “ebb and flow” effect of gradient dynamics with arousal fluctuations. We have added above results in our manuscript (Page 14 Line 8).

Figure R14. Gradient dynamics is related to the heterogeneity of functional connectivity with arousal fluctuations. (Now Figure S16 in our revised Supplementary materials)

(a) Inverted U-shape relationship between arousal index and the entropy of dynamic functional connectivity (FC). Higher entropy indicated more heterogeneous functional connectivity.

(b) Stability of the inverted U-shape relationship between arousal index and the entropy of dynamic FC across individual marmosets.

(c) Significant correlation between the mean entropy of dynamic FC and the mean whole brain absolute strength of functional gradient 1 (upper panel) and 2 (lower panel) of individual marmosets across arousal levels (10 bins).

(d) Illustration of the relative distance of a particular brain region (v.s. the origin) in gradient space.

(e) Entropy of dynamic FC contributed to the “ebb and flow” effect in gradient space. Significant correlation between the mean entropy of dynamic FC and the mean whole brain relative distance in the gradient space (v.s. the origin) across individual marmosets. Each dot represent an individual marmoset across arousal levels (10 bins).

5. (1) How the predicted functional connectivity is generated is not clear from the Methods, however, (2) I would advise to split the data into train and test subsets, so as to evaluate the predictive performance out of sample. This would also benefit the comparison of models.

Response:

(1) We sincerely apologized for make the functional connectivity prediction procedure not clear enough. Now we revised the method section and added more information as following (Page 52 Line 5):

“A general linear model was used to predict the functional connectivity. The predictors were modified structural connectivity and the neuromodulatory receptor similarity matrix. The model was then constructed as

$$FC = b_0 + b_1 \times SC + b_2 \times NS$$

where the output variable FC was the set of whole brain functional connectivity (116 x 116 regions), and the input variables were modified structural connectivity (SC) and neuromodulatory receptor similarity (NS). The regression coefficient b_0 , b_1 and b_2 were then solved by ordinary least squares techniques with Euclid norm constraint. The resulting best fit of empirical FC was termed as the predicted FC, i.e., the linear combination of the regression coefficients and corresponding input variables (SC and NS).”

(2) We sincerely apologize for missing such important information in our original manuscript. In our analysis, the explained variance (cvR^2) between empirical and predicted functional connectivity (Fig.R15a) was obtained using “leave-one-out” cross validation. The full general linear model (GLM) generated the best fit with $cvR^2 = 0.18 \pm 0.002$ (mean \pm sem, $n = 623$ EPI runs) across EPI runs (Fig.R15b). To compute the unique contribution of a particular variable (structural connectivity (SC) or neuromodulatory similarity (NS)), we created reduced models (Fig.R15c) in which only the specified variable was shuffled. The full and reduced models are illustrated in Fig. R15e. To better compare different GLM models (single variable and reduced model), we repeated the same cross-validation analysis on the prediction of functional connectivity with different model inputs, i.e., shuffling SC or NS. We found SC provided more predictive power than NS in both single variable (Fig.R15d, $cvR^2_{SC} = 0.13 \pm 0.001$ v.s. $cvR^2_{NS} = 0.11 \pm 0.002$) and reduced models (Fig.R15d, $\Delta R^2_{SC} = 0.06 \pm 0.0005$ v.s. $\Delta R^2_{NS} = 0.05 \pm 0.001$). We have revised our method section (Page 52 Line 20) and results section (Page 17 Line 7) in our manuscript.

Figure R15. Quantitative evaluation of single variable model and reduced model.

(Now Figure 4 and Figure S22 in our revised manuscript)

(a) Significant correlation between predicted and empirical functional connectivity. (Now Figure 4b in our revised manuscript)

(b) Cross-validated explained variance (cvR²) across all EPI runs (n= 623). Each dot represented an EPI run. The box showed the first and third quartiles; inner line was the median over EPI runs; whiskers represented minimum and maximum values (outliers removed). (Now Figure 4c in our revised manuscript)

(c) Reduced model for investigating the unique distribution of each variable (non-overlapping part). The structure connectivity or neuromodulatory receptor similarity (circle) may have overlapped information with the other one, thus the reduced model (non-overlapping part) provides the unique contribution of each predictor.

(d) Top: cross-validated explained variance (cvR²) maps for different single-variable models. Bottom: unique contribution (ΔR^2) maps for the same variables. The box showed the first and third quartiles; inner line was the median over EPI runs; whiskers represented minimum and maximum values (outliers removed). (Now Figure 4f in our revised manuscript)

(e) Illustration of the full general linear model and the reduced model. (Now Figure S22 in our revised Supplementary materials)

Reviewer #4 (Remarks to the Author):

Tong et al. Anatomical and neuromodulatory basis of large-scale functional topography

The authors combined resting-state fmri with retrograde tracing and gene expression data. They observed (i) different gradient order in monkeys, resembling findings in infants, (ii) inverted u type associations to structural connectivity, and (iii) u-shape associations to arousal level.

1. In the introduction, the authors highlight structure function correspondence at the level of structural and functional features but I believe that more specificity may be warranted when referring to cortical thickness and myelin gradients. In particular, there seems to also be data emphasizing that structural gradients may not exactly follow the spatial pattern as the functional gradients, despite some overall correspondence.

Response:

We really appreciated the Reviewer's insightful comment. We fully acknowledge that the functional gradients do not exactly follow structural gradients, otherwise there would be no need to investigate the potential arousal contribution to functional gradients in our manuscript. Indeed human cortical thickness and myelin gradients have been examined in previous studies^{16,17}, showing relationship with functional gradients. We also conducted such analysis using human data (Fig. R16a,b). However, it was somewhat surprising that such significant correlations between cortical thickness (and myelin maps) and functional gradients in humans did not hold in marmosets (Fig. R16c,d). Such cross-species difference is of interest itself, but is out of the scope of our current study. The marmoset cortical thickness map and myelin content were obtained from a previous study¹⁸ (<https://marmosetbrainmapping.org/>). In contrast, structural gradients and receptor similarity gradients showed significant correspondences with functional ones (Fig. R16d). The rich information in cortical thickness and myelin still warrants future detailed studies, but based on the above result, in our current study we feel it is justified to include structural and receptor similarity gradients in our main results.

Nevertheless, many macroscopic characteristics of the brain, such as structural connectivity, cortical thickness, myelin content and gene expression levels, are ultimately determined by the genome to a large extent, and thus it would not be surprising to observe some correlation among them. Of course the gap between the genome and large-scale functional gradients is so huge, that in the current study we only aim to examine their "next-level" correlates.

We have added above results in our manuscript (Page 11 Line 17).

2. Currently the study appears justified primarily by the availability of different complementary resources, and not so much by a conceptual question or hypotheses. Could the authors elaborate on why combining structural connectivity and arousal / gene expression data is interesting, and whether similar approaches have been performed in other species?

Response:

We sincerely apologize for not making the rationale clear. While many studies have already examined the structural or arousal contribution to resting-state functional connectivity (as we introduced in Page 5), very few of them directly examined whether and how those factors are related to (dynamic) FC gradients, which is the main question that we hope to address in the current study. A previous human study showed the structure-function tethering was heterogeneous and negatively correlated with the principal functional gradient¹⁹. Another mouse study found significant correlations between mouse functional gradients and gene expression patterns²⁰. Although previous studies have suggested the link between functional gradients and structural connectivity or gene expression profiles, it remains unclear whether the functional gradients are dynamic with regard to arousal fluctuations, and if so, how structural connectivity and arousal related gene expression jointly contribute to such dynamics. And marmosets, as an emerging neuroscience animal model, is uniquely suited to address this question with the large awake resting-state fMRI data, retrograde tracing based structural connectivity dataset and ISH gene expression datasets. Particularly, the retrograde tracing based structural connectivity is considered the “gold standard” and does not suffer from relatively low reliability of human diffusion MRI based tractography. Therefore, we believe that the current study is uniquely suited to address the dynamics and biological basis of the functional gradients by combining the above datasets.

Now we added the following part in the introduction (Page 6 Line 3):

“While many studies have already examined the structural or arousal contribution to resting-state functional connectivity as mentioned above, very few of them directly examined whether and how those factors are related to (dynamic) FC gradients. A previous human study showed the structure-function tethering was heterogeneous and negatively correlated with the principal functional gradient¹⁹. Another mouse study found significant correlations between mouse functional gradients and gene expression patterns²⁰. Although previous studies have suggested the link between functional gradients and structural connectivity or gene expression profiles, it remains unclear whether the functional gradients are dynamic with regard to arousal fluctuations, and if so, how structural connectivity and arousal related gene expression jointly contribute to such dynamics. And marmosets, as an emerging neuroscience animal model, is uniquely suited to address this question.”

3. Gradient ordering is interpreted as a key result ("suggesting a potential link between developmental and evolutionary processes"), but switches in gradient ordering between eg first and second gradient could plausibly happen, even in young adult samples (including different HCP subsamples). This has to do with the eigenvalues of the first and

second functional gradients often being quite close to one another. In that respect, the 'link' to children/adolescent/adult findings from completely different datasets with different acquisition parameters etc also appears somewhat selective and qualitative without further analyses, including some more exhaustive assessments of within sample switching in gradient order (eg via bootstraps etc).

Response:

We really appreciate this constructive comment, and we fully agree that the gradient ordering could potentially switch, especially at the individual EPI level. Therefore, we examined the stability of the gradient ordering in our results, and we found gradient ordering in our datasets was generally stable at the level of individual marmosets (Fig. R17a) or humans (Fig. R17b). One possible reason for such stability is that both marmoset and human data are all from adult marmosets (marmoset ages range from 2-4 years, marmosets typically reach adulthood at 1.2 year) or adult humans (22-36 years).

To further investigate the potential link between developmental and evolutionary processes, we made more formal statistical cross-species comparison using the fingerprinting method³⁻⁵. As mentioned in previous studies, the fingerprinting method afforded the ability of cross-species comparison and could bridge the gap between marmoset and human neuro-anatomy. We included 11 target regions based on a previous cross-species study⁶. The corresponding brain regions included 6 heavily myelinated and 5 lightly myelinated regions, i.e., early somatomotor (1), auditory (2), early visual (3), middle temporal (MT) complex (4), parietal (intraparietal sulcus) visual (5), and retrosplenial cortex (6), prefrontal (A), lateral parietal (B), lateral temporal (C), medial parietal (D), and insular (E) cortex (Fig. R17c). First, we extracted gradients values of marmoset, human children and human adults in the 11 target regions and obtained the cross-species gradient fingerprints (Fig. R17d upper panel). Then, we calculated the Pearson's correlation coefficients (C.C.) between each pair of gradient fingerprints (Fig. R17d, lower left part). Finally, we clustered the gradient fingerprints using the method of community detection (Fig. R17d, lower right part), and found the marmoset functional gradients closely resembled the children cortical organization of human.

To further evaluate whether individual marmoset gradients also share the same relationship with human children gradients, we compared the first two gradient maps across individual EPI sessions (Fig. R17e). We found the marmoset functional gradients showed high stability across EPI sessions in both ION and NIH dataset (Fig. R17e). Meanwhile, we calculated the Pearson's correlation coefficients between marmoset and human children functional gradients (Fig. R17f) at the individual EPI level. The scatter plots showed marmoset gradients at the individual EPI level also shared the same relationship with the average functional gradients on cross-species gradient comparison.

Thus, while the functional connectivity profiles were from different datasets, we could still conclude the similarity between marmoset and human children functional gradients is most likely valid. We have revised our manuscript in Page 10 Line 10.

(c) Eleven cross-species landmark regions for gradient fingerprint analysis. The figure was adapted from Van Essen, et. al., PNAS, 2019. (doi:10.1073/pnas.1902299116)

(d) Marmoset functional gradients closely resembled those of human children. Based on gradient values from 11 cortical landmark regions, we extracted the cross-species gradient fingerprints (upper) and calculated the Pearson's correlation coefficients (C.C.) between each pair of gradient fingerprints. Using the method of community detection, we revealed 2 communities of cross-species gradients, suggesting the marmoset functional gradients closely resembled the children cortical organization of human.

(e) High cross-session functional gradient similarities in both ION and NIH dataset. FG, functional gradient.

(f) Marmoset functional gradients also closely resembled the human children gradients at the individual EPI level. Each dot represented an individual EPI session. Black "+" markers indicated marmoset functional gradients derived from the mean functional connectivity matrix.

4. The study utilizes functional connectivity, structural connectivity and gene expression data from different marmoset samples. Unless there is validation data from the sample primates available, can the authors comment on potential limitations of such an approach, given that inference is based on between-dataset associations. Furthermore, could they provide further details on how the marmosetbrain.org tracer based connectivity matrix was derived? Likewise for the marmoset gene expression atlas. Are sex distributions similar across datasets, and are ages comparable?

Response:

We fully agree that it is a limitation that our functional connectivity, structural connectivity and gene expression are from different marmosets. First we'd like to provide more details for generating structural connectivity and gene expression data:

For the procedure of deriving the marmoset structural connectivity matrix, it is described in details in their related publication²¹ (<https://www.marmosetbrain.org/>). Briefly, the raw data include 143 injections of retrograde tracers in 52 young adult (1.4–4.6 years, median age: 2.5 years, 31 male, 21 female), and standard histological procedure was applied. Digitized histological sections were 3D reconstructed and registered to a template. Injection sites and retrograde labeled cells were assigned to cortical areas based on the atlas parcellation. And finally the structural connectivity matrix was generated by compiling data from all injection experiments. Such structural connectivity matrix is available for download and utilized in our current study. We have revised the method section (Page 50 Line 4) to add more descriptions of the procedure.

For the marmoset gene expression atlas, the experimental procedure was described in details in their previous publications^{22,23}. Due to the ethics consideration to reduce animal usage, data were largely from marmoset infants or juveniles (age from neonate to 6 months, with only a few 1 year old animals) from both sexes. This is because some marmosets are born triplets but the marmoset parents can only care for two, so the third one is typically abandoned by the parents and used in such terminal experiment. As for the analysis procedure, we have now substantially expanded our description of how to generate the gene expression data (Fig. R20, shown in Comment 6) and revised our methods section

as follows (Page 51 Line 3):

“The neuromodulatory receptor gene expression information was obtained from the marmoset gene atlas database (<https://gene-atlas.brainminds.riken.jp/>). The related gene list and associated age and sex information is listed as Supplementary file 1. The whole pipeline was summarized in Fig. S20. First, we mapped the receptor expression maps to the Nissl stained images, using “rigid-body transformation” for the coarse whole brain registration and “large deformation diffeomorphic metric mapping (LDDMM)” for more subtle slice-by-slice registration. To facilitate the comparison with fMRI results, the Nissl stained images was registered to the study-specific MRI template by affine nonlinear transformation (“oldnormalize” of SPM12) and the affine transformation matrix was also applied to the receptor expression map. Then, receptor expression data were parcellated into 116 cortical regions of interest, based on the Riken Brain/MINDS cortical parcellation. Finally, the resulting neuromodulatory receptor similarity was calculated using the correlation of normalized gene expression level across each pair of regions.”

For the age and sex information, we have now added the information in their respective method sections (Page 50 Line 4 and Page 51 Line 3). As mentioned above, one notable difference is the gene expression data are from younger animals than the other two types of data, but this gene expression database is currently the only publically available one and leaves us no better alternatives. Nevertheless, a limited number of genes have data from multiple ages and/or both sexes, and we examined two receptor genes (DRD1 and CHRM3) (Fig. R18). For DRD1 expression, data from P0 and 1 year (Fig. R18a-b) old animals showed largely consistent spatial patterns. CHRM3 expression also exhibited consistent patterns in 3 one year old animals (1 female and 2 males, Fig. R18c-e), indicating stable expression between sexes and across individual animals. Systematical investigation of developmental and sex influences on those neuromodulatory receptor expression levels is out of the scope of the current study, but based on the very limited two examples above, at least the neuromodulatory receptor gene expression do not seem to change drastically over the developmental stages or sexes.

To conclude, we added the discussion regarding the limitation of using data from different marmosets as follows (Page 32 Line 9):

“Finally, the current study is limited by the fact that functional connectivity, structural connectivity and gene expression data from different marmosets, and as such, age, sex and the individual differences may limit our inference. In particular, the functional connectivity and structural connectivity data were both from adult marmosets, while the gene expression data were largely from infant or juvenile marmosets (Supplementary File 1). Such age difference may potentially lead to biases across the three data types. Nevertheless, we examined expression patterns of DRD1 and CHRM3 (Fig. S26) and found relative stable patterns across age and sexes. Future detailed examination is required to systemically investigate the age and sex dependence of the neuromodulatory receptor gene expression, especially when adult gene expression data become more readily available. In addition to age and sex, the individual difference may also lead to potential instability in our results. However, due to the nature of the tracer injection and ISH experiments, it is not feasible to collect all data from one single animal and will require technical improvement in the future, such as spatial transcriptomics.”

Figure R18. Consistency of marmoset gene expression information across sex, age and subjects. (Now Figure S26 in our revised Supplementary materials)

(a-b) Expression pattern of marmoset dopamine receptor D1 (DRD1) from P0 (a) and 1 year (b) old marmosets. Left, example of marmoset gene expression images in coronal

view. Right, quantitative evaluation of marmoset gene expression profiles. Insert, definitions of marmoset cortical parcellations used in the right panel. a, anterior; p, posterior. The parcellations and quantifications of expression levels were directly obtained from marmoset gene atlas database (<https://gene-atlas.brainminds.riken.jp/>). (c-e) Similar to (a-b) but for cholinergic receptor muscarinic 3 (CHRM3) from a female 1 year old marmoset (c) and two male 1 year old marmosets (d-e).

5. The HCP dataset contains nested family relationships. (1) Wouldn't it make more sense to study unrelated individuals instead to make data similar in scope to the unrelated marmoset datasets? (2) Also, there are some divergences with respect to the processing of marmoset and human fMRI data. Couldn't this difference also induce differences across species? Please discuss and/or run supporting analyses based on harmonized processing in both species.

Response:

(1) We'd like to thank Reviewer to raise this point about the family relationship in the HCP dataset. For marmosets in the ION dataset, none of them are siblings. But the records of their grandparents are missing so we don't know if they're "cousins". We do not have information regarding the NIH dataset. So indeed it's likely that the family relationship between the marmoset and human datasets is different. However, as we showed in Fig. S4 and Fig. S8, the overall spatial patterns and the ordering of the functional gradients in both human and marmoset are relatively stable at the individual level, so it's unlikely that family relationship difference would significantly affect our cross-species comparison. We have added this point as a limitation in the discussion section (Page 31 Line 21) as follows:

"Finally, for the cross-species comparison of the functional gradient characteristics in marmosets and humans, several factors are different across the animal and human datasets, which may complicate such comparison. For example, the high prevalence of nested family relationship in the human HCP dataset does not exist in our marmoset dataset. However, as we showed in Fig. S4 and Fig. S8, the overall spatial patterns and the ordering of the functional gradients in both human and marmoset are relatively stable at the individual level, so it's unlikely that family relationship difference would significantly affect our cross-species comparison."

(2) As suggested, we applied the same preprocessing on human HCP dataset as in the marmoset dataset, and found highly similar functional gradient profiles with the "12 rp + 10 PCs" regression, compared to the original ICA-FIX strategy (Fig. R19). Furthermore, in the original children functional gradient paper²⁴, functional gradient patterns remained robust with or without global signal regression and with alternate motion exclusion criterion. Thus, we concluded that the different processing for fMRI datasets is unlikely to induce significant differences of our gradient comparison across species. We have added above results in our manuscript (Page 10 Line 17).

Figure R19. Pre-processing strategies did not significantly influence the results of functional gradients. (Now Figure S9 in our revised Supplementary materials)

(a-b) The human functional gradients with different preprocessing strategies, i.e., ICA-FIX (HCP) and “12 rp + 10 PCs” regression (used in marmoset preprocessing strategy).

(c) Significant correspondence between the functional gradients with the two different preprocessing strategies in human HCP data.

6. Further details on the gene expression dataset may be worthwhile. At the moment, this analysis appears underspecified. (1) How were receptor expression maps aligned with nissl images? (2) Are there estimates of accuracy of this cross-modal alignment or prior papers. Ditto for the registration between nissl images and fMRI templates - one could imagine that these sources have quite a different scale. (3) Was there any filtering of gene expression information performed, to e.g. explore cross-subject consistency?

Response:

(1) We sincerely apologized that we did not describe the methods in more details. As mentioned, we revised our method (Page 51 Line 3) in our manuscript (Fig.R20), as follow:

“The neuromodulatory receptor information was obtained from the marmoset gene atlas database (<https://gene-atlas.brainminds.riken.jp/>). Registration of marmoset ISH images to MRI space was summarized in Fig. S20. Briefly, we downloaded the Nissl stained coronal images and neuromodulatory receptor related gene expression maps. Next, we mapped the receptor expression maps to the Nissl stained images, using “rigid-body transformation” for the coarse whole brain registration and “large deformation diffeomorphic metric mapping^{12,13} (LDDMM)” for more subtle slice-by-slice registration. To facilitate the comparison with fMRI results, the Nissl stained images were registered to the study-specific MRI template by affine nonlinear transformation (“oldnormalize” of SPM12) ,and the affine transformation matrix was then applied to the receptor expression map to bring it to the MRI space. Also, the median filter was applied to each set of ISH data to improve the data quality. Next, receptor expression data were parcellated into 116 cortical regions of interest, based on the Riken Brain/MINDS cortical parcellation. Finally, the resulting neuromodulatory receptor similarity was calculated using the correlation of gene expression level across each pair of regions.”

(2) From visual inspection (Fig. R20 c-e), we considered the overall quality of registration satisfactory. As the analysis using the gene expression data was all done at brain regional

level, we believe such registration quality is sufficient.

(3) Median filter was applied to each set of ISH data to improve the data quality. For the marmoset ISH data, as described in details in their previous publications^{22,23}, ISH data of one given gene were collected mostly from one single animal and very few were from multiple animals. In our response to Comment 4 above, we compared CHR3 in 3 animals and overall patterns were consistent. While it is not possible to systematically explore cross-subject consistency, we believe it is unlikely that individual differences would change our results significantly. In the end, almost all studies have to use ISH data from different human subjects or animals, which is an intrinsic limitation of the method and hopefully can be overcome by newer methods such as spatial transcriptomics.

Figure R20. Registration of marmoset ISH images to MRI space. (Now Figure S20 in our revised Supplementary materials)

(a) Computational strategy of marmoset ISH registration and generation of neuromodulatory receptor similarity. The neuromodulatory receptor gene expression data were obtained from the marmoset gene atlas database (<https://gene-atlas.brainminds.riken.jp/>). Briefly, we downloaded neuromodulatory receptor related gene expression maps and the corresponding Nissl stained coronal images. Next, we registered the expression maps to the Nissl stained images, yielding a 3D spatial alignment between anatomical and gene expression images (details in Fig. b). The Nissl stained images were registered to the study-specific MRI template (details in Fig. c), and this transformation was applied to the gene expression maps to bring them to the MRI space. Gene expression data were then parcellated into 116 cortical regions of interest, based on the Riken Brain/MINDS cortical parcellation. Finally, the resulting neuromodulatory receptor similarity was calculated using the correlation of normalized gene expression levels across each pair of regions.

(b) Registration from receptor gene expression maps to Nissl images. Firstly, these ISH images were down-sampled (5 times) in gray scale to reduce the computation load. Then, these Nissl and receptor expression images were iteratively aligned to a weighted average of its neighbors, respectively. The receptor expression images were de-noised using the standard medial filter and further threshold by the median values of the ISH images, termed as the “energy map” (bottom image in b). Finally, the 3D receptor expression maps were registered to the Nissl images using rigid-body transformation, and the transformation matrix was applied to the receptor energy maps. Further adjustment of registration was made slice-by-slice using the method of large deformation diffeomorphic metric mapping (LDDMM).

(c) Cross-modal registration from Nissl images to MRI template. The 3D Nissl images were nonlinearly transformed to the MRI template using the “oldnormalize” of the SPM12. The affine transformation was then applied to the receptor expression energy map.

(d) Example of registered and normalized CHRM3 receptor expression energy map. The expression energy was overlaid on the MRI template in coronal slice view (upper panel) and 3D surface view (lower panel).

7. A common variable contributing to structure-function relationships in the nervous system of humans and animals is spatial proximity and autocorrelation. Can the authors detail in how far sources of **autocorrelation** were addressed in the current study. For example, it was not clear to me how spatial autocorrelation was accounted for when assessing correspondences and associated significances of correlations between structural and functional gradients (e.g. Figure 2b, p-values appear unadjusted). For further details, see eg Burt et al (<https://doi.org/10.1016/j.neuroimage.2020.117038>) or Markello et al. (<https://pubmed.ncbi.nlm.nih.gov/33857618/>).

Response:

We really appreciate Reviewer’s constructive comment. We fully acknowledge that our original result did not treat the spatial continuity and null models very well. Because the

functional gradients are spatially auto-correlated, we adopted a procedure from previous studies⁹⁻¹¹ (like ref #9 Fig.3C) to overcome this issue (Fig. R21a,b). We generated surrogate maps that randomly varied in their particular topographies (n = 1000 times shuffling) but preserved the general spatial autocorrelation (SA) structure (Fig. R21b). As expected, the SA preserving surrogate maps showed a much wider distribution of correlation with the empirical one, compared to the SA independent surrogate maps. Using null distribution generated from SA preserving surrogate maps, the correlations between structural and functional gradients in Fig. 2b remained significant (Fig. R21c). We have now revised Fig. 2 to reflect this change. We have revised our method (Page 54 Line 2) and results section (Page 11 Line 13) in our manuscript.

Reference

- 1 Afonso C Silva, J. V. L., Yoshiyuki Hirano, Renata F Leoni, Hellmut Merkle, Julie B Mackel, Xian Feng Zhang, George C Nascimento, Bojana Stefanovic. *Longitudinal functional magnetic resonance imaging in animal models*. Vol. 711 (Humana Press, 2011).
- 2 Chen, X. *et al.* Sensory evoked fMRI paradigms in awake mice. *NeuroImage* **204** (2020).
- 3 Balsters, J. H., Zerbi, V., Sallet, J., Wenderoth, N. & Mars, R. B. Primate homologs of mouse cortico-striatal circuits. *Elife* **9**, doi:10.7554/eLife.53680 (2020).
- 4 Schaeffer, D. J. *et al.* Divergence of rodent and primate medial frontal cortex functional connectivity. *Proc Natl Acad Sci U S A* **117**, 21681-21689, doi:10.1073/pnas.2003181117 (2020).
- 5 Mars, R. B., Sallet, J., Neubert, F. X. & Rushworth, M. F. Connectivity profiles reveal the relationship between brain areas for social cognition in human and monkey temporoparietal cortex. *Proc Natl Acad Sci U S A* **110**, 10806-10811, doi:10.1073/pnas.1302956110 (2013).
- 6 Van Essen, D. C. *et al.* Cerebral cortical folding, parcellation, and connectivity in humans, nonhuman primates, and mice. *Proc Natl Acad Sci U S A*, doi:10.1073/pnas.1902299116 (2019).
- 7 Lindquist, M. A., Xu, Y., Nebel, M. B. & Caffo, B. S. Evaluating dynamic bivariate correlations in resting-state fMRI: a comparison study and a new approach. *Neuroimage* **101**, 531-546, doi:10.1016/j.neuroimage.2014.06.052 (2014).
- 8 Gu, Y. *et al.* Brain Activity Fluctuations Propagate as Waves Traversing the Cortical Hierarchy. *Cereb Cortex* **31**, 3986-4005, doi:10.1093/cercor/bhab064 (2021).
- 9 Demirtas, M. *et al.* Hierarchical Heterogeneity across Human Cortex Shapes Large-Scale Neural Dynamics. *Neuron* **101**, 1181-1194 e1113, doi:10.1016/j.neuron.2019.01.017 (2019).
- 10 Burt, J. B., Helmer, M., Shinn, M., Anticevic, A. & Murray, J. D. Generative modeling of brain maps with spatial autocorrelation. *Neuroimage* **220**, 117038, doi:10.1016/j.neuroimage.2020.117038 (2020).
- 11 Markello, R. D. & Misic, B. Comparing spatial null models for brain maps. *Neuroimage* **236**, 118052, doi:10.1016/j.neuroimage.2021.118052 (2021).
- 12 Lin, M. K. *et al.* A high-throughput neurohistological pipeline for brain-wide mesoscale connectivity mapping of the common marmoset. *Elife* **8**, doi:10.7554/eLife.40042 (2019).
- 13 Brian C Lee, M. K. L., Yan Fu, Junichi Hata, Michael I Miller, Partha P Mitra. Multimodal cross-registration and quantification of metric distortions in marmoset whole brain histology using diffeomorphic mappings. *The Journal of comparative neurology* **529**,2, doi:10.1111/cne.24946 (2021).
- 14 Okano, H. *et al.* Brain/MINDS: A Japanese National Brain Project for Marmoset Neuroscience. *Neuron* **92**, 582-590, doi:10.1016/j.neuron.2016.10.018 (2016).
- 15 Cross, N. *et al.* Cortical gradients of functional connectivity are robust to state-dependent changes following sleep deprivation. *Neuroimage* **226**, 117547, doi:10.1016/j.neuroimage.2020.117547 (2021).
- 16 Huntenburg, J. M., Bazin, P. L. & Margulies, D. S. Large-Scale Gradients in Human Cortical Organization. *Trends Cogn Sci* **22**, 21-31, doi:10.1016/j.tics.2017.11.002 (2018).
- 17 Wagstyl, K., Ronan, L., Goodyer, I. M. & Fletcher, P. C. Cortical thickness gradients in structural hierarchies. *Neuroimage* **111**, 241-250, doi:10.1016/j.neuroimage.2015.02.036 (2015).
- 18 Liu, C. *et al.* Marmoset Brain Mapping V3: Population multi-modal standard volumetric and

- surface-based templates. *Neuroimage* **226**, 117620, doi:10.1016/j.neuroimage.2020.117620 (2021).
- 19 Vazquez-Rodriguez, B. *et al.* Gradients of structure-function tethering across neocortex. *Proc Natl Acad Sci U S A* **116**, 21219-21227, doi:10.1073/pnas.1903403116 (2019).
- 20 Huntenburg, J. M., Yeow, L. Y., Mandino, F. & Grandjean, J. Gradients of functional connectivity in the mouse cortex reflect neocortical evolution. *Neuroimage* **225**, 117528, doi:10.1016/j.neuroimage.2020.117528 (2021).
- 21 Majka, P. *et al.* Open access resource for cellular-resolution analyses of corticocortical connectivity in the marmoset monkey. *Nat Commun* **11**, 1133, doi:10.1038/s41467-020-14858-0 (2020).
- 22 Shimogori, T. *et al.* Digital gene atlas of neonate common marmoset brain. *Neurosci Res* **128**, 1-13, doi:10.1016/j.neures.2017.10.009 (2018).
- 23 Kita, Y. *et al.* Cellular-resolution gene expression profiling in the neonatal marmoset brain reveals dynamic species- and region-specific differences. *Proc Natl Acad Sci U S A* **118**, doi:10.1073/pnas.2020125118 (2021).
- 24 Dong, H. M., Margulies, D. S., Zuo, X. N. & Holmes, A. J. Shifting gradients of macroscale cortical organization mark the transition from childhood to adolescence. *Proc Natl Acad Sci U S A* **118**, doi:10.1073/pnas.2024448118 (2021).

REVIEWER COMMENTS

Reviewer #1 (Remarks to the Author):

The authors have adequately addressed my concerns.

Reviewer #2 (Remarks to the Author):

Please see attached file

Reviewer #3 (Remarks to the Author):

The authors have satisfactorily responded to each of my concerns.

Reviewer #4 (Remarks to the Author):

I am overall convinced by the authors' revisions. Many thanks.

Reviewer #2

R2C#1 (Reviewer 2's comment #1)

I truly appreciate the diligent work the authors have done to address my questions/comments in the first run. However, some of these new results raised more questions than answered. So, either I don't really understand the methods or the authors made fundamental mistakes in using and interpreting these models.

1. My biggest concern is about the modeling frame, particularly the reduced model, used for the results presented in Figures 4 and 5, which are the major innovation of this study. According to the Methods section, this reduced model is very similar to the single variable model. The only difference is a better control of the degree of freedom with shuffling the other variable. By comparing with the full model, one can estimate the unique contribution of the variable on the response variable (here should be FC) based on the change of explained variance, just like in the ref #60 cited by the paper. But I don't believe it was used properly in this paper. Both Figs. 4d and 4f compared the gradients of predicted FC from these two variables (i.e., the SC and NS), and Fig. 4e is just an illustration without any quantifications about how much SC and NS contribute to the predicted FC and thus the resulting FC gradients. (1) For this part of the analysis, the authors should show how much of the variance of the response variable, i.e., the FC, is explained by two independent variables (i.e., the SC and NS). Or, if the principal gradient of the FC is of the interests, the authors need to show the similarity between the empirical gradient and predicted gradient derived from the single variable model or the reduced model, as what they did for the full model in Figure 4C. (2) I also encourage the authors to check the results shown in Figs. 4d and 4f. As discussed above, the reduced model should produce very similar results as the single variable model by adding a randomly shuffled variable. This appears to be true based on the visual inspection on the gradients. They look indeed very similar to each other. However, it is surprising that the spatial similarity of the gradients from the predicted data was changed dramatically.

Response:

(1) We sincerely apologize that such important information was missing in Fig. 4e. Now the revised Fig. 4e showed the variances partitioning on a full model prediction of empirical gradients, in which the unique contribution of SC and NS was 41.07% and 26.35%, respectively, while the overlap of SC and NS effects was 32.58% (Fig. R7a).

And also, now we added a supplementary figure (Fig. R7f, now added as Figure S22) to better explain the full and reduced model.

R2C#2

How were those numbers obtained? They are inconsistent with the other results shown in the paper. These numbers are inconsistent with the % variance reported in other parts of the figure, e.g., Figs. 4b, 4c, 4f, 5b, 5d, that are much lower than those numbers. It seems that the % contributions in Fig 4e sums up to 100%, which cannot be true. The % contribution is often quantified by the explained % variance,

which shouldn't be normalized! This needs to be corrected.

(2) Thanks a lot for the reviewer's insightful comment. After close inspection, we indeed found that our original results in Fig. 4d and 4f showed very similar profile, potentially leading to the misunderstanding of the modeling results. We found this phenomenon was mainly due to the narrow range of the color bar. After broadening the range of color bars in revised Fig. 4d and 4f, the predicted functional gradients from single variable model and reduced model appears more dissimilar upon visual inspection (Fig. R7b-e).

Nevertheless, it is somewhat expected that the spatial similarity of the gradients from the predicted data changed dramatically from the single variable model to the reduced model. As shown in Fig. R7a, there is large overlap between the SC (Fig. R7a, green circle) and NS (purple circle), resulting the spatial similarity on the single variable model prediction (Fig. R7b). Thus, we utilized the reduced model to capture the unique contribution of each variable. In the reduced model, the variable of interest was randomly shuffled 1000 times, and the resulting loss of explained variance (compared to the full model) was the unique contribution of this particular variable in the reduced model (Fig. R7f). Therefore, for a particular variable, the reduced model and the single variable model is expected to be different.

The dramatic decrease of spatial correlation on the reduced model attributed to the separation of unique contributions of SC (Fig. R7a, pure green part) and NS (pure purple part). In other words, the non-overlapping parts of SC and NS effects on reduced model resulted the non-significant spatial similarity on predicted gradients (Fig. R7c). We have revised our manuscript and added more details about the reduced model (Line 17 Line 5).

R2C#3

I believe that the authors had some fundamental misunderstanding about the GLM. The authors keep saying that they "utilized the reduced model to capture the unique contribution of each variable". According to the description and also Fig R7f, they actually used the R^2 DIFFERENCE between the full model and the reduced model to capture the unique contribution of each variable. They are different! The only explanation I can think of is that the authors only used the "reduced model" to denote the R^2 difference between the full model and real reduced model. Then, my question is how the authors obtained those predicted gradients shown in Fig 4h. If they used the predicted values in the reduced models, i.e., \hat{FC}_a and \hat{FC}_b in Fig R7f, it is totally wrong. If not, how did they get those maps?

Figure R7. Reduced GLM model dissected unique contributions of structural connectivity and neuromodulatory inputs to the functional gradients.

(a) Reduced model for investigating the unique distribution of each variable (non-overlapping part). The structure connectivity or neuromodulatory receptor similarity (circle) may have overlapped information with the other one, while the reduced model (non-overlapping part) provides the unique contribution of each predictor. (Now Figure 4e in our revised manuscript).

(b) Low dimensional topography contributed by structure connectivity (SC) and neuromodulatory similarity (NS), respectively. Significant spatial correlation indicated largely overlaps between predictors.

(c) As in (b) but for reduced model. Distinct gradient profile (red and blue arrows) of the unique contribution from structure connectivity and neuromodulatory similarity exhibited non-significant spatial correlation.

(d) Similarity between SC and NS driven gradient 1 for single variable model prediction (left panel) and reduced model prediction (right panel). Each dot represented a brain region. Red solid line indicated the best linear fitting, with two dash lines as 95% confidence interval. (e) As in (d) but for gradient 2.

(f) Illustration of the full general linear model and the reduced model. (Now Figure S22 in our revised Supplementary materials)

2. The principal gradient of the neuromodulatory similarity should be presented and compared with the FC gradient, similar to what they've done for the structural connectivity (Fig. 2). The gradient maps of the neuromodulatory similarity shown in Figures 4d and 4f do not appear to be similar to the FC gradients to me.

Response:

As suggested, we conducted similar analyses on the neuromodulatory similarity, and found the neuromodulatory similarity gradients were also correlated with the functional gradients (Fig. R8), but to a less extent compared to the structural gradients. We have added the neuromodulatory gradients in our manuscript (Page 16 Line 6).

R2C#4

Thanks for the new results that I suggested in the 1st-round of review. The NG2 doesn't look like the FG 2 (Fig. 1a), and they showed opposite signs in frontal and visual cortices). Something appears to be wrong for the statistical testing shown in Fig R8b (the second column in the bottom row). The area of the null distribution on the right side of the vertical lines is apparently larger than 0.01% of the total area. So, I don't know how those vertical lines (real correlations) would be corresponding to a p-value of $<10^{-4}$.

Figure R8. Gradients of neuromodulatory similarity in marmoset. (Now Figure S21)

in our revised Supplementary materials)

(a) First four gradients of marmoset neuromodulatory similarity based on marmoset gene atlas. Areal borders were based on the Riken Brain/MINDS cortical parcellation. NG, neuromodulatory gradient.

(b) Topographical similarity between neuromodulatory and functional gradients. High similarity between neuromodulatory and empirical functional gradients (upper panel). Significant difference between the empirical similarity of neuromodulatory-functional gradients and null distributions (lower panel), in which null distributions was derived from the Pearson's correlation coefficients (gray shades) between neuromodulatory gradients and the spatial autocorrelation preserving surrogate maps of functional gradients. Each dot represented a brain region. Red and blue lines represented the Pearson's correlation between empirical neuromodulatory and functional gradients.

3. The gradient dynamics shown in Figure 5 is somewhat expected, especially if the neuromodulatory similarity doesn't produce similar gradients as the FC. The predicted FC (fitted FC) is the projection of empirical FC onto a subspace spanned by the independent variables, i.e., the SC or NS, and thus contained the information of real data. As shown in Figure 2, the SC gradients are highly similar to the FC gradients. It is thus not surprising to see a similar invert-U shape of modulation of SC-predicted gradient on the arousal index. At the same time, one would expect an opposing U-shape modulation of the gradients for any other variables that are not similar to the FC gradients, which could be the case for the NS-predicted FC gradients. In other words, the U-shape modulation should not be specific to the NS-predicted FC gradients but to anything not having a tight link with the FC gradients, as the "others" shown in Figure 5h.

Response:

We hope we understand your comment correctly. First, as shown in our response to the previous comment (Fig. R8), the neuromodulatory similarity gradients are also correlated to the FC gradients, but such correlation is lower than that of structural connectivity. This result is also expected, as the two variables (NS and SC) showed large overlap (Figure R8a) in our model. Not surprisingly, NS and SC showed correlation (Fig. R9a).

Next, we do not feel it can be entirely predicted that SC-FC gradient similarity alone could guarantee an "invert-U shape of modulation of SC-predicted gradient on the arousal index". The SC related modulation on the FC gradient dynamics could be any shape, or even no modulation at all, given there's "other" variables in the model. Indeed in our results, we observed a U shape of unique modulation of NS-predicted gradient on the arousal index, opposite to that of SC-predicted gradient (invert-U shape).

To empirically evaluate random or non-NS related variables' contribution, we added a noise term (Fig. R9b) or an expression similarity matrix based on randomly selected genes ((Fig. R9c) (gene list can be found in Supplementary File 1) in our model. The results showed that the random noise component did not exhibit significant contribution across arousal fluctuations (Fig. R9d-e). And the random expression similarity matrix, exhibiting very low degree of correlation with empirical FC (Fig. R9f), showed very minor contribution compared to the NS (Fig. R9g). Therefore, the above new analysis suggests that the NS

and SC contribution to the arousal related gradient dynamics is likely to be specific. However, we fully agree that our current model is in no way exhaustive, in terms of including all potential predictors in the model. We hope we can reveal additional contributing factors in the future, with more marmoset brain resources available. Now, we have revised our manuscript and added such control analysis in Page 19 Line 13.

R2C#5

Again, the new results confused me more.

First, what is the “explained variance %” in Fig 5b and 5d-e. Are those R^2 from the full and reduced models? If yes, they at most represent the contribution of SC and/or NS to FC, but not necessarily to the FC gradients, since we don’t know which specific part of FC leads to the FC gradients. Similar to what I suggested in R2C3, I don’t understand how those predicted gradients are related to this explained variance %.

Second, why did the Fig R9e-R9g use the “delta R^2 ” but not the explained variance % as shown in Fig 5. If the “delta R^2 ” between the full model and the control B model are around zero as shown in Fig R9g , does this mean that the ISH similarity also account for the FC variance just like NS? Or, the “delta R^2 ” is not the difference between the full model and control B. Again, I just want to say that these results are very confusing.

Figure R9. Noise component and random ISH expression similarity did not drive inverted U-shape relationship on unique gradient dynamics with arousal fluctuation. (Now Figure S24 in our revised Supplementary materials)

(a) Significant correlation between structural connectivity (SC) and neuromodulatory similarity (NS), suggesting the overlap between SC and NS.

(b-c) Control analysis of the U-shape modulation for the neuromodulatory similarity (NS) The control analysis was conducted by adding a noise matrix (NM) as another variable (b) or replacing the NS with random ISH expression similarity (c, RS).

(d) Example of a random noise component which did not have a tight link with the empirical functional connectivity.

(e) No significant arousal relevant unique contribution from the noise component. Across arousal levels, the unique contribution (ΔR^2) was not significantly different from zero (one-sample t-test, two tails).

(f) Example of the random ISH genetic similarity which showed a significant correlation with the empirical functional connectivity.

(g) Inverted U-shape relationship of arousal relevant unique contribution from the random ISH genetic gradient 1, with no significant arousal relevance on the gradient 2.

5. I also have some questions regarding the last part of the results related to the receptor maps. First, the spatial correspondence between the receptor maps and the FC gradients are very weak with a R-square of 1~2% (Figs. 6 and S12). The authors showed some of these spatial correlations are statistically significant. But there were two potential issues. (1) These brain maps are spatially continuous, which means that the parcels/voxels are not independent and thus the real degrees of freedom should be much smaller than the number of parcels/voxels. Therefore, simply shuffling the parcels/voxels for the receptor maps is not a correct way of building the null distribution. The reported results were not corrected for multiple comparison. (2) Second, if the paper meant to emphasize the role of the neuromodulatory system in generating the FC gradients, the proper controls need to be included. The authors need to show that the neuromodulatory receptors show a stronger spatial correspondence with the FC gradients than other non-neuromodulatory receptors. (3) Also, the spatial similarity itself is not a strong indicator of a tight relationship. For example, it has been shown that the primary gradient of gene expression in the brain is similar to the FC gradient in human (the FC gradient #2 in this paper) (Burt, JB et. al., Nature Neuroscience, 2018). One would thus expect that a significant proportion of genes have significant spatial correlation with the FC gradient. However, it doesn't mean that all of these genes would play a role in generating this FC gradient.

Response:

(1) We really appreciate Reviewer's constructive comment. We fully acknowledge that our original result did not treat the spatial continuity and null models very well. Because the neuromodulatory receptors are spatially auto-correlated, we adopted a procedure from previous studies⁹⁻¹¹ (similar to Fig.3C in Demirtas et al., ref #9) to overcome this issue (Fig. R10a).

First, we generated surrogate maps that randomly vary in their particular topographies ($n = 1000$ times shuffling) but preserve the general spatial autocorrelation (SA) structure of corresponding receptors (Fig. R10a). As expected, the SA preserving surrogate maps showed a much wider distribution of correlation with the empirical one, compared to the SA independent surrogate maps (Fig. R10b). Using those SA preserving surrogate maps, we found similar results compared with our original result from SA independent surrogate maps (Fig. R10c, CHRM3 as an example). Moreover, to evaluate whether the arousal modulation from each neuromodulatory receptor was overestimated, we calculated the Pearson's C.C. (right tail) between the surrogate similarity (to the empirical one, $n = 1000$) and the unique contribution on functional gradients of a particular receptor (Fig. R10d, like Fig.3E in Demirtas et al., ref #9). The rationale is: if the correlation is not significant, it means random receptor maps could contribute similar arousal modulation, i.e., the empirical receptor does not specifically contribute to arousal dynamics. Alternatively, if the correlation is significant, it means larger spatial map shuffling causes larger loss of the unique contribution for the corresponding receptor, i.e., the empirical receptor does contribute to arousal dynamics. Our result showed CHRM3 receptor contributed significantly to gradient 1 arousal dynamics, but not significantly to gradient 2 (Fig. R10d). Finally, we applied the false discovery rate (FDR) correction on all receptors and found several receptors showing statistically significant contributions on the gradient dynamics with arousal fluctuations (Fig. R10e).

The above result is an example using the CHRM3 expression profile, and the procedure is summarized in Fig. R10e and applied to all neuromodulatory receptors in Fig. S25. With the above procedure controlling the SA issue, our significance level (dashed lines in Fig. 6b) became more stringent, but still several neuromodulatory receptors exhibited statistical significance in our revised Fig. 6b. We have revised our method section (Page 54 Line 9) and added above results (Page 21 Line 1) in manuscript.

(2) We really appreciate this constructive comment. Per Reviewer's suggestion, we now added two new expression similarity matrices, one based on glutamate receptors and another one based on randomly selected genes (Fig. R11). Those glutamate receptor and random genes are listed in Supplementary File 1. Both of them showed significantly reduced contributions to arousal related dynamics (Fig. R11 c, g, e). Therefore, we can conclude that neuromodulatory receptor expression does specifically contribute to the arousal related dynamics.

(3) We fully agree that the overall gene expression may be related to the FC at the gradient level. However, such static relationship is not the main focus of the current study. Instead we tried to show 1) FC gradient strength is dynamic with arousal fluctuations, and 2) such FC gradient dynamics is related to the neuromodulatory receptors. The above analysis in Fig. R11 demonstrated that such contribution is likely to be more pronounced to neuromodulatory receptors, compared to the non-modulatory receptor of glutamate or randomly selected genes.

Now, we have revised our manuscript accordingly and added such control analysis in Page 19 Line 13.

R2C#6

First, this comment was made with respect to Fig 6. I thought that the R^2 in Fig 6a

represents the spatial similarity between the receptor maps and the FG maps. Now, with the clarification in their response, I understand that it represents the R^2 decrease with changing one receptor map in calculating NS.

But I do have some new questions regarding the new figures. Why is the cvR^2 for the control B model, which includes the SC term, close to zero (Fig. R11c) ? Why is the Fig R11g so different from the Fig. R9g? Are they for different random ISH gene maps? If yes, does it mean that the randomly selecting ISH gene may not be a good way of getting the control, give a big difference among different gene maps.

Figure R10. Multiple comparison for the unique contribution of each neuromodulatory receptor using spatial autocorrelation preserving surrogate maps.

(a) The CHRM3 receptor expression map and example surrogate maps with matched independent and preserving spatial autocorrelation (SA).

(b) Histogram of spatial correlation between surrogate maps and empirical one. SA preserving surrogate maps showed wider distribution of correlation with the empirical one, compared to the SA independent surrogate maps, e.g., CHRM3 receptor.

(c) Arousal relevant dynamics of the unique contribution for CHRM3 receptor. Similar inverted U shape relationship using SA preserving surrogate maps as shuffling, compared with SA independent surrogate maps. Errorbar, Mean \pm SEM, n=1000 times

shuffling.

(d) Pearson's correlation coefficients (right tail) between surrogate correlation and the variance of the unique contribution of CHRM3 receptor. If the correlation is not significant, it means random receptor maps could contribute similar arousal modulation, i.e., the empirical receptor does not specifically contribute to arousal dynamics. Alternatively, if the correlation is significant, it means larger spatial map shuffling causes larger loss of the unique contribution for the corresponding receptor, i.e., the empirical receptor does contribute to arousal dynamics. Each dot represented a random SA preserving shuffling of corresponding receptor ($n = 1000$ times).

(e) Revised criterion for identifying the receptors with significant contribution to the arousal relevant gradient dynamics.

Figure R11. Control analysis of the unique U-shape contribution from the neuromodulatory similarity. (Now Figure S24 in our revised Supplementary materials)

(a-b) The control analysis was conducted by replacing the NS with glutamate receptor similarity (a, GluS) or random ISH expression similarity (b, RS).

(c) Significant difference (pair-wise t-test) of cross-validated explained variance (cvR²) from the general linear model (GLM) across the original, control A and control B groups.

(d) Glutamate receptor similarity showed a significant correlation with the empirical

functional connectivity.

(e) U-shape relationship of arousal relevant unique contribution (ΔR^2) from the random ISH genetic gradient 1 & 2.

(f) Example of the random ISH genetic similarity which showed a significant correlation with empirical functional connectivity.

(g) Inverted U-shape relationship of arousal relevant unique contribution from the random ISH genetic gradient 1, with no arousal relevance on the gradient 2.

R2C#7

I often try not to raise new questions in the 2nd-round review. However, I just noticed that the arousal index was computed as single-frame fMRI correlations with a spatial template that was believed to reflect arousal-related brain activation pattern. Then, the two ends of the AI axis likely represent the fMRI time frames with strong positive and negative correlations with the template. My own experience is that the resting-state fMRI signals often show a similar pattern of deactivation right after the co-activations. Based on this, is it possible that the two types of time points (with positive and negative correlations to the template) are closer to each other than to the others? So, the dynamic FC for the low and high AI actually was actually derived from more overlapped time points, and thus the invert-U or U shape curves were observed along the AI axis. The authors may want to look into this.

R2C#1 (Reviewer 2's comment #1)

I truly appreciate the diligent work the authors have done to address my questions/comments in the first run. However, some of these new results raised more questions than answered. So, either I don't really understand the methods or the authors made fundamental mistakes in using and interpreting these models.

Response:

Thank you very much for your constructive comments on our manuscript, which we believe substantially improved our manuscript. We sincerely apologize that there were some errors and missing parts in our previous revision, which created confusions for readers. Now with the current revision following the reviewer's comments, we hope we solved those confusions.

R2C#2

How were those numbers obtained? They are inconsistent with the other results shown in the paper. These numbers are inconsistent with the % variance reported in other parts of the figure, e.g., Figs. 4b, 4c, 4f, 5b, 5d, that are much lower than those numbers. It seems that the % contributions in Fig 4e sums up to 100%, which cannot be true. The % contribution is often quantified by the explained % variance, which shouldn't be normalized! This needs to be corrected.

Response:

We really appreciate this suggestion. In our previous revision, we indeed normalized the explained variance, and this error is now corrected following the reviewer's suggestion. Fig.4e was revised accordingly (Fig. R1) in our revised manuscript (Page 16 Line 17).

Figure R1. Reduced model for investigating the unique distribution of structural connectivity and neuromodulatory inputs to the functional gradients.

Venn diagram based on the overlapping cross-validated explained variance (cvR^2) between structure connectivity or neuromodulatory receptor similarity (circle), compared with the empirical functional connectivity (FC). (Now Figure 4e in our revised manuscript).

R2C#3

I believe that the authors had some fundamental misunderstanding about the GLM. The authors keep saying that they "utilized the reduced model to capture the unique contribution of each variable". According to the description and also Fig R7f, they actually used the R^2 DIFFERENCE between the full model and the reduced model to capture the unique contribution of each variable. They are different! The only explanation I can think of is that the authors only used the "reduced model" to denote the R^2 difference between the full model and real reduced model. Then, my question is how the authors obtained those predicted gradients shown in Fig 4h. If they used

the predicted values in the reduced models, i.e., \widehat{FC}_a and \widehat{FC}_b in Fig R7f, it is totally wrong. If not, how did they get those maps?

Response:

First, we sincerely apologize for the confusion about the definition of the “reduced model” in our previous manuscript. The “reduced model” in our manuscript was not meant to refer to the “reduced model” as conventionally defined in linear regression, i.e., “restricted model” ($y_i = b_0 + \varepsilon_i$) vs. the full general linear model, i.e., “unrestricted model” ($y_i = b_0 + b_1x_{i1} + \varepsilon_i$). As Reviewer correctly pointed out, the reduced model in the current manuscript was defined as the difference between the full GLM model and the “randomly perturbed” GLM model ($y_i = b_0 + b_1\widetilde{x}_{i1} + \varepsilon_i$) in which a particular variable \widetilde{x}_{i1} was randomly shuffled 1000 times. This usage of the “reduced model” was adopted from a previous study (<https://doi.org/10.1038/s41593-019-0502-4>), in which unique contributions of various spontaneous behaviors to calcium imaging data in mice were dissected. We now revised the manuscript to clearly point out this difference (Results part: Page 16 Line 10; Methods part: Page 47 Line 16) to avoid confusion for readers.

Then, we apologize again for the confusing annotation in our previous Fig. R7f. In our previous manuscript, we emphasized the “R² difference” but did not mention how these predicted gradients were derived. In addition to the R² difference between the full model and reduced model (as mentioned in Fig. R7 in our previous revision), we also calculated the difference of predicted functional connectivity ($\Delta\widehat{FC}$) and decomposed the $\Delta\widehat{FC}$ into unique gradients of SC or NS via diffusion embedding mapping. This part is now added in the revised Fig. S22a.

Figure R2. Illustration of the full general linear model and the reduced model.

In the reduced model, the variable of interest was randomly shuffled 1000 times, and the resulting difference (compared to the full model) was the unique contribution of this particular variable in the reduced model. SC, structure connectivity; NS, neuromodulatory receptor similarity. (Now Supplementary Figure 22a in our revised manuscript).

R2C#4

Thanks for the new results that I suggested in the 1st-round of review. The NG2 doesn't look like the FG 2 (Fig. 1a), and they showed opposite signs in frontal and visual cortices). Something appears to be wrong for the statistical testing shown in Fig R8b (the second column in the bottom row). The area of the null distribution on the right side of the vertical lines is apparently larger than 0.01% of the total area. So, I don't know how those vertical lines (real correlations) would be corresponding to a p-value of $<10^{-4}$.

Response:

We are very grateful to reviewer for pointing out this issue. After close inspection, we found the p values were generated from *t* test in original results and were not replaced by *p* values using empirical null distributions of the spatial correlation between SA-dependent surrogate maps and empirical one. Accordingly, we modified our results about *p* values based on the empirical distribution of the Pearson's correlations between the empirical gradient and 1000 SA-preserving (blue) surrogate maps.

After correcting the *p* values, statistical significances of first four gradients did not change qualitatively compared to previous results in 1st round review, except for gradient 2 (from $p < 10^{-4}$ to $p = 0.062$ for ION dataset and 0.076 for NIH dataset). Now the non-significance agrees well with the visual comparison between NG2 and FG2. Nevertheless, this change doesn't affect our main conclusion.

We'd like to apologize again for this mistake and really appreciate reviewer's detailed reading.

Figure R3. Modified p-value provided a formally statistical evaluation using a proper hypothesis of the null distribution. (Now Supplementary Figure 21b in our revised manuscript).

Topographical similarity between neuromodulatory and functional gradients. Significant difference between the empirical similarity of neuromodulatory-functional gradients and null distributions, in which null distributions was derived from the Pearson's correlation coefficients (gray shades) between neuromodulatory gradients and the spatial autocorrelation preserving surrogate maps of functional gradients. Each dot represented a brain region. Red and blue lines represented the Pearson's correlation between empirical neuromodulatory and functional gradients.

R2C#5

Again, the new results confused me more.

(1a) First, what is the “explained variance %” in Fig 5b and 5d-e. (1b) Are those R^2 from the full and reduced models? If yes, they at most represent the contribution of SC and/or NS to FC, but not necessarily to the FC gradients, since we don't know which specific part of FC leads to the FC gradients. Similar to what I suggested in R2C3, I don't understand how those predicted gradients are related to this explained variance %. (2) Second, why did the Fig R9e-R9g use the “delta R^2 ” but not the explained variance % as shown in Fig 5. If the “delta R^2 ” between the full model and the control B model are around zero as shown in Fig R9g, does this mean that the ISH similarity also account for the FC variance just like NS? Or, the “delta R^2 ” is not the difference between the full model and control B. Again, I just want to say that these results are very confusing.

Response:

(1a and 2) The “explained variance %” (in Fig 5b and 5d-e) represented the percentage of the total variance explained by each gradient, which was derived from the predicted functional connectivity (FC) of GLM (Fig.5b) or reduced model (Fig. 5d-e).

We sincerely apologize for the wrong labelling of “explained variance %” and “ R^2 ” in our previous revision. The “delta R^2 ” in previous Fig. R9e-g should be the “delta explained variance %”. We have corrected the labelling in Fig.R4.

(1b) We must sincerely apologize again that such important information was missing about the computational details of gradients and corresponding ΔExpVar (Explained variance) using the reduced model. Firstly, for the full general linear model (GLM), we calculated the predicted FC across arousal bins, and obtained the first two gradients and corresponding ExpVars (Fig.R5, left panel). Because of the high spatial similarities of the gradients across arousal bins, we only presented the gradient profiles in mid-arousal level (bin=5). Then, for the unique contribution on functional gradients of SC (or NS), we conducted similar analyses to the full GLM. The unique gradient profile was derived from the difference ($\Delta\widehat{FC}$) between the full GLM model (\widehat{FC}) and the reduced model (\widehat{FC}_{SC}) (Fig.R5, right upper panel). Finally, the $\widehat{FC}_{SC_{1,2,\dots,10}}$ were further decomposed into a set of gradients and corresponding ExpVars. The unique contribution of explained variance (ΔExpVar) was the ExpVar difference between full GLM and reduced model (Fig.R5, right lower panel). We now have revised our manuscript in Page 16 Line 19 (Results part) and Page 48 Line 1 (Methods part).

Figure R4. Noise component and random ISH expression similarity did not drive inverted U-shape relationship on unique gradient dynamics with arousal fluctuation. (Now Figure S24 in our revised Supplementary materials)

(a) Significant correlation between structural connectivity (SC) and neuromodulatory similarity (NS), suggesting the overlap between SC and NS.

(b-c) Control analysis of the U-shape modulation for the neuromodulatory similarity (NS) The control analysis was conducted by adding a noise matrix (NM) as another variable (b) or replacing the NS with random ISH expression similarity (c, RS).

(d) Example of a random noise component which did not have a tight link with the empirical functional connectivity.

(e) No significant arousal relevant unique contribution from the noise component. Across arousal levels, the unique contribution \ was not significantly different from zero (one-sample t-test, two tails).

(f) Example of the random ISH genetic similarity which showed a significant correlation with the empirical functional connectivity.

(g) Inverted U-shape relationship of arousal relevant unique contribution from the random ISH genetic gradient 1, with no significant arousal relevance on the gradient 2.

R2C#6

First, this comment was made with respect to Fig 6. I thought that the R^2 in Fig 6a represents the spatial similarity between the receptor maps and the FG maps. Now, with the clarification in their response, I understand that it represents the R^2 decrease with changing one receptor map in calculating NS.

But I do have some new questions regarding the new figures. (1) Why is the cv R^2 for the control B model, which includes the SC term, close to zero (Fig. R11c)? (2) Why is the Fig R11g so different from the Fig. R9g? Are they for different random ISH gene maps? If yes, does it mean that the randomly selecting ISH gene may not be a good way of getting the control, give a big difference among different gene maps.

Response:

(1) Thank you very much for the careful check. After close inspection, we found 6 abnormal gene expressions patterns in the estimation of random ISH similarity matrix in 1st round review (Fig. R6a). The ISH images of those 6 genes were severely damaged and now is excluded in the current results. We must sincerely apologize for our negligence.

The abnormal ISH expression patterns led to a large amount of regions with zero expression level, resulting excessive global positive correlation coefficients across brain regions (Fig. R6b). After excluding the above 6 abnormal genes, we obtained the modified random ISH similarity matrix (Fig. R6c, 2nd round review).

Replaced by the revised ISH similarity matrix (Fig. R6c), the cvR^2 for the control B model significantly increased (Fig. R6d). The major reason of “close to zero” cvR^2 in our previous revision may attribute to the over regularization with the improper penalty factor ($\lambda = 0.01$). Throughout our analyses, we optimized the selection of penalty factor ($\lambda = 10^9$ to 10^{-20}) and chose the $\lambda = 0.01$ to solve noise impact using least-squares regression. However, the penalty factor $\lambda = 0.01$ was not suitable for the abnormal ISH similarity matrix (1st round review), resulting in under fitting of GLM and reduced model (Fig. R6e). Now we have revised previous results in Fig. R6f. In addition to the modified ISH similarity (Control B in 2nd round review), we also systematically re-checked and ensured that the $\lambda = 0.01$ was an appropriate penalty factor for other combinations of model variables using GLM and reduced models.

(2) Firstly, we are very sorry again for the improper usage of “explained variance %” and “ R^2 ” (same as above R2C#5) and have made corresponding corrections in Fig. R7.

Next, indeed two different groups of random ISH gene maps were used in Fig. R11g and Fig. R9g in our previous revision, which was not our intention but rather a difference between results of two versions. While the numbers were quantitatively different, they were qualitatively similar as they were all much lower than the results of NS. As we discovered the issues with 6 abnormal genes in Fig. R11g as discussed above, we now used the same gene list (revised Supplementary Table 4) to avoid confusion and revised the results (Page 19 Line 3) in Fig.R7.

arisen from the penalty factor. The constant penalty factor λ was chosen to be 0.01 throughout our analysis. The aberrant ISH similarity (in 1st round review) resulted in under fitting of GLM and reduced model using the least-square method. After we rejected these abnormal genetic expression patterns, the modified ISH similarity (in 2nd round review) exhibited higher explained variances with $\lambda = 0.01$.

(f) Significant difference (pair-wise t-test) of cross-validated explained variance (cvR²) from the general linear model (GLM) across the original, control B and control C groups. (Now Supplementary Figure 24j in our revised manuscript)

Figure R7. Control analysis of the unique U-shape contribution from the neuromodulatory similarity. (Now Supplementary Figure 24a-i in our revised

manuscript)

(a-c) The control analysis was conducted by (a) adding a noise matrix (NM) as another variable, (b) replacing the NS with random ISH expression similarity (RS) and (c) replacing the NS with glutamate receptor ISH expression similarity (GluS).

(d) Example of a random noise component which did not have a tight link with the empirical functional connectivity.

(e) No significant arousal relevant unique contribution from the noise component. Across arousal levels, the unique contribution was not significantly different from zero (one-sample t-test, two tails).

(f) Example of the random ISH genetic similarity which showed a significant correlation with empirical functional connectivity.

(g) Inverted U-shape relationship of arousal relevant unique contribution from the random ISH genetic gradient 1, with no arousal relevance on the gradient 2.

(h) Glutamate receptor similarity showed a significant correlation with the empirical functional connectivity.

(i) U-shape relationship of arousal relevant unique contribution from the random ISH genetic gradient 1 & 2.

R2C#7

I often try not to raise new questions in the 2nd-round review. However, I just noticed that the arousal index was computed as single-frame fMRI correlations with a spatial template that was believed to reflect arousal-related brain activation pattern. Then, the two ends of the AI axis likely represent the fMRI time frames with strong positive and negative correlations with the template. My own experience is that the resting state fMRI signals often show a similar pattern of deactivation right after the co-activations. Based on this, is it possible that the two types of time points (with positive and negative correlations to the template) are closer to each other than to the others? So, the dynamic FC for the low and high AI actually was actually derived from more overlapped time points, and thus the invert-U or U shape curves were observed along the AI axis. The authors may want to look into this.

Response:

We appreciate this valuable suggestion and fully understand the reviewer's concern. Firstly, in the current work, dynamic FCs were estimated using the dynamic conditional connectivity (DCC) strategy, rather the sliding window analysis (<https://doi.org/10.1016/j.neuroimage.2014.06.052>).

Then, in our results, we found the averaged minimum interval between extreme low (bin=1) and high (bin=10) arousal level was 25.5 s, suggesting two types of time points (with positive and negative correlations to the template) were temporally far away from each other and such gap was larger than other arousal bins (Fig. R8a).

Nevertheless, there may still be some temporal relationship among adjacent time points, as reviewer suggested. To address this concern, we sampled frames with 10 s (or 20 s) gap (i.e., picking the first frame every 10 s or 20 s) and then binned and averaged to obtain averaged FCs (and subsequent gradients) across arousal bins. The averaged minimum intervals after 10 s (or 20 s) sampling showed larger intervals (10

s sampling: $66.5 \text{ s} \pm 4.9 \text{ s}$; 20 s sampling: $116.0 \text{ s} \pm 7.3 \text{ s}$, mean \pm std.) among arousal bins (Fig. R8b-c). This sampling strategy largely avoided potential issues with temporally overlapped or adjacent frames, and the results of gradient dynamics (Fig. R8d-e) were very similar to our original results as in Fig. 3. We have add the control analysis in our manuscript (Page 13 Line 2).

Therefore, it is less likely that the temporally overlapped time points (which typically happens in sliding window analysis) would contribute to our observed gradient dynamics.

Large temporal gaps (25.5 s) was shown between extremely low (bin=1) and high (bin=10) arousal levels.

(b-c) Apparent increases of averaged minimum temporal intervals among different arousal bins with 5 times (b) or 10 times (c) down-sampled time points.

(d-e) Consistent inverted U-shape relationship between gradient explained variances and 5 times (d) or 10 times (e) down-sampled time points. (Top) inverted U-shape relationship between arousal level and explained variance across gradients. (Bottom) highly stable gradient profiles across arousal levels.

REVIEWER COMMENTS

Reviewer #2 (Remarks to the Author):

I thank the authors for their diligent work in addressing my comments/questions. I am also glad that my comments helped them identify several errors in the paper. I only have a few minor suggestions as listed below

1. The FCsc_hat in the middle panel of Figure R2 should be FCns_hat, since it is the predicted FC from NS with shuffled SC. Its difference from the full model FC_hat would be more appropriately named as delta_FCsc_hat. The same rationale applies to the FCns_hat in the right panel of the Figure R2, as well as the FCsc_hat in Figure R5.
2. The bottom of Figure R5 suggested that the delta_ExpVarsc was derived for Figure 5d and 5e. However, the labels of y-axis of these two panels were shown as Explained Variance rather than delta Explained Variance.

Reviewer #2 (Remarks to the Author):

I thank the authors for their diligent work in addressing my comments/questions. I am also glad that my comments helped them identify several errors in the paper. I only have a few minor suggestions as listed below

1. The FC_{sc_hat} in the middle panel of Figure R2 should be FC_{ns_hat} , since it is the predicted FC from NS with shuffled SC. Its difference from the full model FC_{hat} would be more appropriately named as ΔFC_{sc_hat} . The same rationale applies to the FC_{ns_hat} in the right panel of the Figure R2, as well as the FC_{sc_hat} in Figure R5.

Response:

We really appreciated the Reviewer's comment and revised Supplementary Figure 22a-b (as Figure R2 and R5 in the previous revision) accordingly in our manuscript.

2. The bottom of Figure R5 suggested that the $\Delta ExpVarsc$ was derived for Figure 5d and 5e. However, the labels of y-axis of these two panels were shown as Explained Variance rather than Δ Explained Variance.

Response:

Thanks a lot for Reviewer's careful reading. Per Reviewer's suggestion, we corrected the labels of y-axis of Figure 5d and 5e, respectively.